# Coral microbiomes as reservoirs of unknown genomic and biosynthetic diversity





Fabienne Wiederkehr[1,30], Lucas Paoli[1,2,30]✉, Daniel Richter[3,4,30], Dora Racunica[3], Hans-Joachim Ruscheweyh[1], Martin Sperfeld[1], James O'Brien[1], Samuel Miravet-Verde[1], Alena B. Streiff[3], Jessica Ransome[3], Clara Chepkirui[3], Taylor Priest[1], Anna Sintsova[1], Guillem Salazar[1,5], Kalia S. I. Bistolas[6], Teresa Sawyer[7], Karine Labadie[8], Kim-Isabelle Mayer[9], Aude Perdereau[8], Maggie M. Reddy[10,11], Clémentine Moulin[12], Emilie Boissin[13], Guillaume Bourdin[14], Juliette Cailliau[12], Guillaume Iwankow[13], Julie Poulain[8,15], Sarah Romac[16], Tara Pacific Consortium Coordinators*, Serge Planes[13,15], Denis Allemand[17,18], Sylvain Agostini[19], Chris Bowler[20], Eric Douville[21], Didier Forcioli[17,22], Pierre E. Galand[23], Fabien Lombard[15,24], Pedro H. Oliveira[8], Olivier P. Thomas[10], Rebecca Vega Thurber[6,25], Romain Troublé[12,15], Christian R. Voolstra[9], Patrick Wincker[8,15], Maren Ziegler[26], Jörn Piel[3]✉ & Shinichi Sunagawa[1]✉

Coral reefs are marine biodiversity hotspots that provide a wide range of ecosystem services[1]. They are reservoirs of bioactive metabolites, many produced by microorganisms associated with reef invertebrate hosts[2]. However, for the keystone species of coral reefs—the reef-building corals—we still lack a systematic assessment of their microbially encoded biosynthetic potential and the molecular resources at stake due to the alarming decline in reef biodiversity. Here we analysed microbial genomes reconstructed from 820 reef-building coral samples of three representative coral genera collected at 99 reefs across 32 islands throughout the Pacific Ocean (*Tara* Pacific expedition)[3]. By contextualizing our analyses with the microbiomes of other reef species, we found that only 10% of the 4,224 microbial species and less than 1% of the 645 species exclusively identified in *Tara* Pacific samples had genomic information available. Furthermore, the biosynthetic potential of reef-building coral microbiomes rivalled or surpassed that of traditional natural product sources such as sponges. Among the biosynthetically rich bacteria in the reef microbiome, we identified new groups of Acidobacteriota that encode previously unknown enzymology, in turn opening promising avenues for functional protein engineering. Together, this study underscores the importance of conserving coral reefs as vital reservoirs of molecular diversity.

Coral reefs are one of the most biodiverse and productive ecosystems on Earth. Despite covering less than 0.2% of the ocean floor, they are home to a third of all named marine multi-cellular species[4,5]. Coral reefs provide a wide range of ecosystem services, such as food, livelihoods and coastal protection, to millions of people around the globe, and they serve as a source of bioactive metabolites[1,2]. However, climate change, emerging diseases and other anthropogenic stressors have caused a decline in live coral cover of more than 50% since the 1950s[1,6]. Given the projections of further reef decline[7], there is a pressing need to capture what is at stake under this continued biodiversity loss.

The biodiversity, productivity and structural complexity of reef systems are linked fundamentally to the ecological functions provided by calcareous skeleton-forming (reef-building) corals, such as stony and fire corals[1]. Like other organisms, these sessile invertebrates

[1]Department of Biology, Institute of Microbiology and Swiss Institute of Bioinformatics, ETH Zürich, Zürich, Switzerland. [2]Global Health Institute, School of Life Sciences, EPFL Lausanne, Lausanne, Switzerland. [3]Department of Biology, Institute of Microbiology, ETH Zürich, Zürich, Switzerland. [4]Medical Research Council Laboratory of Molecular Biology, Cambridge, UK. [5]Institute for Integrative Systems Biology I2SysBio, Universitat de València-CSIC, Paterna, Spain. [6]Department of Microbiology, Oregon State University, Corvallis, OR, USA. [7]Electron Microscopy Facility, Oregon State University, Corvallis, OR, USA. [8]Génomique Métabolique, Genoscope, Institut François Jacob, CEA, CNRS, Université Evry, Université Paris-Saclay, Evry, France. [9]Department of Biology, University of Konstanz, Konstanz, Germany. [10]School of Biological and Chemical Sciences, Ryan Institute, University of Galway, Galway, Ireland. [11]Department of Biological Sciences, University of Cape Town, Cape Town, South Africa. [12]Fondation Tara Océan, Base Tara, Paris, France. [13]PSL Research University: EPHE-UPVD-CNRS, UAR 3278 CRIOBE, Laboratoire d'Excellence CORAIL, Université de Perpignan, Perpignan, France. [14]School of Marine Sciences, University of Maine, Orono, ME, USA. [15]Research Federation for the Study of Global Ocean Systems Ecology and Evolution, FR2022/Tara GOSEE, Paris, France. [16]Sorbonne Université, CNRS, Station Biologique de Roscoff, AD2M, UMR 7144, ECOMAP, Roscoff, France. [17]Laboratoire International Associé Université Côte d'Azur-Centre Scientifique de Monaco (LIA ROPSE), Monaco, Principality of Monaco. [18]Centre Scientifique de Monaco, Monaco, Principality of Monaco. [19]Shimoda Marine Research Center, University of Tsukuba, Shizuoka, Japan. [20]Institut de Biologie de l'École Normale Supérieure (IBENS), École Normale Supérieure, CNRS, INSERM, Université PSL, Paris, France. [21]Laboratoire des Sciences du Climat et de l'Environnement (LSCE), CEA, CNRS, UVSQ, Université Paris-Saclay, Gif-sur-Yvette, France. [22]Institute for Research on Cancer and Aging in Nice (IRCAN), Université Côte d'Azur, CNRS, INSERM, Nice, France. [23]Sorbonne Université, CNRS, Laboratoire d'Ecogéochimie des Environnements Benthiques (LECOB), Observatoire Océanologique de Banyuls, Banyuls-sur-Mer, France. [24]Sorbonne Université, Institut de la Mer de Villefranche, Laboratoire d'Océanographie de Villefranche, Villefranche-sur-Mer, France. [25]Marine Science Institute, UC Santa Barbara, Santa Barbara, CA, USA. [26]Department of Holobiont Biology, Justus Liebig University Giessen, Giessen, Germany. [30]These authors contributed equally: Fabienne Wiederkehr, Lucas Paoli, Daniel Richter. *A list of authors and their affiliations appears at the end of the paper. ✉e-mail: lucas.paoli@epfl.ch; jpiel@ethz.ch; ssunagawa@ethz.ch

depend on a diverse community of microorganisms (microbiome)[8,9]. The microbiome of corals provides its host with vital nutrients, such as carbon, nitrogen and phosphorus, as well as vitamins and essential amino acids[10]. Furthermore, it supports its host in coping with changing environmental conditions[11] and can protect it from infectious diseases[12]. Microbes associated with reef-building corals are suggested to produce bioactive metabolites to fend off pathogens, predators and competitors[12]. However, although diverse metabolites such as anti-microbial[13], anti-inflammatory[14] and anti-tumour[15] agents have been discovered (with some undergoing clinical trials[16]) in other reef invertebrates[2] such as sponges and soft corals, little is known about the bioactive potential of reef-building coral microbiomes.

Historically, the discovery of bioactive metabolites has relied on screening chemical extracts from either the invertebrate host, which may depend on the supply of unsustainable amounts of animal biomass[17], or from microbial producers isolated from their hosts[15]. The latter approach is, however, constrained by our limited ability to cultivate most microorganisms under standard laboratory conditions[18]. Furthermore, both methods are prone to the persistent challenge of rediscovering the same or similar metabolites[19]. As a more recent strategy, metabolic pathways linked to biosynthetic gene clusters (BGCs) that encode the synthesis of bioactive compounds can be discovered by screening reconstructed genome sequences[20]. These genomes may originate from microbial culture collections[18] as well as from uncultivated single cells or whole microbial communities (metagenomes) from, in principle, any environment or host organism[21–24]. However, for the microbiome of reef-building corals, such genomic information remains scarce[18,25].

We thus aimed to explore systematically the genome-resolved diversity, host-specificity and BGC-encoded biosynthetic potential of reef-building coral microbiomes, compare them with the microbiomes of other reef hosts (such as sponges) and the surrounding environment, and determine whether corals host any BGC-rich lineages as promising biotechnological targets. To this end, we reconstructed more than 13,000 metagenome-assembled genomes (MAGs) from reef-building coral samples collected as part of the *Tara* Pacific expedition[3] (Supplementary Table 1) and from publicly available coral reef metagenomic datasets (Supplementary Table 2). For almost 90% of the 4,224 microbial species in total, or more than 99% of those from *Tara* Pacific samples for which we reconstructed MAGs, no genome-resolved information was available previously. Coral and sponge microbiomes were largely host-specific and we found the biosynthetic potential (per microbial species) in reef-building corals (particularly fire corals) to be as rich as, or even richer than, that of sponges or the surrounding waters. By detecting new, biosynthetically rich bacterial lineages and characterizing unusual enzymology and bioactive compounds within coral-associated Acidobacteriota[26] spp., our work clearly underscores not only the value of reef-building corals from a biotechnological perspective, but also the implications of their potential loss.

## Coral reef microbiome genomic resources

To fill the gap in the availability of microbial genome data from reef-building corals, we collected 820 metagenomes from two stony coral genera (*Porites* and *Pocillopora*) as well as fire corals (*Millepora*) around 32 islands (99 reefs) throughout the Pacific Ocean as part of the Tara Pacific expedition[3] from 2016 to 2018 (Fig. 1a,b; Supplementary Table 1; for details on coral host lineages, see ref. 27). To facilitate a comprehensive assessment of the phylogenomic novelty and biosynthetic potential of reef-building coral microbiomes, and to contextualize this information across different coral reef-inhabiting species (such as sponges and soft corals), we supplemented the dataset with existing metagenomes from 412 coral samples (from 29 genera, including 22 stony and five soft coral genera, as well as from black and fire corals) and 371 sponge samples (from 32 genera) (Fig. 1c,d; Extended Data Fig. 1a and Supplementary Table 2).

Applying a previously benchmarked bioinformatic workflow[24] to this dataset, we reconstructed 13,446 coral- and sponge-associated MAGs from bacteria and archaea (Fig. 1e and Extended Data Fig. 1b). Of the 2,046 coral-associated MAGs, 1,964 were from reef-building corals and 1,524 originated from the *Tara* Pacific expedition[28]. With 57% (1,171) of all coral-associated MAGs, fire corals contributed more microbial genomes than stony (39%; 793) and soft (4%; 72) corals combined (Supplementary Table 3). Focusing on the *Tara* Pacific samples (equal sampling effort across coral hosts), we reconstructed 1,170 genomes from *Millepora* samples, 305 from *Porites* samples and 48 from *Pocillopora* samples. Although these numbers do not necessarily reflect microbial species richness, they are congruent with the observed differences across these hosts based on 16S rRNA gene sequencing results[9]. Furthermore, linking these genomic data to transmission electron microscopy (TEM) images, we found the high number of MAGs from fire corals to correspond to a high load of extracellular microorganisms (Fig. 1b and Extended Data Fig. 1c).

Overall, our efforts increased the number of available coral-associated MAGs tenfold (Supplementary Table 3). The data analysed in this study, which also includes 103 available isolate genomes (Supplementary Table 4), are referred to as the Reef Microbiomics Database (RMD) (Fig. 1f; Methods; 'Data availability').

## Genomic novelty and host-specificity

Establishing the RMD enabled us to capture systematically the number of microbial species we had reconstructed genomes for and to assess the degree to which these species lacked previous genomic information. Specifically, we annotated all genomes using the Genome Taxonomy Database (GTDB; r207) toolkit and clustered genomes using a 95% whole-genome average nucleotide-identity threshold[29], which defined 4,224 species-level clusters (species). Close to 90% (3,774) of all species, and 99% (638) of the 645 identified in *Tara* Pacific corals, were not present in the GTDB (Fig. 2a and Supplementary Table 3). Three-quarters of the species represented in the RMD had their genomes reconstructed from sponge metagenomes (3,206 versus 971 reconstructed from coral metagenomes) and less than 1% (36) of all microbial species were shared between sponge and coral metagenomes (Fig. 2a and Supplementary Table 3). Within corals, we reconstructed genomes from 516, 460 and 26 microbial species from stony, fire and soft corals, respectively (Fig. 2b). Stony and fire corals shared as few as 37 microbial species, and no species were shared with soft corals. More specifically, 95% of microbial species were unique to a particular host genus (Supplementary Table 3).

Beyond reconstructing genomes, we sought to validate this pervasive host-specificity by comparative compositional profiling of the host-associated microbial species identified here. To this end, we used a single-copy-marker-gene-based method[30] to determine the species-level taxonomic composition of all (1,603) host-associated metagenomes, 387 seawater metagenomes collected during the *Tara* Pacific expedition[28] (Fig. 1b) and 84 seawater metagenomes from previous coral- and sponge-focused studies that we included in our analysis (Methods). Overall, the microbial species profiles of coral metagenomes were not only distinct from those of seawater metagenomes (Fig. 2c and Extended Data Fig. 1d,e) but were also host specific (Fig. 2d).

Finally, we used the *Tara* Pacific dataset to test the detectability of coral-associated microorganisms in seawater sampled at increasing distances from their host, that is, coral-surrounding water, reef water and open ocean water. We detected as little as 20% of all microbial species from the coral metagenomes in the water samples. Furthermore, both the number and relative abundance of the detected species decreased with distance from the sampled coral colony within the reef, and they were barely detected in the open ocean (Fig. 2e).

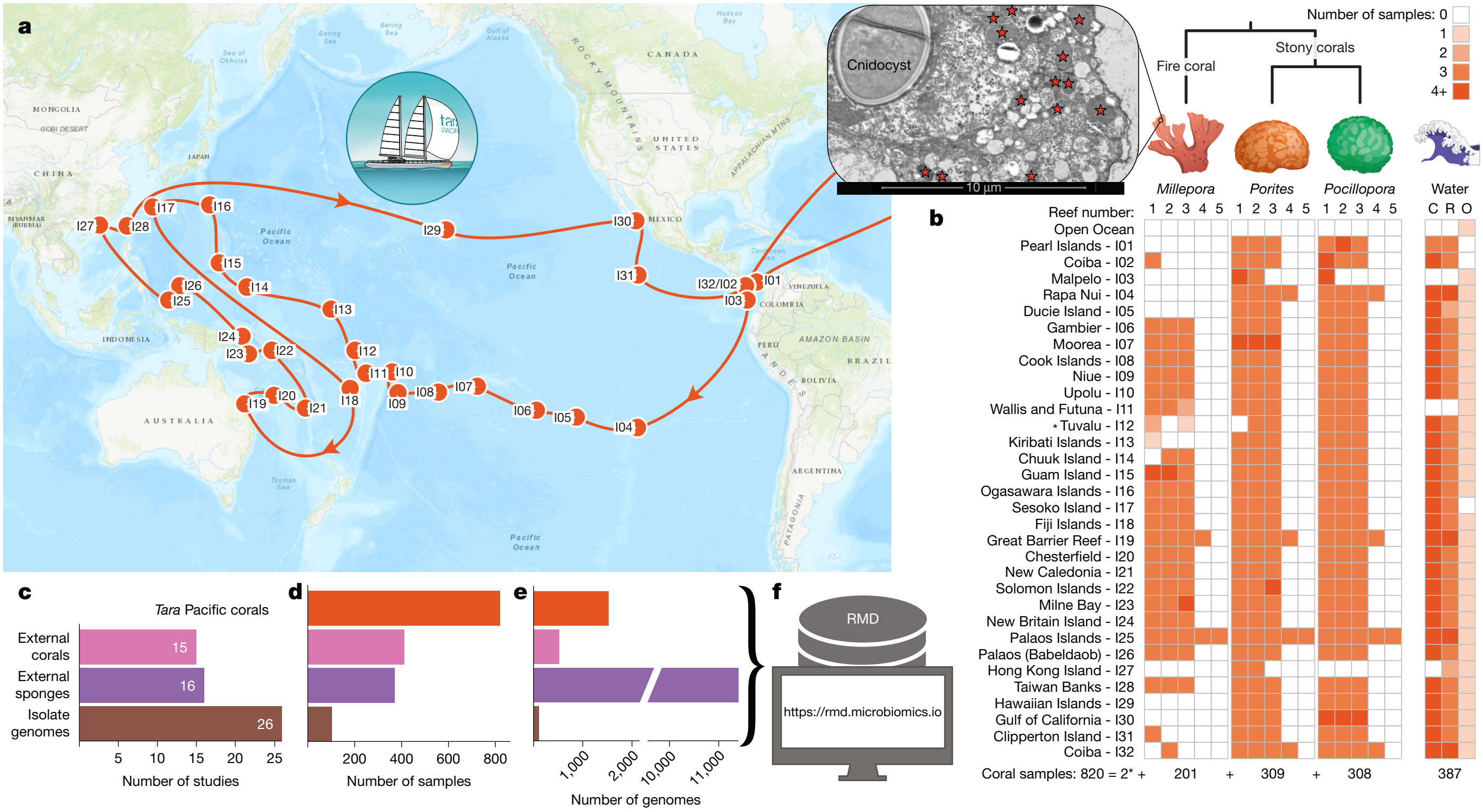


**Fig. 1 | The *Tara* Pacific expedition sampled in coral reefs across the Pacific Ocean. a**, The *Tara* Pacific expedition (2016–2018)[3] included the sampling of corals at 99 reefs across 32 islands throughout the Pacific Ocean. **b**, At each reef, *Millepora*, *Porites* and *Pocillopora* colonies (asterisk, plus (exceptionally) two *Heliopora* specimens at one reef in Tuvalu) were sampled[28], resulting in a total of 820 reef-building coral-associated metagenomes (Supplementary Table 1). In addition, the plankton microbiome was collected from the water surrounding *Pocillopora* colonies, from representative water within each reef, as well as from oceanic water[28], resulting in 387 metagenomes (Supplementary Table 1). C, coral-surrounding water; R, reef water; O, open ocean water. Inset, TEM image of a *Millepora* tissue sample, with bacteria-sized cells in the ectoderm indicated by stars (Extended Data Fig. 1c; Methods). **c**,**d**, To contextualize the data generated from the *Tara* Pacific expedition, we aggregated a total of 412 coral (from 29 genera) and 371 sponge (from 32 genera) metagenomes from 15 and 16 publicly available datasets, respectively (Extended Data Fig. 1a and Supplementary Table 2), showing number of studies (**c**) and number of samples (**d**). **e**, From this metagenomic dataset, we reconstructed 13,446 MAGs, of which 1,524 were from *Tara* Pacific metagenomes (mostly from fire corals). In addition, we collected 103 isolate genomes from 26 studies (Supplementary Table 4). **f**, All genomic data were compiled to generate the RMD at https://rmd.microbiomics.io. Map in **a** adapted from Esri (https://server.arcgisonline.com/arcgis/rest/services/World_Topo_Map/MapServer). Imagery in this work is owned by Esri and its data contributors and are used herein with permission. 

Together, our results complement previous metagenomic efforts to reconstruct microbial genomes from diverse environments[22], greatly expand the number and diversity of reef host-associated microbial taxa with genomic information available and—besides enabling us to investigate their functional potential—support earlier marker-gene-based reports[8,9,31] by demonstrating a high degree of host-specificity based on genome-resolved data.

## Functional and biosynthetic potential

To determine whether the phylogenomic novelty of the reef microbiome also correlated with an uncharted functional and biosynthetic potential, we used previously established methods[32] to generate a reef microbial gene catalogue based on the protein-coding genes predicted for all genomes from the 4,224 species represented in the RMD. This resulted in 16.3 million non-redundant genes (genes) with a mean of 3,857 genes per species. By comparison, a recently published microbial gene catalogue from the open ocean derived from the initial version of the Ocean Microbiomics Database (OMD)[24], built from almost twice as many microbial species (8,304), contained substantially fewer genes per species (2,135). The higher gene richness encoded by reef host-associated microbial species was also reflected in a larger estimated genome size per species (3.6 versus 2.2 Mbp for the reef versus ocean microbiome; Extended Data Fig. 2a). This difference might stem from lower streamlining pressure or, vice-versa, higher gene-maintenance pressure acting on microbial genomes in the nutrient-rich, complex host-associated ecological niche[33,34]. Supporting the host-associated nature of these organisms, compared with the OMD, we identified a substantial enrichment of eukaryotic-like proteins (ELPs), which have been implicated in host infection and symbiosis establishment[35,36], in the RMD (Extended Data Fig. 2b). Finally, compared with the open ocean microbiome, the reef microbiome contained a higher fraction (34% versus 16%) of as-yet uncharacterized genes (Fig. 3a), further highlighting the novelty of the genomic information uncovered in this study.

For the microbial genome collection in the RMD, we predicted BGCs with antiSMASH (35,000 BGCs from 13,000 genomes) to map and contextualize the reef biosynthetic potential and to compare its magnitude and diversity with that found in open ocean microbial genomes (40,000 BGCs from 35,000 genomes; Methods). To ensure comparability, we grouped the antiSMASH-predicted BGCs into biosynthetic gene cluster families (GCFs), identified pathways predicted to encode similar natural products[37] and accounted for the inherent redundancy (the same BGC encoded in several genomes) as well as the fragmented nature of metagenomic assemblies (Methods)[38]. We found that the reef microbiome encoded a richer (6,612 GCFs or 1.57 GCF per species) biosynthetic potential than the ocean microbiome (5,877 GCFs or 0.71 GCF per species) (Fig. 3b and Extended Data Fig. 2c). The two environments shared less than 10% of GCFs (411), and the proportions of the detected natural product classes also varied. For example, the reef

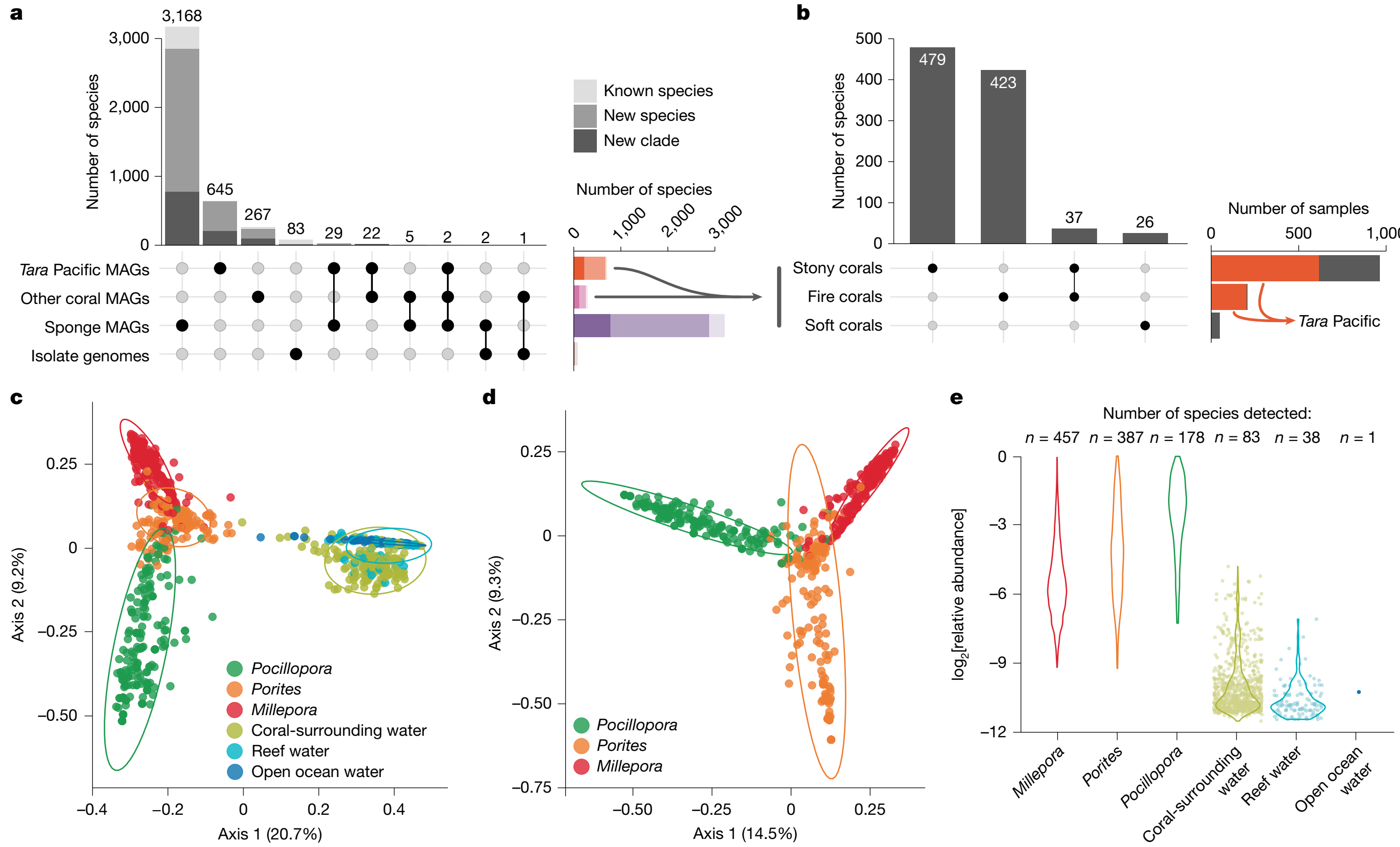


**Fig. 2 | The microbiome of reef invertebrates is genomically uncharacterized and host-specific.** **a**, By grouping the 13,549 genomes in the RMD at the species-level (95% average nucleotide identity), we found them to represent 4,224 microbial species. Taxonomic annotation with the GTDB (r207) showed that close to 90% (3,774) of all species, and 99% (638) of those reconstructed from *Tara* Pacific corals, represented new species or clades at higher taxonomic ranks. **b**, The species displayed high specificity for the different types of coral (stony, fire and soft). **c**, A Jaccard distance-based PCoA of microbial species detected across the *Tara* Pacific coral and seawater metagenomes showed a clear separation between the microbiomes of corals (*Pocillopora* $n = 207$, *Porites* $n = 177$, *Millepora* $n = 201$) and seawater (coral-surrounding water $n = 181$, reef water $n = 95$, open ocean water $n = 111$) (PERMANOVA, $P$ value $\leq 0.001$, $R^2 = 0.31$). **d**, Likewise, the coral metagenomes from the three coral genera (*Pocillopora*, *Porites*, *Millepora*) targeted during the *Tara* Pacific expedition differed significantly (PERMANOVA, $P$ value $\leq 0.001$, $R^2 = 0.18$). **e**, The number and abundance of coral-associated microbial species decreased in seawater samples as a function of distance to the coral host. Individual data points are shown for groups containing fewer than 100 data points.

microbiome contained a higher proportion of ribosomally synthesized and post-translationally modified peptides (RiPPs) (0.31 versus 0.174) and type I polyketide synthases (PKS) (T1PKS) (0.107 versus 0.036), and a lower share of terpenes (0.215 versus 0.282) and other classes (0.161 versus 0.279), including aryl polyene, homoserine, beta lactones or siderophores (Fig. 3c). These shifts may represent adaptations in the metabolism of reef invertebrate-associated compared with free-living microorganisms, and contrast previous reports that found relatively fewer ribosomal natural product pathways in sponges compared with aquatic ecosystems[22] (Supplementary Table 5).

Comparing the GCFs detected in coral reef microbiomes with biochemically characterized (MIBiG) and predicted (BiG-FAM) pathways (Methods), we found as little as 0.3% (21) overlap (Extended Data Fig. 2c) and 64% to be newly identified in this study (Fig. 3d and Extended Data Fig. 2d). To further support the notion that reef host-associated microorganisms represent a rich and unique source to discover new biosynthetic enzymes and natural products, we tested whether GCFs found in coral-associated microorganisms were also detected in the reef seawater, which was the case for only 25% of them (which echoes the 20% overlap between corals and seawater microbiomes at the level of microbial species).

Given that reef-building corals, unlike sponges (followed by soft corals), have received relatively little attention as sources of marine natural products[39], we compared the biosynthetic potential in the RMD across these different groups of hosts. Overall, we recovered fewer GCFs from corals than from sponges (2,781 versus 3,920 GCFs; Fig. 3e), primarily because of differences in effective sequencing depth (Extended Data Fig. 2e) and host genera sampled (Extended Data Fig. 2f). However, after we normalized by sampling effort (including sequencing depth and number of host genera sampled; Methods), the coral microbiome was richer in GCFs both per genome (1.4 versus 0.3 GCFs per genome) and per microbial species (2.9 versus 1.2 GCFs per species). In addition, although the coral microbiome encoded fewer ribosomal natural product pathways (25% of all GCFs in corals compared with 36% in sponges), it harboured a larger share of the biotechnologically important class of non-ribosomal peptide synthetases[40] than the sponge microbiome (25% compared with 9%). To further contextualize our predictions, we searched the MarinLit database[41] for natural products isolated from any of the 15 coral genera (Supplementary Table 6) for which we predicted microbially encoded BGCs. The few identified coral-derived bioactive compounds belong exclusively to terpenes and alkyne fatty acids. Moreover, for terpenes from *Eunicella* and *Heliopora*, previous studies have demonstrated that the corresponding terpene synthases are of animal (and thus, host) origin[42], emphasizing the limited knowledge of microbial contributions to coral-derived natural products and positioning our BGC resource as a key tool for uncovering new ones, along with the biosynthetic enzymes required for their synthesis.

Among corals, our analyses revealed the fire coral microbiome to be particularly rich in biosynthetic pathways. We detected almost twice the number of GCFs per species in fire corals (4.0 GCF per species)

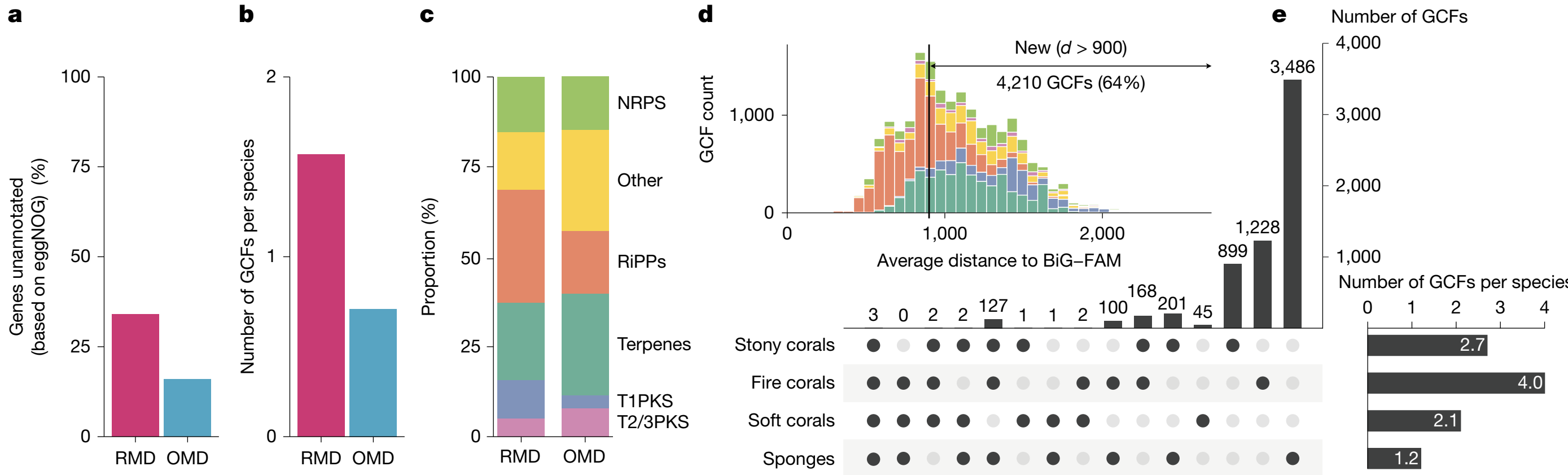


**Fig. 3 | The reef microbiome harbours a high degree of functional and biosynthetic novelty. a**, The microbiomes of reef invertebrates represented in the RMD harboured 18% more uncharacterized genes than the global open ocean microbiome (RMD, 16.3 million gene representatives; OMD, 17.7 million gene representatives). **b**, The RMD encoded more than twice the number of GCFs per species (1.57 GCFs per species) compared with the OMD (0.71 GCFs per species). **c**, The proportion of predicted natural products for the reef and open ocean microbiomes reveals an increased representation of RiPPs and T1PKS in the reef microbiome. NRPS, non-ribosomal peptide synthetases; T2PKS, type II PKS; T3PKS, type III PKS. **d**, Most GCFs from the reef microbiome are new compared with previously sequenced BGCs (BiG-FAM; Extended Data Fig. 2d; Methods). **e**, GCFs detected in reef invertebrates are largely specific to the different host types (stony corals, fire corals, soft corals and sponges). With 4.0 GCFs per species and 2.7 GCFs per species encoded in the microbiomes of fire and stony corals, respectively, reef-building corals are particularly BGC-rich.

than in soft corals (2.1 GCF per species), with stony corals (2.7 GCF per species) in between (Fig. 3e). Although our results indicate that the number of GCFs detected is still increasing equally with each new host (Extended Data Fig. 2f) or island (Extended Data Fig. 2g) sampled, and although the RMD contains relatively few genomes from soft corals (72 MAGs) and other coral reef species, our results highlight that the microbiome of reef-building corals, and fire corals in particular, encodes an immense and yet untapped source of new biosynthetic enzymes and natural products.

## BGC-rich microbial lineages in corals

Biosynthetically rich microorganisms (BGC-rich; at least 15 BGCs) are particularly valuable targets for isolating natural products. Such talented producers (super-producers), often identified through cultivation-independent approaches, hold tremendous biotechnological promise[24,43]. To identify BGC-rich microbial lineages in our reef dataset, we screened the RMD and, after manual inspection (Methods), identified 20 candidate BGC-rich microbial species (two times more than in the OMD[24]) from five phyla, including Acidobacteriota, Proteobacteria and Cyanobacteria (Fig. 4a and Supplementary Table 7). Notably, none of these BGC-rich species were represented in the GTDB (r207), suggesting them as candidate new species (11 of these even candidate new genera and one a candidate new family). Across the studied reef hosts, reef-building corals, in particular fire corals, harboured the highest proportion of BGC-rich lineages (Extended Data Fig. 3a).

To validate the BGC content predicted by short-read sequencing, we ranked the microbial species by the number of complete BGCs and followed-up on the top three species, all members of the phylum Acidobacteriota (Fig. 4a)—Thermoanaerobaculia sp. (MAG48), Acidobacteriota sp. (MAG13) and Acanthopleuribacteraceae sp. (MAG20). By generating two long-read metagenomes (Methods), we were able to reconstruct high-quality genomes, increasing the predicted number of complete BGCs by 15–83% (Fig. 4b).

Highlighting Acidobacteriota spp. as particularly BGC-rich (Fig. 4a,b) aligns with the growing recognition of this phylum as containing candidate natural product super-producers[22,23,26]. Building on this, we assessed the prevalence and abundance of Acidobacteriota spp. across reef-building corals using the *Tara* Pacific 16S rRNA gene amplicon dataset[9] that matched the *Porites*, *Pocillopora* or *Millepora* metagenomes studied here (812 samples; Methods). Acidobacteriota lineages, including those from which we recovered BGC-rich genomes, were abundant and prevalent across the three coral genera (Fig. 4c), potentially influencing coral hosts, as indicated by their impact on *Pocillopora* gene expression (using a generalized dissimilarity model; Supplementary Information; Extended Data Fig. 3b,c and Supplementary Table 8), enrichment of ELPs (Supplementary Information and Extended Data Fig. 2b) and transcriptional and metabolic responsiveness to coral-derived compounds (demonstrated through a co-cultivation experiment, RNA-sequencing and high-performance liquid chromatography (HPLC) coupled to heated electrospray ionization high-resolution tandem mass spectrometry (MS/MS) (HPLC–MS/MS); Supplementary Information and Extended Data Fig. 4).

## New Acidobacteriota natural product enzymology

Our findings demonstrate that reef-building corals are enriched in host-specific (Fig. 2) and biosynthetically rich bacterial lineages, including Acidobacteriota spp. (Fig. 4). Although the nature and specificity of their relationship with corals remain to be further explored to validate true symbiotic interactions, several uncultivated clades of coral-associated Acidobacteriota emerge as new biotechnological targets[26] given their untapped yet promising potential to yield new natural products. To move beyond computational predictions of BGCs[22,23,43] and overcome the limitation of accessing only the biosynthetic potential of cultivated microorganisms[26], we targeted select RiPP BGCs from reconstructed genomes for functional characterization through synthetic biology. First, we investigated a BGC encoded by a member (MAG13; Fig. 4) of an unclassified acidobacterial family (*Candidatus* UBA6911 class, *Ca.* RPQK01 order), which we named the *aci* cluster (Fig. 5a). We selected this class III lanthipeptide cluster, which contains genes encoding precursors for the putative RiPP natural products and a type III lanthionine synthetase homologue[44], as it encodes unusual, glycine-rich C-terminal precursor regions (Extended Data Fig. 5). We combined heterologous expression in *Escherichia coli* with HPLC–MS/MS to gain biochemical insights into the modified *aci* precursors (Supplementary Information; Methods). We found seven dehydrations (Fig. 5a) and (methyl-)lanthionine or labionin macrocyclization along a serine–threonine motif ($S/TX_2$–$(SX_2)_5$) in both precursors (Extended Data Fig. 5), similar to known class III and IV lanthipeptides. Differences

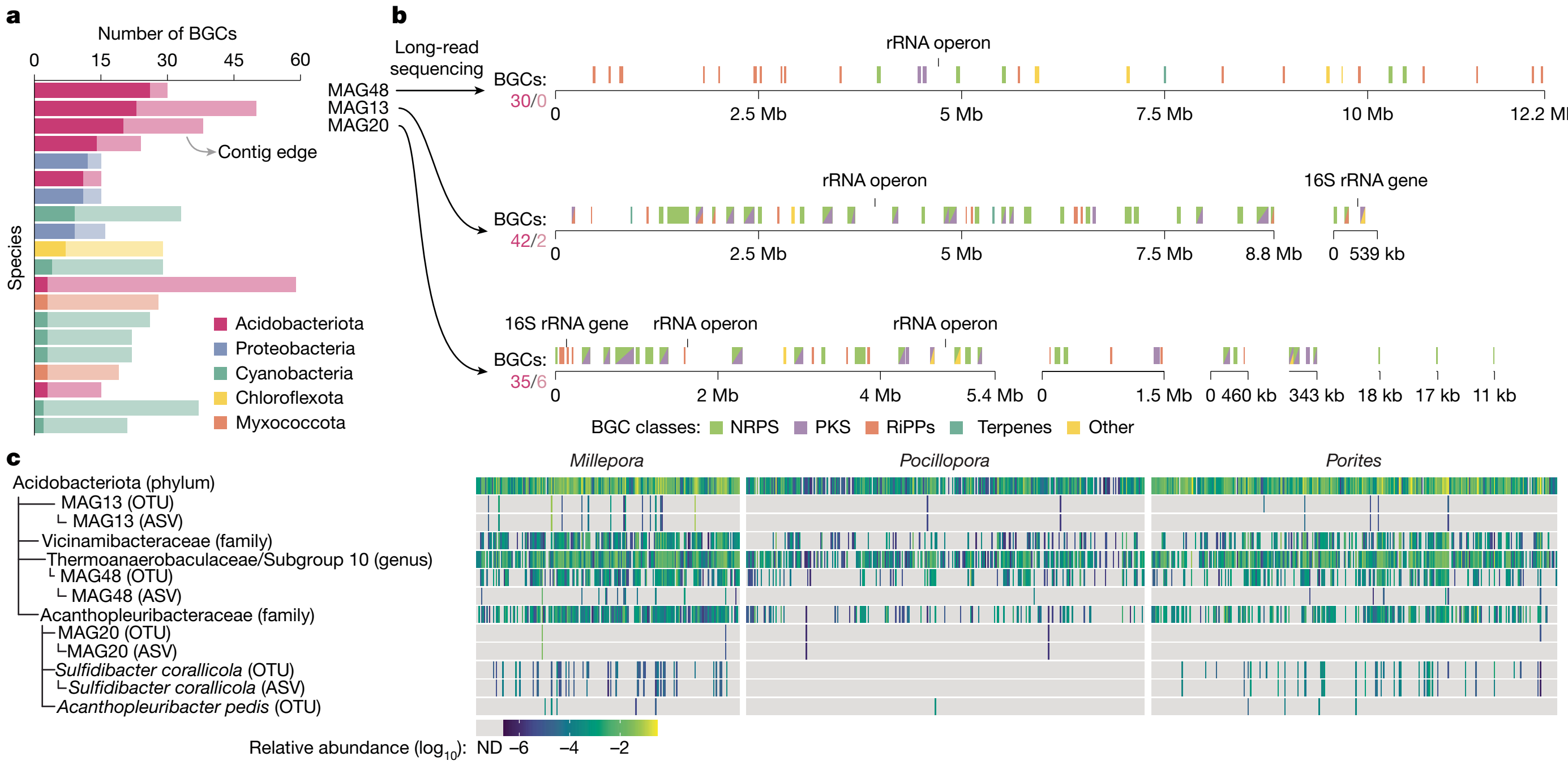


**Fig. 4 | Acidobacteriota spp. as BGC-rich lineages. a**, Screening the RMD, we identified 20 candidate natural product super-producer species (that is, BGC-rich lineages) from the five phyla Acidobacteriota, Proteobacteria, Cyanobacteria, Chloroflexota and Myxococcota. None of these species had representatives in the GTDB (r207). Lighter shades indicate BGCs located at the edge of a contig (Methods). Species are ordered by the number of predicted complete BGCs. **b**, By generating long-read metagenomes (Methods), we reconstructed complete or near-complete MAGs for the three species with the highest number of complete BGCs in the short-read dataset. For each MAG, the number of BGCs (complete/contig-edge) is shown. **c**, We explored the abundance of Acidobacteriota spp. overall, as well as specific BGC-rich taxonomic groups within this phylum (that is, the Vicinamibacteraceae and Acanthopleuribacteraceae families, and the Thermoanaerobaculaceae/Subgroup 10 genus) across the *Tara* Pacific 16S rRNA gene amplicon dataset (812 matching samples) based on taxonomic annotations (Methods). We detected perfect matches (ASVs) and/or close relatives (OTUs) for the two isolated strains *Acanthopleuribacter pedis* and *Sulfidibacter corallicola* as well as for two of the three species shown in **b** (MAG20 and MAG48). For the third species (MAG13), we identified an unambiguous match based on abundances (Methods).

in the spacing of cysteine residues could affect enzyme recognition or peptide bioactivity, opening future opportunities for biotechnological explorations.

Second, we focused on a BGC encoded by a member (MAG48; Fig. 4) of an unclassified genus of the *Ca*. UBA5704 family (*Ca*. UBA5704 order, Thermoanaerobaculia), which we termed the *tha* cluster. This cluster encodes a putative lanthipeptide precursor protein with an N-terminal leader sequence and a C-terminal cysteine-rich core (ThaA), a type II lanthionine synthetase homologue (ThaM)[45] and a glucose–methanol–choline (GMC)-family oxidoreductase (ThaO) (Fig. 5b). The latter, which is a flavin adenine dinucleotide-dependent enzyme family, is functionally diverse, known to catalyse a wide range of reactions and includes examples such as *Drosophila melanogaster* glucose dehydrogenase, *Ogataea polymorpha* methanol oxidase and *E. coli* choline dehydrogenase[46,47]. As none of these enzymes have been described in RiPP biosynthesis, we reasoned that ThaO may represent new enzymology. Again combining heterologous expression in *E. coli* with HPLC–MS/MS analysis (Supplementary Information; Methods), we detected two dehydrations in the precursor protein ThaA catalysed by ThaM and the formation of two lanthionine-based macrocycles with D,L-stereochemistry (Fig. 5b and Extended Data Fig. 6). Through stable-isotope labelling, introduction of point mutations and MS/MS, we uncovered thiazole formation in the glycine–cysteine motif of ThaA when co-produced with ThaM and ThaO, which was corroborated by nuclear magnetic resonance (NMR) data (Fig. 5b, Supplementary Information and Extended Data Figs. 7 and 8). As thiazoles are crucial components in many pharmaceutical and bioactive compounds[48], we tested the modified ThaA core for bioactivity against the proteases cathepsin B, trypsin, chymotrypsin and human neutrophil elastase. For the latter, we detected inhibition at low micromolar concentrations (half maximal inhibitory concentration ($IC_{50}$) = 2.6 µM) (Fig. 5c). Although RiPP thiazoles usually form through adenosine triphosphate (ATP)-dependent phosphorylation catalysed by YcaO-type enzymes[49], here we report the production of thiazole by a GMC oxidase member that lacks homology to ATP-binding enzymes. This protein could be of interest for biocatalytic applications.

Prompted by these prospects, we further explored the diversity of RiPP-associated GMC-family oxidoreductases beyond coral reefs with the aim of finding a lanthionine synthetase-independent homologue. We identified a promising candidate (Methods) in an unclassified Thermoanaerobaculia species recovered from a soil metagenomic sample (MAG, GenBank accession: GCA_035278895.1), which we termed the *the* cluster. Heterologous expression of this cluster and HPLC–MS/MS analysis revealed that indeed, TheO efficiently modified TheA without requiring co-production of the associated lanthionine synthetase TheM. The resulting mass shift (−20 Da) in TheA near the GGC motif is consistent with thiazole formation (Fig. 5d and Extended Data Fig. 9).

To further explore this cluster's biotechnological potential, we modified, shortened and cargo-fused TheA. First, we introduced glycine-to-alanine point mutations and found that a substitution of either glycine adjacent to the cysteine in TheA impaired enzyme activity, whereas the glycine preceding the thiazole could be replaced by alanine without loss of activity (Extended Data Fig. 9). Second, TheA variants with truncated cores of 20, 14 and eight amino acids (but not a two-amino-acid variant) were thiazole-modified when co-produced with TheO (Fig. 5d and Extended Data Fig. 9), indicating that much of the peptide is dispensable for recognition by TheO. Third, encouraged by this relaxed substrate specificity, we explored enzymatic thiazole installation N-terminal to proteins of interest. Because thiazoles restrict peptide backbone conformation and can enhance activity, stability and

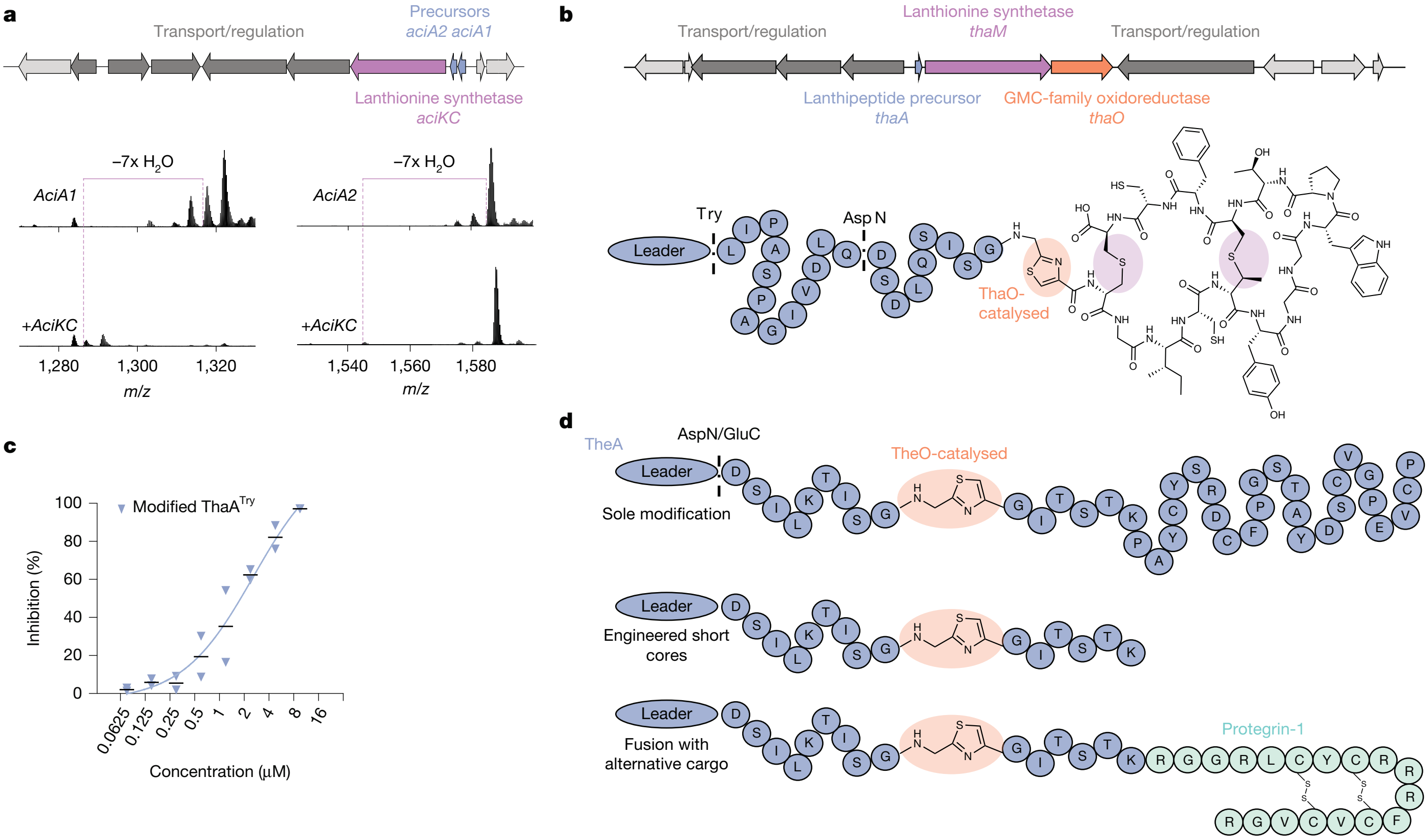


**Fig. 5 | The coral microbiome is a source of new enzymes and natural products.** **a**, The antiSMASH-predicted *aci* cluster encodes two precursor proteins (blue, AciA1 and AciA2) and a type III lanthionine synthetase (purple, AciKC), in addition to predicted proteins involved in transport and regulation (dark grey). The maturase installs seven dehydrations in specific regions of each precursor (Supplementary Table 9). **b**, The antiSMASH-predicted *tha* cluster encodes one precursor protein (blue, ThaA), a type II lanthionine synthetase (purple, ThaM) and a GMC-family oxidoreductase (orange, ThaO), in addition to predicted proteins involved in transport and regulation (dark grey). We resolved structurally the enzymatically modified C-terminal region (released by trypsin and AspN proteolysis) of ThaA when co-produced with ThaM and ThaO (Supplementary Information and Supplementary Table 10; Methods). **c**, The tryptic fragment of fully modified ThaA inhibits human neutrophil elastase in vitro ($n = 2$ technical replicates, $IC_{50} = 2.6$ μM). Black lines represent the mean. **d**, The homologous *the* cluster (Supplementary Information) encodes the GMC-family oxidoreductase TheO, which installs thiazoles in its associated precursor without requiring a lanthionine synthetase, modifies truncated core sequences and installs thiazoles in sequences fused to the antimicrobial peptide Protegrin-1 (Supplementary Table 11).

other physicochemical properties of peptide-mimetic therapeutics[48], we fused porcine Protegrin-1—an 18-residue β-sheet antimicrobial peptide[50]—to the C-terminus of the smallest modifiable core peptide. We detected thiazole formation in the fusion protein, demonstrating that TheO can install thiazoles on engineered hybrid peptide substrates. Future work will explore truncating the N-terminal sequence and testing different cargo proteins.

## Conclusions

The systematic, basin-scale sampling effort of the *Tara* Pacific expedition facilitated access to a wealth of host-specific, previously unavailable genome-resolved information for coral-associated microbial species. Comparative analyses, enabled by establishing a reef microbiome-wide database for key coral reef organisms, revealed stony and fire corals as particularly rich in new, microbially encoded BGCs as well as BGC-rich microbial lineages. On the one hand, this enormous, largely untapped biosynthetic potential of reef-building corals, which have so far contributed only a minimal portion of the diversity of marine natural products[2,39], represents a promising outlook for sustainable, efficient and non-redundant bioprospecting[25]. As such, we characterized experimentally the first RiPP modifications from Acidobacteriota, described a new RiPP maturase enzyme family and, through the identification of a suitable homologue, demonstrated its promising potential for biocatalytic applications. This outlook is particularly striking considering that our work focused on only a small fraction of the globally described number of extant reef-building coral species (current estimates: over 750)[51] and their associated microbial diversity[9], the vast majority of which is yet undescribed genomically. On the other hand, our findings highlight what is at stake under the continued, anthropogenic pressure-driven loss of host-level biodiversity, as exemplified in our study of coral reefs, and the unique microbial diversity it supports and is supported by ref. 52.

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

**Tara Pacific Consortium Coordinators**

**Emilie Boissin[13], Serge Planes[13,15], Denis Allemand[17,18], Sylvain Agostini[19], Chris Bowler[20], Eric Douville[21], Didier Forcioli[17,22], Pierre E. Galand[23], Fabien Lombard[15,24], Olivier P. Thomas[10], Rebecca Vega Thurber[6,25], Romain Troublé[12,15], Christian R. Voolstra[9], Patrick Wincker[8,15], Maren Ziegler[26], Shinichi Sunagawa[1], Colomban de Vargas[16], J. Michel Flores[27], Paola Furla[17,22], Eric Gilson[17,22,28], Stéphane Pesant[29], Stephanie Reynaud[17] & Didier Zoccola[17]**

[27]Department of Earth and Planetary Sciences, Weizmann Institute of Science, Rehovot, Israel. [28]Department of Medical Genetics, CHU Nice, Nice, France. [29]European Molecular Biology Laboratory, European Bioinformatics Institute, Wellcome Genome Campus, Hinxton, UK.

## Methods

### Generating data from the *Tara* Pacific expedition

**Metagenomic data generation from *Tara* Pacific samples.** The *Tara* Pacific expedition collected coral and seawater samples from 99 reefs across 32 islands throughout the Pacific Ocean[3,53]. The detailed sampling protocols are presented in ref. 28. Based on morphology, we targeted three reef-building corals: the stony corals *Pocillopora meandrina* and *Porites lobata* and the fire coral *Millepora* cf. *platyphylla*. However, because corals within the same genus can be difficult to differentiate by eye, we here aggregated the results at the genus level (*Pocillopora*, *Porites* and *Millepora*) and performed population genomic analyses of the sampled morphotypes to confirm their identities in a subset of 11 islands[27]. The design and use of diagnostic genomic fragments[54] will allow future studies to identify coral species in near real time. In addition, two samples of *Heliopora* were collected at one site.

Samples from three colonies per species were collected for metagenomic analysis at each site using hammer and chisel. The fragments were stored underwater in individual Ziploc bags and preserved in tubes with DNA/RNA Shield (Zymo Research) at −20 °C once on board the schooner *Tara*. Furthermore, plankton was collected on 0.2–3-µm filters by filtering 100 l of open ocean water with a pump offshore, 100 l of reef water with a pumping system and tubing within the reefs and 50 l of coral-surrounding water with a pumping system and diver-held tubing from two *Pocillopora* colonies per reef. All filters were preserved in cryovials in liquid nitrogen. The sample provenance and environmental context are available at Zenodo (https://doi.org/10.5281/ZENODO.4068292)[55].

DNA was extracted with commercial kits after mechanical cell disruption, and metagenomic sequencing was performed on a NovaSeq6000 or HiSeq4000 system (Illumina) to produce 100 million of 150-bp paired-end reads per sample. The detailed protocols are provided in ref. 56.

**TEM images of coral tissue samples.** The *Tara* Pacific expedition also enabled the microscopical inspection of coral tissue. For this, 0.1 g of coral was separated using bone cutters and preserved immediately on board in electron-microscopy-grade glutaraldehyde. In the laboratory, the coral fragments were decalcified in 500 µl of 10% EDTA (pH 7). Once decalcified, the samples were rinsed with 0.1 M sodium cacodylate buffer and embedded in agarose for post-fixation staining with 1.5% potassium ferrocyanide and 2% osmium tetroxide in water. Osmium tetroxide staining was followed by uranyl acetate and lead nitrate staining[57]. Next, the samples were dehydrated in serial acetone baths (10%, 30%, 50%, 70%, 90%, 95% and 100%) for 10–15 min each before being infiltrated with araldite resin and sectioned using a ultramicrotome (RMC). Samples were imaged on a FEI Helios Nanolab 650 microscope in scanning TEM mode at the Oregon State University electron microscopy facility.

### Building the RMD

**Inclusion of publicly available sequence data from other corals and sponge microbiomes.** We complemented the newly generated *Tara* Pacific metagenomic dataset with publicly available datasets. Specifically, we searched the European Nucleotide Archive (ENA) biosamples database (December 2022) to identify metagenomic datasets from coral or sponge samples. Upon literature review, we included metagenomic data from 15 coral[58–72] and 16 sponge[23,73–87] focused studies (Supplementary Table 2). In total, these datasets included 412 coral (from 29 genera, including 22 stony and five soft coral genera, as well as black and fire corals) and 371 sponge samples of global distribution (Extended Data Fig. 1a). An additional 103 microbial genomes from cultivated coral-associated strains, reported across 26 studies[18,88–108], were included (Supplementary Table 4).

**Metagenomic data assembly and binning.** Metagenomic data were processed as described in ref. 24. Briefly, sequencing reads from all metagenomes were quality filtered using BBMap (v.38.79) by removing sequencing adaptors from the reads, removing reads that mapped to quality control sequences (PhiX genome) and discarding low quality reads using the parameters *trimq = 14*, *maq = 20*, *maxns = 1* and *minlength = 45*. Downstream analyses were performed using quality-controlled reads or, if specified, merged quality-controlled reads (bbmerge.sh *minoverlap = 16*). For host-associated metagenomes (that is, excluding seawater samples), quality-controlled reads from *Tara* Pacific metagenomes were normalized (bbnorm.sh *target = 40*, *mindepth = 0*) and all metagenomes were assembled individually with metaSPAdes (v.3.14.1 or v.3.15 if required)[109]. The resulting scaffolded contigs (hereafter scaffolds) were filtered by length (at least 1 kbp).

The 1,603 metagenomic samples were grouped into several sets and, for each sample set, the quality-controlled metagenomic reads from all samples were mapped individually against the scaffolds of each sample. Publicly available metagenomes were processed by study, while the *Tara* Pacific coral metagenomes were processed by host (creating several sets of approximately 100 samples within each host genus). Reads were mapped with BWA (v.0.7.17-r1188)[110], allowing the reads to map at secondary sites (with the *-a* flag). Alignments were filtered to be at least 45 bases in length, with an identity of at least 97% and covering at least 80% of the read sequence. The resulting BAM files were processed using the jgi_summarize_bam_contig_depths script of MetaBAT 2 (v.2.12.1)[111] to provide within- and between-sample coverages for each scaffold. The scaffolds were finally binned by running MetaBAT 2 on all samples individually with parameters *--minContig 2000* and *--maxEdges 500* for increased sensitivity.

**Quality evaluation of metagenomic bins and additional genomes.** The quality of each metagenomic bin and external genome was evaluated using both the 'lineage workflow' of CheckM (v.1.1.3)[112] and Anvi'o (v.7.1)[113]. Metagenomic bins and external genomes were retained for downstream analyses if either CheckM or Anvi'o reported a completeness/completion (cpl) of at least 50% and a contamination/redundancy (ctn) of less than or equal to 10%. Completeness and contamination were averaged across both tools and the selected genomes were further attributed a quality score (Q) as follows: $Q = cpl - 5 \times ctn$. Based on the quality score, the bins were classified as MAGs of different quality levels according to community standards[114] as follows: high quality, $cpl \geq 90\%$ and $ctn \leq 5\%$; good quality, $cpl \geq 70\%$ and $ctn \leq 10\%$; medium quality, $cpl \geq 50\%$ and $ctn \leq 10\%$; fair quality, $cpl \leq 50\%$ or $ctn \geq 10\%$ (edge case due to the average between the CheckM and Anvi'o).

**Species-level clustering of the genome collection.** The genomes were grouped subsequently into species-level clusters on the basis of whole-genome average nucleotide identity (ANI) using dRep (v.3.0.0)[115] with a 95% ANI threshold[29,116] (*-comp 0 -con 1000 -sa 0.95 -nc 0.2*). Whenever mentioned, a representative genome was selected based on the maximum quality score defined by dRep ($Q' = cpl - 5 \times ctn + ctn \times (\text{strain heterogeneity})/100 + 0.5 \times \log[N50]$) for each species-level cluster.

**Genome size estimates.** We estimated the genome size of microbial species using a method robust to potential MAG incompleteness or contamination[24]. Briefly, we retained only genomes of at least 70% completeness (which represents more than 80% of all species in the RMD), and computed an average estimated genome size per species accounting for incompleteness by multiplying actual genome sizes by 100/completeness.

**Taxonomic and functional genome annotation.** Prokaryotic genomes were annotated taxonomically using GTDB-Tk (v.2.1.0)[117] with the default parameters and the fulltree option using the GTDB r207

release[118]. The taxonomic annotation of a species is defined as the one of its representative genome.

Each genome was annotated functionally by first predicting complete genes using Prodigal (v.2.6.3)[119], Barrnap (v.0.9) and Aragorn (v.1.2.41)[120], with the taxonomy parameter specified based on GTDB annotations, where appropriate. The predicted proteins were then used to identify universal single-copy marker genes with fetchMGs (v.1.2)[121]. The gene sequences were additionally used as input to identify BGCs in the genomes using antiSMASH (v.6.1.1)[20] and a minimum contig length of 5 kbp.

### Assessing host-specificity within the coral reef microbiome

**Species-level profiling with mOTUs.** Genomes were added to the database (v.3.1) of the metagenomic profiling tool mOTUs[121] to generate an extended mOTUs reference database. Only genomes with at least six out of the ten universal single-copy marker genes were kept. The resulting 10,511 genomes grouped into 3,673 mOTUs species-level clusters, 3,588 (98%) of which represented new clusters in the extended database. The difference in the number of species-level clusters reported by mOTUs (3,673) and dRep (4,224) can be explained by more stringent filtering of the genomes (10,511 instead of 13,549) and a different clustering strategy. Yet, the two independent methods yielded very similar clustering patterns with a V-measure of 0.99, a homogeneity of 0.99 and a completeness of 0.98. Profiling of the 2,074 metagenomes was performed using the default parameters of mOTUs (v.3.1).

To assess the distribution (detection, abundance) of microbial species across host and seawater samples, the resulting profiles were filtered to retain the 3,673 mOTUs for which genomes were available. We subsequently removed mOTUs that were detected in fewer than ten samples and samples that had a total scaled abundance of three or less. The resulting matrix was used to compute Jaccard distances between samples. These were visualized by a Principal Coordinate Analysis (PCoA) (package ape (v.5.7-1)[122] in R) and the statistical significance was tested using PERMANOVA. In case of unbalanced groups (for instance, 337 sponge metagenomes versus 47 sponge-associated seawater metagenomes), a random subsample (here 50) of the larger group was selected to perform the PERMANOVA and the statistics were derived from 100 repetitions.

### Assessing the functional and biosynthetic potential within the coral reef microbiome

**Functional profiling.** Similar to the methods described previously[24,32,123], we clustered the more than 37 million protein-coding genes at 95% identity and 90% coverage of the shorter gene using CD-HIT (v.4.8.1)[124] into 16.3 million gene clusters. The longest sequence was selected as the representative gene of each gene cluster. The representative genes were annotated by assigning them to orthologous groups with emapper (v.2.1.7)[125] based on eggNOG (v.5.0.2)[126] and by performing queries against the KEGG database (release v.2022-04)[127]. This last step was performed by aligning the corresponding protein sequences using DIAMOND (v.2.0.15.153)[128] with a query and subject sequence coverage of at least 70%. The results were further filtered on the basis of the bitscore being at least 50% of the maximum expected bitscore (query sequence aligned against itself) in accordance with the thresholds implemented in the NCBI Prokaryotic Genome Annotation Pipeline[129].

The 1,603 host-associated and 471 seawater metagenomes were then mapped to the cluster representatives with BWA (v.0.7.17-r1188) (*-a*) and the resulting BAM files were filtered to retain only alignments with a percentage identity of at least 95%, an alignment length of at least 80% of the query length and at least 45 bases. Gene abundance was calculated by first counting inserts from best unique alignments and then, for ambiguously mapped inserts, adding fractional counts to the respective target genes in proportion to their unique insert abundances. These abundance profiles were subsequently length-normalized based on the length of the gene representatives and sequencing depth-normalized using the total mOTUs counts from the corresponding sample.

**Screening for ELPs.** To predict ELPs across the RMD and the OMD, we used a list of ELP-associated PFAMs defined previously[18]. We processed the eggNOG annotations of the gene catalogues to identify predicted ELPs across the ocean and coral microbiomes. The results were then summarized per genome and normalized by the total number of predicted proteins per genome.

**Clustering of BGCs into GCFs.** All BGCs predicted across all genomes, along with those from the OMD[24] and MIBiG[130] (for contextualization), were grouped into biosynthetic GCFs using clust-o-matic with a 0.5 threshold[131] based on the comparison of biosynthetic genes. This sequence identity-based clustering was previously shown to perform similarly to BiG-SLiCE[37], yet to provide an alternative that is less sensitive to BGC fragmentation. Each GCF was attributed a natural product class on the basis of antiSMASH annotations of the individual BGCs, as defined in BiG-SCAPE[132]. The fraction of new GCFs was estimated based on the distances to databases of computationally predicted (the RefSeq database within BiG-FAM)[133] and experimentally validated (MIBiG v.2.0)[130] BGCs.

**GCF normalization.** We normalized the number of GCFs by sampling effort, taking into account several parameters: first, we accounted for sequencing depth by normalizing the number of GCFs detected in a given sample by the total mOTUs count—a proxy for the number of bacterial and archeal cells sequenced in a metagenome. Second, we tested the impact of host diversity by building rarefaction curves (100 iterations) of new GCFs recovered based on the number of host genera sampled across host types (sponges, stony and soft corals; fire corals were excluded as they represent a single genus). Third, we tested the impact of geographic coverage based on the *Tara* Pacific data. We built rarefaction curves (500 iterations) of new GCFs recovered based on the number of islands sampled across coral genera (*Millepora*, *Porites* and *Pocillopora*).

### Investigating Acidobacteriota spp. as BGC-rich members of the reef-building coral microbiome

**Defining candidate BGC-rich species.** To define candidate BGC-rich species, we selected a representative genome per species based on the number of complete BGCs, the quality Q-score (based on completeness − 5 × contamination) and the N50 (the shortest contig length that needs to be included to cover 50% of the genome). The number of complete BGCs and the N50 were normalized per species using $y = \log(x)/\max(\log[x]) \times 100$. The three indices (each ranging from 0 to 100) were summed to form a composite index. If the highest scoring genome encoded at least 15 BGCs, we considered that species a candidate BGC-rich species. For comparative purposes, we used the same definition of BGC-rich species on the OMD MAGs.

**Validation of BGC-rich acidobacterial MAGs by long-read metagenomics.** A HiFi library was prepared from leftover DNA from *Tara* Pacific samples SAMEA6034818 and SAMEA6035815 using the SMRTbell Express Template Prep kit v.2.0 (PacBio) for ultra low input PacBio sequencing. A 5-kbp size-selection was performed on the library using Ampure PacBio beads diluted with PacBio elution buffer to 35% (v/v), following manufacturer's instructions. Three runs were sequenced to generate a total of 151.6 Gbp of deduplicated PacBio HiFi reads for SAMEA6034818 and one run was sequenced to generate a total of 59.2 Gbp of deduplicated PacBio HiFi reads for SAMEA6035815. Using blastn (v.2.15.0+), we aligned the reads to the corresponding BGC-rich acidobacterial MAGs reconstructed from the short-read metagenome of the same sample (that is, TARA_SAMEA6034818_MAG_00000048, TARA_SAMEA6034818_MAG_00000020 or

TARA_SAMEA6035815_MAG_00000013). Retaining reads with a minimum length of 1 kbp that aligned to the reference with at least 95% identity across at least 200 bp, we assembled the reads with Flye (v.2.9.3)[134] (with the options --*pacbio-hifi* and --*scaffold*) to generate new MAGs and repeated the read mapping and assembly step to generate two MAGs of one (MAG48; GCA_977880245), two (MAG13; GCA_977880235) and seven (MAG20; GCA_977880255) contigs, respectively. We annotated the long-read MAGs with Barrnap (v.0.9) and antiSMASH (v.6.1.1)[20] as described for the short-read MAGs.

**Tara Pacific 16S rRNA gene amplicon dataset.** To further explore the distribution of selected microbial lineages across the coral hosts sampled by *Tara* Pacific, we used the *Tara* Pacific 16S rRNA gene (16S) amplicon dataset (v.1.1)[9] available at *Zenodo* (https://doi.org/10.5281/ZENODO.4073268)[135] and selected the 812 samples that matched the 820 coral metagenomes. Candidate BGC-rich MAGs reconstructed in this study were matched to 16S amplicon sequence variants (ASVs) by (1) recovering, whenever available, the genomic 16S sequences directly from the genomes (MAG48 and MAG20 as well as the Acanthopleuribacteraceae spp. isolates) and aligning them to the *Tara* Pacific ASVs (using blastn v.2.15.0+), (2) identifying the 16S sequences from genus-level relatives in the GTDB and aligning those to the 16S-based operational taxonomic units (OTUs) or (3) identifying unambiguous genome to ASV matches based on mOTUs and 16S abundances (MAG13). ASV-level matches are reported for alignments of 100% identity over the length of the ASV and OTU-level matches are reported for alignments of 97% identity over the length of the ASV.

**Characterizing natural products encoded by Acidobacteriota spp.**

**Structural predictions and plasmid construction.** We selected BGCs TARA_SAMEA6023455_MAG_00000007-scaffold_11-biosynth_1 (*Candidatus* UBA6911 order, termed the *aci* cluster, distance to RefSeq: 698), TARA_SAMEA6034818_MAG_00000048-scaffold_6-biosynth_2 (*Ca.* UBA5704 order, termed the *tha* cluster, distance to RefSeq: 1104), and contig DATEIO010000468.1 (MAG, GenBank accession: GCA_035278895.1, termed the *the* cluster). Protein and protein–complex structures were predicted using AlphaFold (v.2.2.0)[136]. We ordered the cloning plasmids (Twist Bioscience) encoding precursor peptides and enzymes either as codon-optimized gene fragments (for *E. coli*) and subcloned into pACYC expression vectors using *Nco*I and *Hin*dIII restriction sites, or directly as pET-24(+) expression vectors (Supplementary Tables 9–11). The plasmid encoding $His_6$-bdSUMO-ThaA-S2A-T6A was constructed using mutagenic primers (Supplementary Table 10), Phusion Polymerase (NEB), and the Kinase-Ligase-*Dpn*I reaction mix (NEB), following the manufacturer's instructions. Plasmids were purified using the NucleoSpin Plasmid Purification kit (Macherey–Nagel) and transformed into chemically competent *E. coli* DH5α (Invitrogen, for plasmid assembly) or *E. coli* BL21(DE3) (Invitrogen, for protein expression) strains. Transformation involved incubating 100 µl of cells with plasmid DNA on ice for 15 min, followed by a 30-s heat shock at 42 °C, and recovery at 37 °C for 60–90 min in 900 µl of lysogeny broth. We plated transformants on lysogeny broth agar with appropriate antibiotics. Plasmids were either isolated (NucleoSpin Gel and PCR Clean-up kit; Macherey–Nagel) or directly picked from colonies and sequenced in the region of interest by Microsynth. The resulting plasmids, regulated by isopropyl-β-D-1-thiogalactopyranoside, encoded resistance to chloramphenicol (precursors) or kanamycin (enzymes).

**Protein expression and purification.** Plasmids or combinations of plasmids were transformed into chemically competent *E. coli* BL21(DE3) as described above. We inoculated 5 ml of lysogeny broth containing appropriate antibiotics (chloramphenicol or kanamycin; Applichem) with transformants from agar plates or glycerol stocks. After overnight incubation at 37 °C with shaking (180 rpm), the cultures were used to inoculate 30–1,000 ml of terrific broth with antibiotics (1% v/v). Cultures were shaken (180 rpm) at 37 °C until $OD_{600}$ reached 0.8–1.4, then cooled to 4 °C for 30 min. Isopropyl-β-D-1-thiogalactopyranoside was added to a final concentration of 1 mM. Cultures were incubated at 16–24 °C with shaking (180 rpm) for 16–64 h. Cells were collected by centrifugation (6,000 × *g*, 4 °C, 15 min) and the supernatant discarded. The cell pellets were resuspended in 1–50 ml of NPI-10 or NPI-20 buffer (scaled to expression volume; NPI buffers (pH 8.0) contain 50 mM phosphate, 300 mM NaCl, 10% glycerol and 10–250 mM imidazole, where NPI-X corresponds to X mM imidazole). Cells were lysed by sonication (Qsonica Q700 Sonicator) with a tip radius of 2 mm (1-ml suspension), 6 mm (4- to 8-ml suspension) or 12 mm (more than 20-ml suspension), applying six to ten 10-s cycles with a 10-s pause at 25–60% amplitude. Lysates were centrifuged (18,000–21,000 × *g*, 4 °C, 45 min) and the supernatant was incubated with 0.1–5 ml of Protino Ni-NTA agarose (Macherey–Nagel) for at least 30 min at 4 °C with shaking. Samples were loaded into a pre-washed (NPI-10 buffer) polypropylene column and washed with NPI buffers of increasing imidazole concentration (once with NPI-10, twice with NPI-20, once with NPI-40), followed by elution with 0.5–20 ml of NPI-250 buffer. Elution fractions were concentrated using Amicon Ultra-15 or Amicon Ultra-4 filter units (Merck) according to the manufacturer's instructions, with buffer exchange (when appropriate) using PD-10 G-25 or PD MidiTrap G-25 columns (Cytiva). Samples were stored at −20 °C.

**Proteolytic digests and HPLC–MS/MS analysis.** Proteolytic digests were performed using appropriate endo- or exo-proteases at recommended protease concentrations and buffers and supplemented with 10 mM Tris(2-carboxyethyl)phosphine (Bond-Breaker TCEP Solution; Thermo Fisher Scientific). Tryptic (Promega) digests were incubated overnight at room temperature or 37 °C, endoproteinases AspN and GluC (NEB) digests overnight at 37 °C. Samples were diluted with acetonitrile plus 0.1% formic acid and centrifuged before HPLC–MS/MS (Q Exactive Hybrid Quadrupole-Orbitrap Mass Spectrometer; Thermo Fisher Scientific) analysis. The gradient method we used was 95:5 A:B for 30 s ramped up to 5:95 A:B over 20 min (solvent A, water plus 0.1% formic acid; solvent B, acetonitrile plus 0.1% formic acid; column, Phenomenex Kinetex 2.6 µm C18-XB 100 Å (150 × 4.6 mm); flow rate, 0.7 ml $min^{-1}$). For MS/MS, we used a normalized collision energy of 24 (depending on the observed fragmentation properties of peptide fragments). The MS was operated in positive ionization mode at a scan range of 400–2,000 *m/z*, automatic gain control target $2 \times 10^5$, maximum injection time 100 ms and a resolution of 70,000 at 400 *m/z*. The spray voltage was set to 5.0 kV, probe heater temperature to 475 °C and the capillary temperature to 270 °C. Columns were heated to 50 °C. Data were analysed using Thermo Xcalibur Qual Browser v.4.1 (Thermo Fisher Scientific).

**HPLC purification.** Large-scale digests were subjected to solid-phase extraction (SPE) before HPLC. SPE columns (5 g; Phenomenex Strata C18-E) were activated with acetonitrile plus 0.1% formic acid and equilibrated with water plus 0.1% formic acid. Samples were loaded, the flowthrough collected and again loaded up to three times, and eluted using increasing acetonitrile concentrations in water plus 0.1% formic acid (0%, 10%, 30%, 50% and 80%). Fractions containing the target compound were identified by HPLC–MS/MS, dried by lyophilization and resuspended in 30% acetonitrile in water plus 0.1% formic acid for reversed-phase HPLC purification. The gradient method used was 80:25 A:B for 30 s ramped up to 5:95 A:B over 16 min (solvent A, water plus 0.1% formic acid; solvent B, acetonitrile plus 0.1% formic acid; column, Phenomenex Luna 2.6 µm C18 100 Å (150 × 4.6 mm); flow rate: 2.0 ml $min^{-1}$). Fractions containing the target compound were identified by HPLC–MS/MS, dried by lyophilization and stored at −20 °C.

**Iodoacetamide and Raney nickel derivatization.** Derivatization was performed on purified and digested peptides (nickel-affinity purification) after SPE (eluted in 30–60% acetonitrile plus 0.1% formic acid). For iodoacetamide (IAA) derivatization (following methods described

previously[137,138]), peptides were dried by evaporation, resuspended in 25 µl of IAA buffer (50 mM Tris-HCl, 7.5 M urea, 5 mM DTT, pH 8.0) and incubated for 1 h at 37 °C with shaking. Then, 4.4 µl of 0.1 M IAA was added, followed by 30 min incubation in the dark at room temperature. Reactions were quenched with 1.5 µl of 0.1 M dithiothreitol, diluted in 120 µl of water plus 0.1% formic acid and analysed by HPLC–MS/MS. For Raney nickel derivatization (derivatization of non-modified cysteine residues without linearization; following methods described previously[139]), dried samples were resuspended in 300 µl of buffer (8 M guanidine hydrochloride, 20 mM EDTA, 20 mM Tris-HCl, pH 8.0) and shaken at 37 °C for 1 h. Raney nickel (25 mg, active slurry) was added and samples were incubated at 55 °C for 16 h. Reactions were centrifuged, removing and desalting the supernatant using $C_{18}$ ZipTips (Merck) according to the manufacturer's instructions and analysed by HPLC–MS/MS.

**Marfey's derivatization for stereochemistry determination of (Me) Lan bridges.** Peptides were hydrolysed in 6 M HCl at 110 °C for 18 h, dried by heat evaporation and resuspended in 75 µl of water, 25 µl of 1 M $NaHCO_3$ and 125 µl of *N*-α-(2,4-dinitro-5-fluorophenyl)-L-valinamide (4 mg ml$^{-1}$ in acetone) as described previously[138,140]. Samples were heated at 45 °C for 1 h, reactions quenched with 25 µl of 1 M HCl and diluted with water plus 0.1% formic acid (1/1 v/v), and the hydrolysed peptides analysed by HPLC–high-resolution mass spectrometry. Stereochemistry was determined by comparing elution times with the characterized standard nisin A (Sigma Aldrich). The gradient method used was 70:30 A:B for 30 s ramped up to 10:90 A:B over 12 min (solvent A, water plus 0.1% formic acid; solvent B, acetonitrile plus 0.1% formic acid). Full mass spectra (heated electrospray ionization in positive ion mode; spray voltage, 3,500 V; capillary temperature, 268.75 °C; probe heater temperature, 350 °C; S-lens level, 70) were detected at a resolution of 70,000 (automatic gain control target: $1 \times 10^6$; maximum injection time, 100 ms).

**Nuclear magnetic resonance.** HPLC-purified peptides were dissolved in 150 µl of dimethylsulfoxide (DMSO)-$d_6$ and transferred into 3-mm NMR tubes (Bruker). NMR spectra were acquired on a Bruker 600 (NEO) spectrometer with TopSpin 3.5/4.1 (Bruker). $^1$H-NMR chemical shifts are reported in parts per million relative to $SiMe_4$ ($\delta = 0$), with internal referencing to residual DMSO protons ($\delta = 2.50$ for DMSO-$d_6$). Coupling constants are reported in Hertz, with peak assignments based on calculated chemical shifts and multiplicity. Splitting patterns are reported as singlet (s), doublet (d), triplet (t), quartet (q) or multiplet (m).

**Inhibition assays.** Protease inhibition against human neutrophil elastase and cathepsin B was assessed using Neutrophil Elastase Inhibitor and Cathepsin B Inhibitor Screening Assay kits (BPS Bioscience, catalogue nos. 82090-1 and 79590-1, respectively), following the manufacturer's protocols. Trypsin (Promega) and chymotrypsin (Promega) inhibition assays were performed as follows: 2 µl of enzyme solution (3 nM for trypsin, 1 nM for chymotrypsin) was mixed with 48 µl of protease buffer (40 mM Tris-HCl, 10 mM $CaCl_2$, pH 8.0) and 25 µl of peptide solution (2× dilution in 50 mM Tris-HCl pH 8.0 + 10% DMSO). After 10 min at room temperature, 23 µl of buffer (50 mM Tris, pH 8.0) and 2 µl of fluorogenic substrate (500 µM of Boc-Ile-Glu-Gly-Arg-AMC (PeptaNova catalogue no. 3114-v) for trypsin, Suc-Ala-Ala-Pro-Phe-AMC (PeptaNova catalogue no. 3094-v) for chymotrypsin, respectively, in DMSO) were added. Plates were incubated at 37 °C in the dark for 1 h and enzyme activity was measured by monitoring fluorescence ($\lambda_{ex} = 342$ nm, $\lambda_{em} = 440$ nm) using BioTek Synergy H1 microplate reader (Agilent). Data were processed using Prism v.9.

### Statistics and reproducibility

Statistical analyses were performed in R (v.4.2.2–v.4.3.1). Wherever appropriate, correction for multiple testing was performed using false-discovery rate correction. Wherever appropriate and if not specified otherwise, statistical tests performed were two-sided. UpSet plots were generated using the R packages UpSetR (v.1.4.0)[141] and ComplexUpset (v.1.3.3)[142,143], boxplots and violin plots using ggplot2 (v.3.4.2)[144], heatmaps using ComplexHeatmap (v.2.14.0)[145] and maps using leaflet (v.2.1.2)[146].

### Reporting summary

Further information on research design is available in the Nature Portfolio Reporting Summary linked to this article.

## Data availability

*Tara* Pacific short-read metagenomic data generated in this study were submitted to the ENA at the EMBL European Bioinformatics Institute under the *Tara* Pacific umbrella project PRJEB47249 (accession numbers provided in Supplementary Table 1). Sample provenance and environmental context are available at *Zenodo* (https://doi.org/10.5281/ZENODO.4068292)[135]. *Tara* Pacific long-read metagenomic data were submitted to ENA (ERR14224704, ERR14224705, ERR14224706). The publicly available metagenomic data used in this study were downloaded from the ENA, and a summary of their accession numbers is provided in Supplementary Table 1. The MIBiG and BiG-FAM databases can be accessed at https://mibig.secondarymetabolites.org/ and https://bigfam.bioinformatics.nl/, respectively. The reef genomic data used in this study (RMD) are available online (https://rmd.microbiomics.io), and can be interactively explored through the Ocean Microbiomics Database (OMDB; https://omdb.microbiomics.io) as well as contextualized with non-marine environments through the mOTUs online database (https://motus-db.org). All files used for characterizing natural products and all other supporting data are available at Zenodo (https://doi.org/10.5281/ZENODO.14050210)[147] and (https://doi.org/10.5281/ZENODO.10182966)[148].

## Code availability

The code used for the analyses performed in this study is available at GitHub (https://github.com/SushiLab/reef-microbiomics-paper/) and at Zenodo (https://doi.org/10.5281/zenodo.10201847)[149].

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

**Acknowledgements** We thank the *Tara* Ocean Foundation, the RV *Tara* crew and the *Tara* Pacific expedition participants (full list of scientists, sailors, artists and citizens is available at *Zenodo* (https://doi.org/10.5281/zenodo.3777759))[150]. We thank L. Bering, Z. Blocka, C. Carlström, C. Eigenmann, L. Feer, S. Kammerecker, P. Nicholson and L.-G. Williams for their assistance in experimental work and DNA sequencing, and A. Babics and P. Ugolini from Zoologischer Garten Basel for providing access to fresh coral material. We also particularly thank S. Planes, D. Allemand and the *Tara* Pacific consortium. We are keen to thank the commitment of the following institutions for their financial and scientific support that made this unique 3-year *Tara* Pacific expedition possible: CNRS, PSL, CSM, EPHE, Genoscope, CEA, Inserm, Université Côte d'Azur, ANR, agnès b., the Veolia Foundation, the Prince Albert II de Monaco Foundation, Région Bretagne, Lorient Agglomération, L'Oréal, Biotherm, Billerudkorsnas, AmerisourceBergen Company, Oceans by Disney, France Collectivités, Fonds Français pour l'Environnement Mondial (FFEM), UNESCO-IOC, Etienne Bourgois and the *Tara* Ocean Foundation teams. *Tara* Pacific would not exist without the continuous support of the participating institutes. We would like to express our deepest gratitude towards all the countries and local authorities that have enabled and supported the sampling conducted as part of the *Tara* Pacific expedition (Supplementary Information). The sampling permits associated with the data generated as part of this study are available (Supplementary Information and Supplementary Table 12). Although the sequence information associated with these samples has been archived on public repositories according to best practices in the field, it should be noted that any downstream commercial use should be in accordance with international law (Supplementary Table 12). This study was supported in part by FRANCE GENOMIQUE (ANR-10-INBS-0009). S.S. acknowledges the support of the Swiss National Science Foundation (SNSF) (project grant nos. 205321_184955, 205320_215395 and the NCCR Microbiomes grant no. 51NF40_180575) and support from the ETH IT services for computations on the ETH Euler cluster. S.M.-V was supported by a Human Frontiers Science Programme (HFSP) long-term fellowship (LT0050/2023-L1). J.P. is grateful for funding from the Gordon and Betty Moore Foundation (grant no. 9204, https://doi.org/10.37807/GBMF9204), the SNSF (grant no. 20530_219638) and the Dutch Research Council (grant no. KICH1.LWV04.21.013). This is publication number 38 of the *Tara* Pacific consortium.

**Author contributions** S.S., J. Piel and L.P. conceived and co-supervised the project. F.W., L.P., D. Richter, S.S., J. Piel and *Tara* Pacific Consortium Coordinators designed the research. F.W., D. Richter, D. Racunica, M.S., J.O.'B., A.B.S., J.R., C.C., K.S.I.B., T.S., K.L., A.P., J. Poulain, P.H.O. and P.W. produced data. L.P., D. Richter and H.-J.R. curated data. F.W., L.P., D. Richter, D. Racunica, M.S., J.O.'B., S.M.-V., A.B.S., C.C., T.P., A.S., G.S., K.S.I.B. and T.S. analysed data. L.P., H.-J.R. and S.M.-V. designed and implemented web and data resources. K.-I.M., M.M.R., C.M., E.B., G.B., J.C., G.I., J. Poulain, S.R., S.A., C.B., E.D., D.F., P.E.G., F.L., P.H.O., O.P.T., R.V.T., R.T., C.R.V., P.W., M.Z., S.S., D.A. and S.P. managed the project, provided resources, or coordinated the field work. *Tara* Pacific Coordinators provided constructive criticism throughout the study. F.W., L.P., D. Richter, S.S. and J. Piel drafted the manuscript with input from all authors. All of the authors reviewed the submitted manuscript and approved the final version.

**Competing interests** The authors declare no competing interests.

**Additional information**


**Correspondence and requests for materials** should be addressed to Lucas Paoli, Jörn Piel or Shinichi Sunagawa.


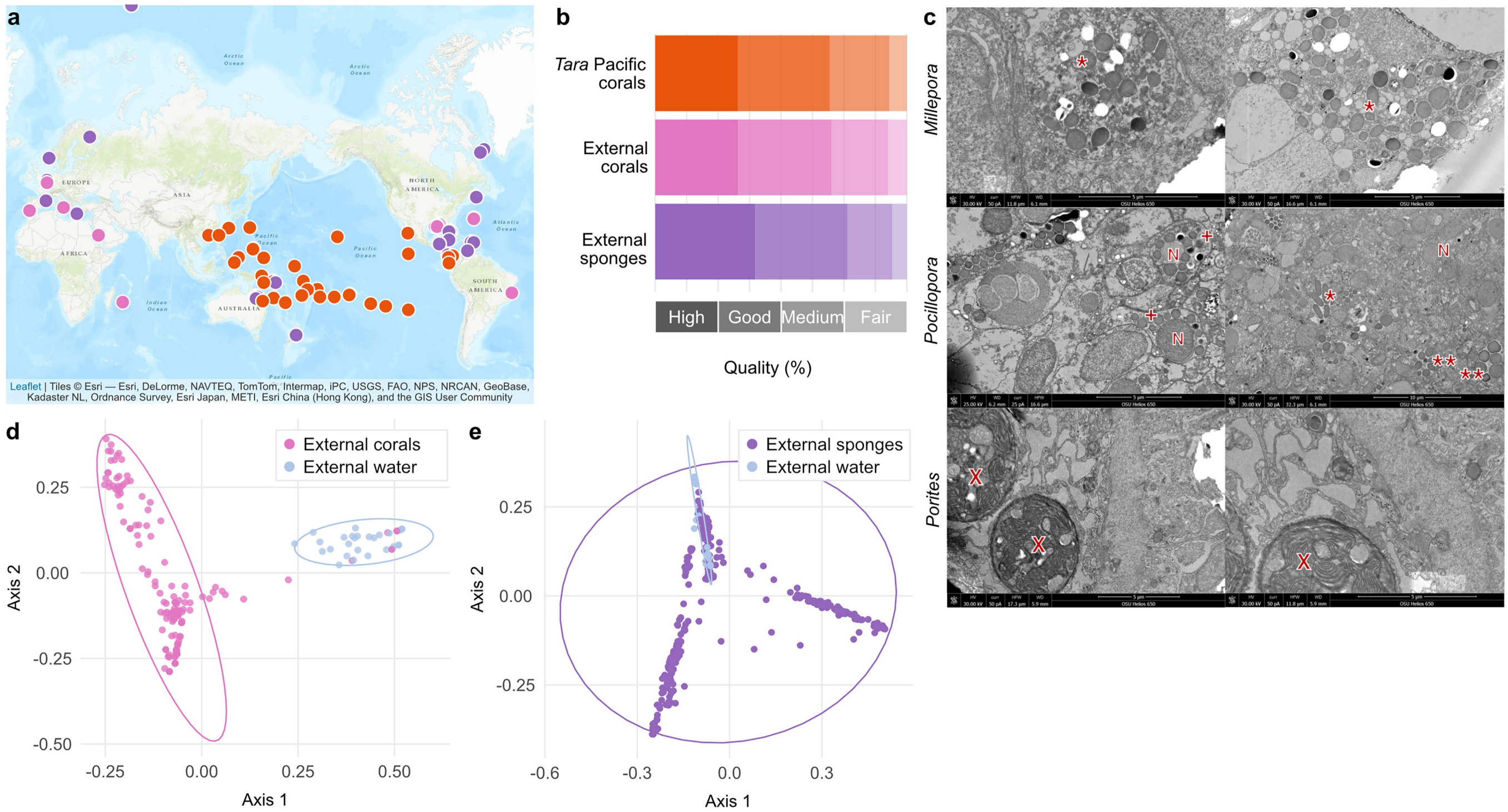


**Extended Data Fig. 1 | Global distribution of publicly available coral and sponge metagenomes, MAG quality assessment, localization of microbial aggregates in coral tissues and host-specificity of coral and sponge microbiomes. a**, The coral reef metagenomes used in this study are distributed across the globe. We generated a genomic resource with global representation by integrating the 820 coral metagenomes sampled by *Tara* Pacific (orange) with publicly available coral metagenomes (412) from 15 studies (pink) and sponge metagenomes (371) from 16 studies (purple). **b**, We attributed each metagenome-assembled genome (MAG) a quality score based on its completeness and degree of contamination (Methods). For each dataset, at least one third of all MAGs were of high quality (completeness ≥ 90% and contamination ≤ 5%) and more than two thirds were of good quality or higher (completeness ≥ 70% and contamination ≤ 10%), which compares favourably to previous large-scale efforts focusing on other microbiomes[22,24]. **c**, By comparing the transmission electron microscopy images of the three coral genera *Millepora* (top), *Pocillopora* (middle), and *Porites* (bottom) (sampled in Guam in 2016), we find differences in the number of microbial cells present as well as in the niche the microorganisms colonize. Annotations denote possible: (*) extracellular bacteria/archaea, (**) coral-associated microbial aggregates (CAMAs), (+) intracellular bacteria/archaea, (X) Symbiodinaceae (symbiont, including intracellular thylakoids), (N) coral nuclei. **d**, A clear separation between the microbiomes of corals and seawater is found based on a Jaccard distance-based PCoA of the microbial species detected in coral ($n = 143$) and seawater ($n = 28$) metagenomes from publicly available coral studies (PERMANOVA, *P* value ≤ 0.001, $R^2 = 0.17$; Methods). The coral samples found within the water sample cluster originate from the surface mucus layer of corals, which is known to be influenced by the water column[151]. **e**, Similarly, the sponge metagenomes ($n = 336$) harbour microbial species that are distinct from those found in seawater metagenomes ($n = 47$), although the difference is weaker than what was observed between coral and seawater metagenomes (PERMANOVA, *P* value ≤ 0.001, $R^2 = 0.08$). Map in **a** adapted from Esri (https://server.arcgisonline.com/arcgis/rest/services/World_Topo_Map/MapServer). Imagery in this work is owned by Esri and its data contributors and are used herein with permission. 

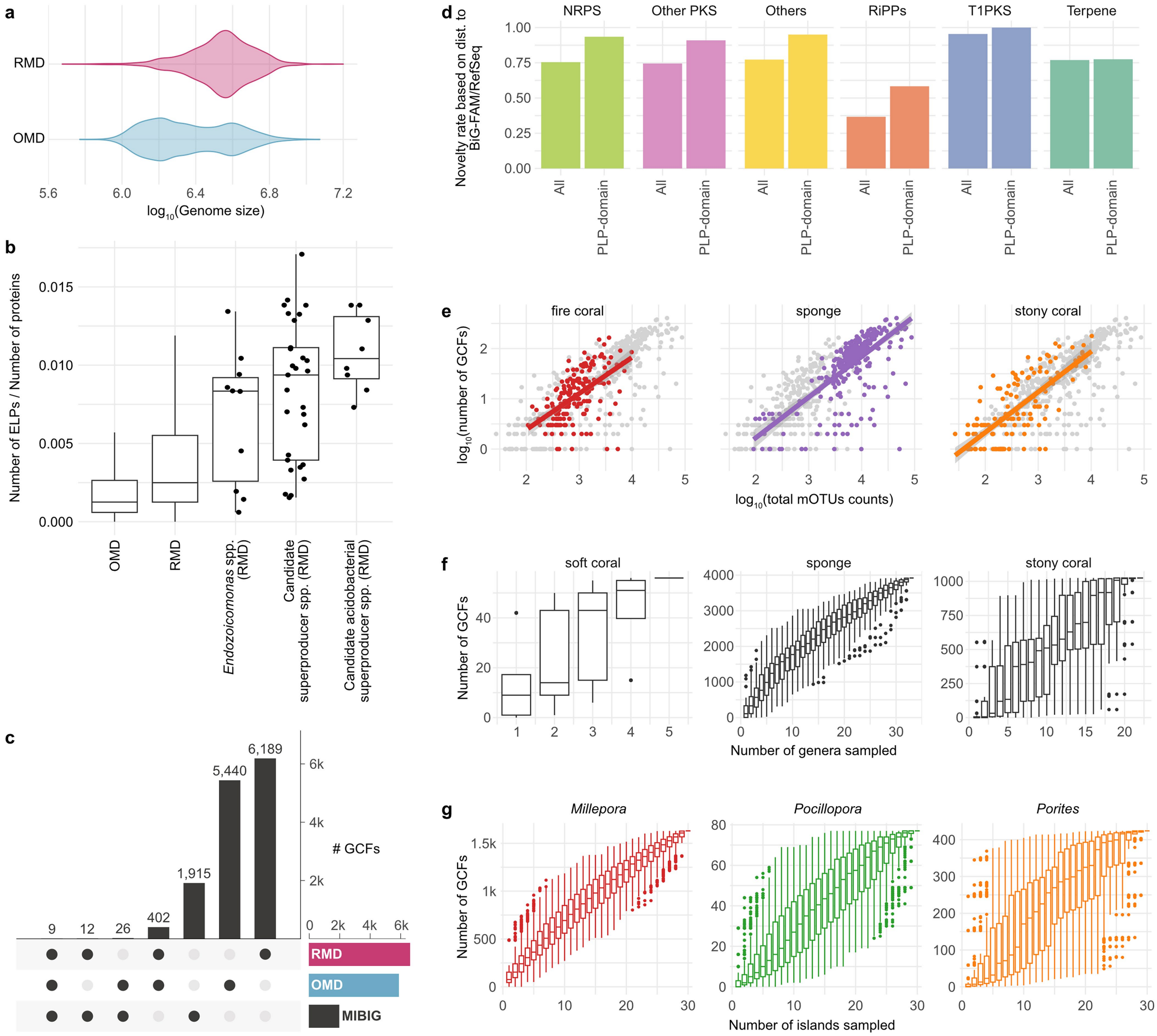


**Extended Data Fig. 2 |** See next page for caption.

**Extended Data Fig. 2 | Genomic features, biosynthetic richness and novelty and sampling depth effects in reef-associated microbiomes. a**, Using genome size estimates correcting for incompleteness (Methods)[24], we found the median genome size of microbial species associated with reef invertebrates ($n = 3,567$) to be 3.6 Mbp, which was larger than that of the open ocean ($n = 5,740$) (2.2 Mbp; Wilcoxon-test, $P$ value < 2.2e-16). **b**, Compared to the OMD ($n = 5,514$), we identified a substantial enrichment of eukaryotic-like proteins (ELPs; implicated in host infection and symbiosis establishment) in the RMD ($n = 3,388$). Amongst the reef host-associated microorganisms, we found Acidobacteriota spp. ($n = 8$) and candidate super-producer spp. ($n = 29$) in general to be most enriched in ELPs (and more so than other previously reported coral symbionts such as *Endozoicomonas* spp. ($n = 10$)). Boxes show the median (centre line) and interquartile range (box bounds, the 25th and 75th percentiles), with whiskers extending to the largest or smallest value no further than 1.5× the interquartile range. Individual data points are shown for $n \leq 100$ and outliers are not shown for $n \geq 3,000$ due to visualization constraints. **c**, We found the GCFs detected in the reef microbiome to be clearly distinct from those previously detected in the open ocean[24], and only 21 of them to overlap with biochemically characterized ones (MIBiG). **d**, We screened all biosynthetic core genes of the RMD BGCs for putative PLP domains using the PFAM HMM model of PLP-dependent enzymes (PF00291) and hmmsearch (HMMER v3.4) with the option --*cut_nc*. We identified a total of 856 PLP-domains containing BGCs, spanning 152 GCFs. Those GCFs were found to have an increased novelty rate (84%) compared to the novelty of all GCFs across the RMD (64%), with variation across GCF classes. **e**, We sought to investigate the impact of sequencing depth (as captured by the total mOTUs count, a proxy for the number of bacterial and archeal cells sequenced) on the number of GCFs identified in a metagenome across host groups. The higher number of GCFs identified in sponge metagenomes can be attributed to a higher number of microbial cells sequenced. The overall trends are similar across host groups after normalising by sequencing effort (the 95% confidence intervals of the slope estimates overlap). **f**, The rarefaction curves of the number of GCFs identified against the number of coral/sponge genera sampled by previous studies (based on 100 randomizations) do not reach saturation and therefore suggest that we can expect to uncover additional GCFs by increasing the host diversity range. The higher variation across soft and stony corals may suggest that the biosynthetic potential of their microbiome is more heterogeneous between genera. However, this higher variation may also result from the variable sampling effort and the various sampling protocols used (as they originate from many different studies) across genera. Boxes show the median (centre line) and interquartile range (box bounds, the 25th and 75th percentiles), with whiskers extending to the largest or smallest value no further than 1.5× the interquartile range and outliers beyond whiskers shown as individual data points. **g**, The rarefaction curves of the number of GCFs identified across coral genera sampled by *Tara* Pacific (*Millepora*, *Porites*, *Pocillopora*) against the number of islands sampled (based on 500 randomisations) do not reach saturation and therefore suggest that we can uncover additional GCFs by further increasing the geographic range of sampling. The higher variation in *Porites* may indicate that the biosynthetic potential of its microbiome is more geographically heterogeneous. Boxes show the median (centre line) and interquartile range (box bounds, the 25th and 75th percentiles), with whiskers extending to the largest or smallest value no further than 1.5× the interquartile range and outliers beyond whiskers shown as individual data points.

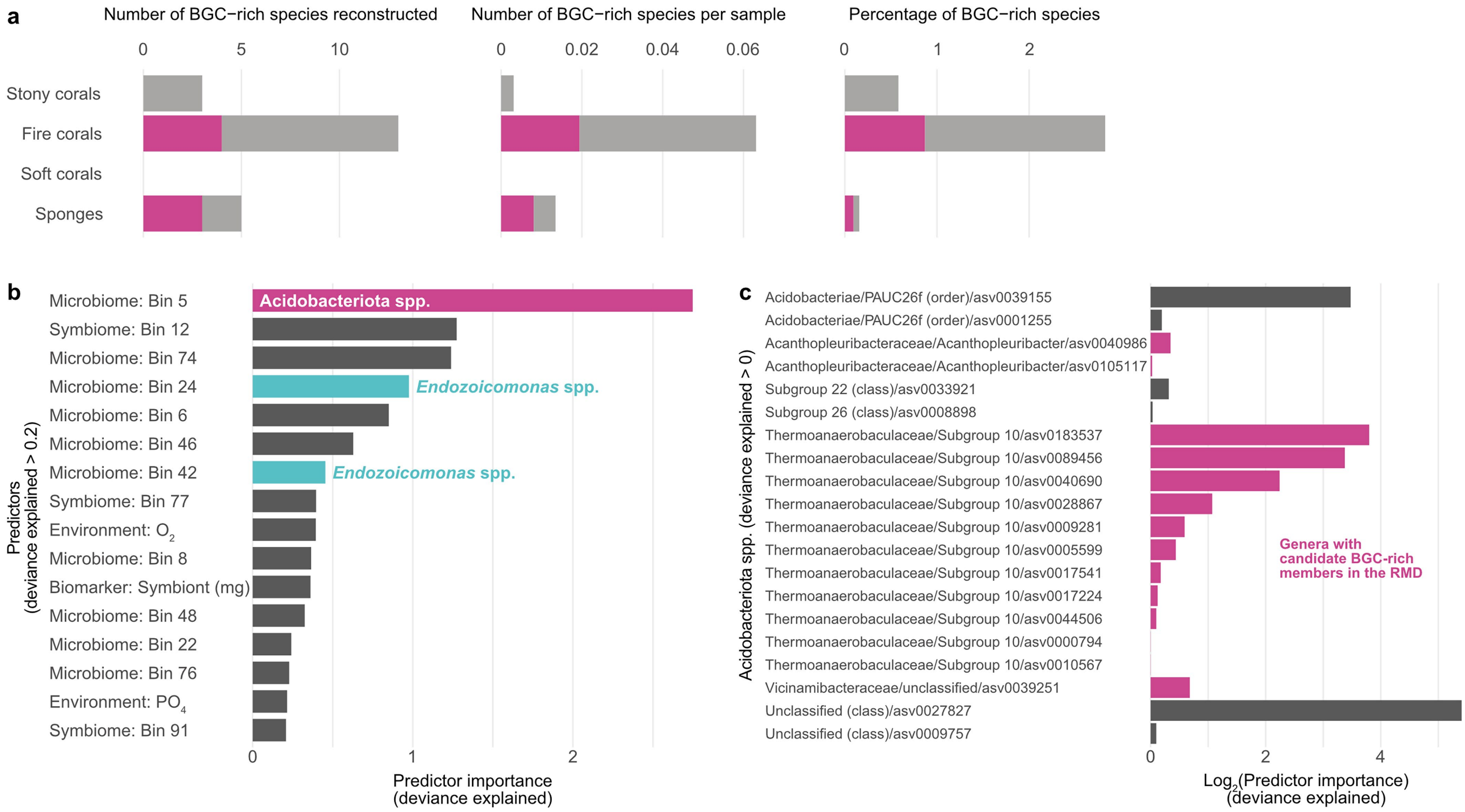


**Extended Data Fig. 3 | BGC-rich microbial lineage representation and association with coral host gene expression. a**, Fire corals host a particularly high number of BGC-rich species compared to other reef invertebrates (and we detected none in soft corals), a difference even more striking when normalizing by the sampling effort (Methods). Compared to both sponges and soft corals, the microbiomes of reef-building corals host a higher proportion of BGC-rich species. The coloured fraction of the bars indicate Acidobacteriota spp. **b**, We identified a bin (group of ASVs) exclusively containing Acidobacteriota spp. to best predict *Pocillopora* host transcriptome dissimilarity using algal endosymbiont (ITS2 amplicons) and microbiome (16S rRNA gene ASVs) compositions as well as environmental and biomarker data in a generalized dissimilarity model (GDM). The model was generated based on 29 samples for which all data types were available. Dimensionality of algal symbiont and microbiome community composition was reduced by taxonomic and abundance binning (Methods). **c**, A GDM based on a larger dataset of 58 samples with paired transcriptome and microbiome data highlights the individual contributions of Acidobacteriota ASVs, and in particular BGC-rich candidates (Methods), in predicting host transcriptome dissimilarities.

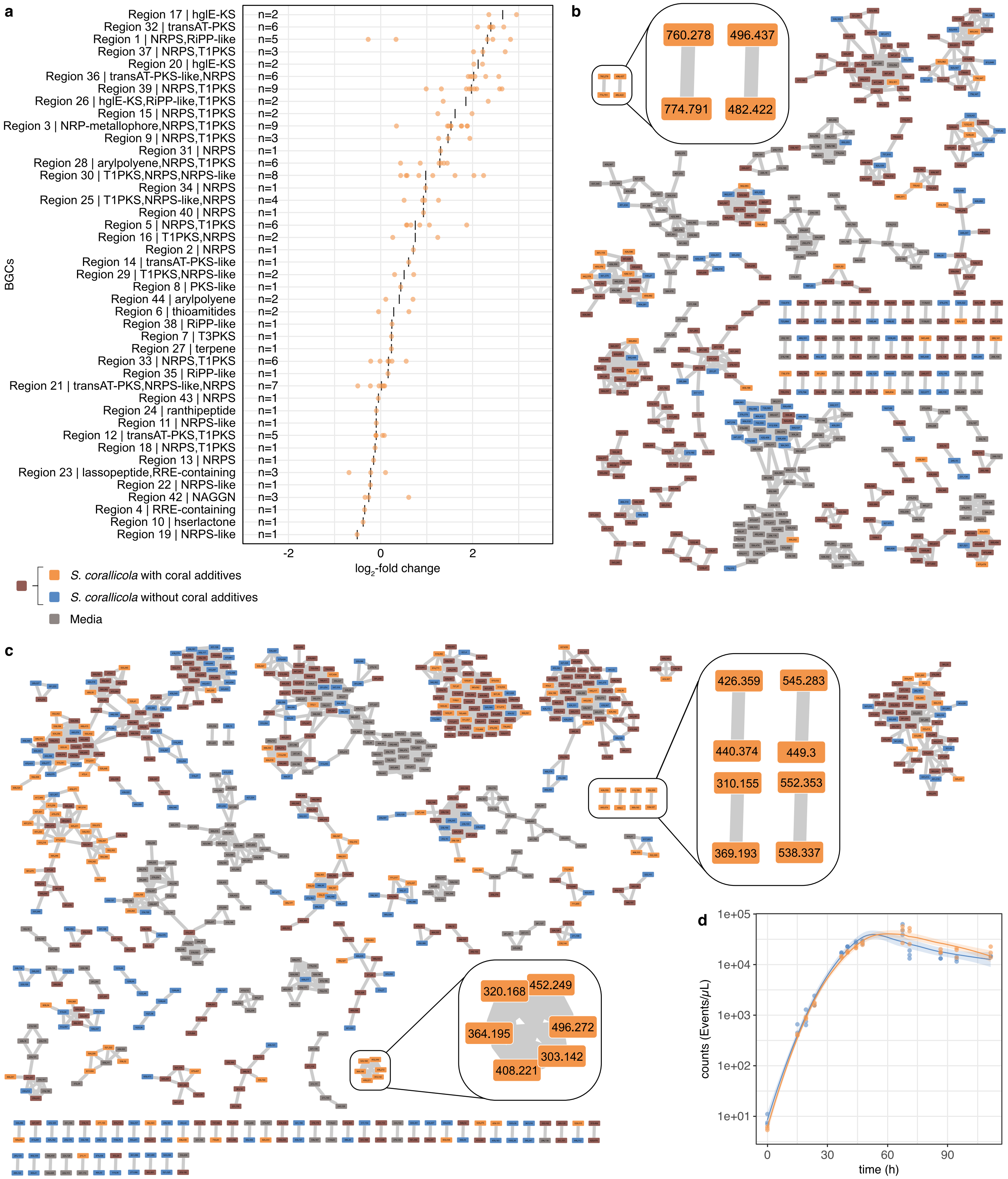


**Extended Data Fig. 4 |** See next page for caption.

**Extended Data Fig. 4 | Transcriptional and metabolic responses of *Sulfidibacter corallicola* to coral-derived compounds.** To examine the effect of the host on the growth, transcription profile, and actual metabolite production of the coral-associated bacterium *S. corallicola*, we cultivated the bacteria in 1/10 MB liquid medium with and without *Porites* coral tissue extract (Methods). The flasks were incubated at 28 °C and 150 rpm in the dark. **a**, RNA-sequencing was conducted with cells after reaching stationary phase (that is, when bacteria are best known to produce specialized metabolites). Each dot represents the core genes of respective BGC regions as identified by antiSMASH. Gene expression is shown as $\log_2$-fold changes (DESeq2), with positive values indicating upregulation in the presence of coral extracts (with the black line being the median). Among the 43 biosynthetic core genes, 21 BGCs (49%) were > 1.5-fold (median $\log_2$-fold change > 0.585) and 13 BGCs (30%) > 2-fold upregulated (median $\log_2$-fold change > 1.0). **b**,**c**, Molecular networks of *S. corallicola* extracts from the bacterial pellet (**b**) and the supernatant (**c**) were constructed from HPLC-HESI-HRMS/MS data. Nodes for metabolites expressed with or without coral additives are shown in orange and blue, respectively, while brown nodes represent those expressed in both conditions. Grey nodes show metabolites present in the culture medium (with or without coral additives). Numbers within nodes indicate ion masses (*m/z*). Enlarged regions highlight networks with only orange nodes, representing pathways exclusively activated by coral additives. **d**, For enumerating the microbial cells, we stained bacteria with SYBR-Green I (final concentration 1:20,000) and incubated them in the dark and at room temperature for 20 min. For each sample, we recorded forward scatter (FSC) and green fluorescence (488 nm, SYBR) to identify bacterial cells. Sample flow rate was set to 30 µl min$^{-1}$ and time to record to 30 s. The solid line represents the smoothed trend in bacterial counts over time, and the shaded area shows the 95% confidence interval around this fitted curve.

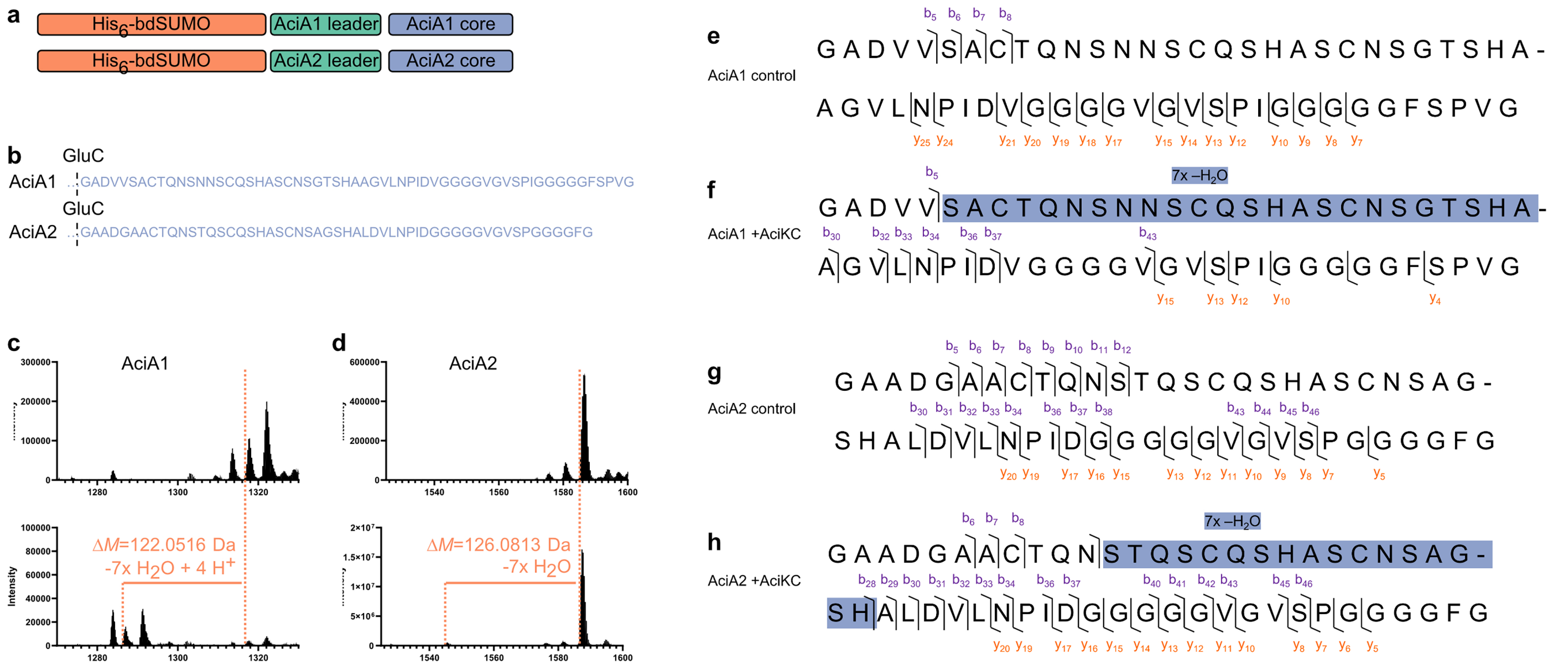

**Extended Data Fig. 5 | Biochemical characterization of AciKC-modified lanthipeptide precursor proteins. a**, *aci* precursor proteins were heterologously produced as N-terminal fusion proteins with His$_6$-bdSUMO. **b**, Core peptide sequences in AciA1 and AciA2. No clear leader-core boundaries could be predicted. The used GluC cleavage site is indicated. **c**,**d**, Mass spectra showing precursor expression alone (top) and in the presence of AciKC (bottom). For both precursors, masses corresponding to seven dehydrations were detected. In the case of AciA1 two disulfide bridges appear to be broken, most likely involved in (methyl-)lanthionine or labionin formation. **e**–**h**, Localization of dehydrations in His$_6$-bdSUMO-AciA1 and His$_6$-bdSUMO-AciA2 catalysed by AciKC. Summary of observed b (above, purple) and y (below, orange) fragmentation ions for unmodified (**e**,**g**) and AciKC-modified (**f**,**h**) AciA1 (**e**,**f**) and AciA2 (**g**,**h**) from TOP10-mediated fragmentation (CE 24). The regions containing seven dehydrations are highlighted in blue.

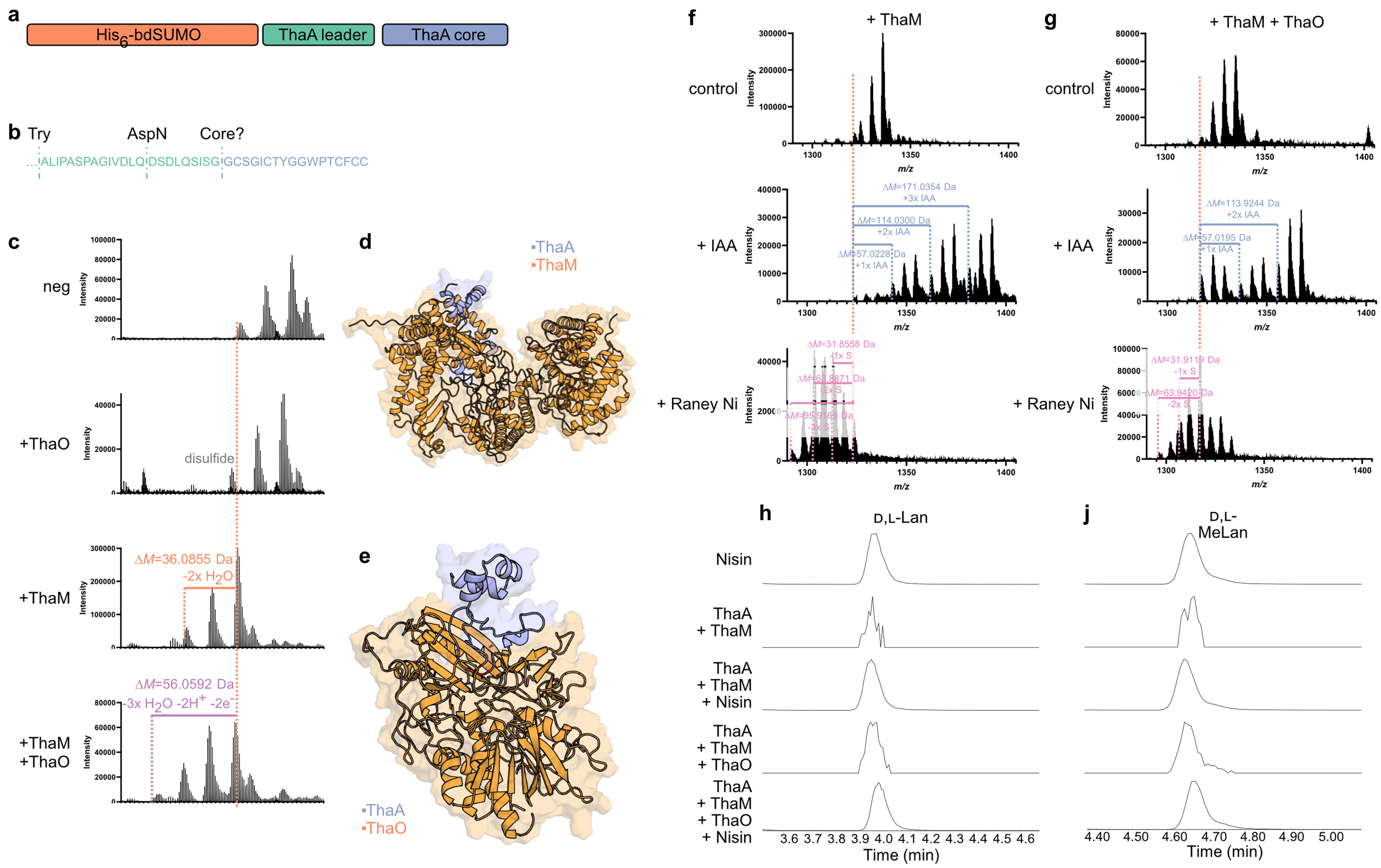


**Extended Data Fig. 6 | Biochemical characterization of ThaM- and ThaO-catalysed modifications in ThaA. a**, ThaA was heterologously produced as an N-terminal fusion protein with His$_6$-bdSUMO. **b**, Core peptide sequences of ThaA. The used Trypsin and AspN cleavage sites are indicated. **c**, Mass spectra showing precursor expression alone (top row), in the presence of ThaO (second row), ThaM (third row), and both ThaM and ThaO (bottom row). Dehydration was observed in the presence of ThaM and a further mass loss of 20 Da was detected in the presence of ThaO. **d**,**e**, AlphaFold2-predicted multimer structures of ThaA with ThaM (**d**) and ThaO (**e**). **f**,**g**, Derivatization of purified, dehydrated (**f**) and fully modified (**g**) ThaA digested with trypsin treated with nothing (control, top row), iodoacetamide (IAA; middle row), and Raney nickel (Raney Ni; bottom row). The dehydrated peptide is modified by three equivalents of iodoacetamide and loses three sulphur equivalents in the presence of Raney nickel. The fully modified peptide is modified by two equivalents of iodoacetamide and loses two sulphur equivalents in the presence of Raney nickel, suggesting modification of a cysteine residue by ThaO. **h**,**j**, Marfey's derivatization for stereochemistry determination of (methyl-)lanthionine bridges. Extracted ion chromatogram for **h** *m/z* 489.1405 ($[M + H]^+$, lanthionine, Lan) and **j** *m/z* 503.1561 ($[M + H]^+$, methyllanthionine, MeLan) of hydrolysed and derivatized nisin A (top row), tryptic fragment of dehydrated ThaA (second row), and fully modified ThaA (fourth row). The (methyl-)lanthionines from dehydrated and fully modified ThaA co-elute with the same retention time as the ones from nisin A, indicating D,L-(methyl)-lanthionine formation. Co-injections of dehydrated ThaA (third row) and fully modified ThaA (bottom row) with nisin A co-elute.

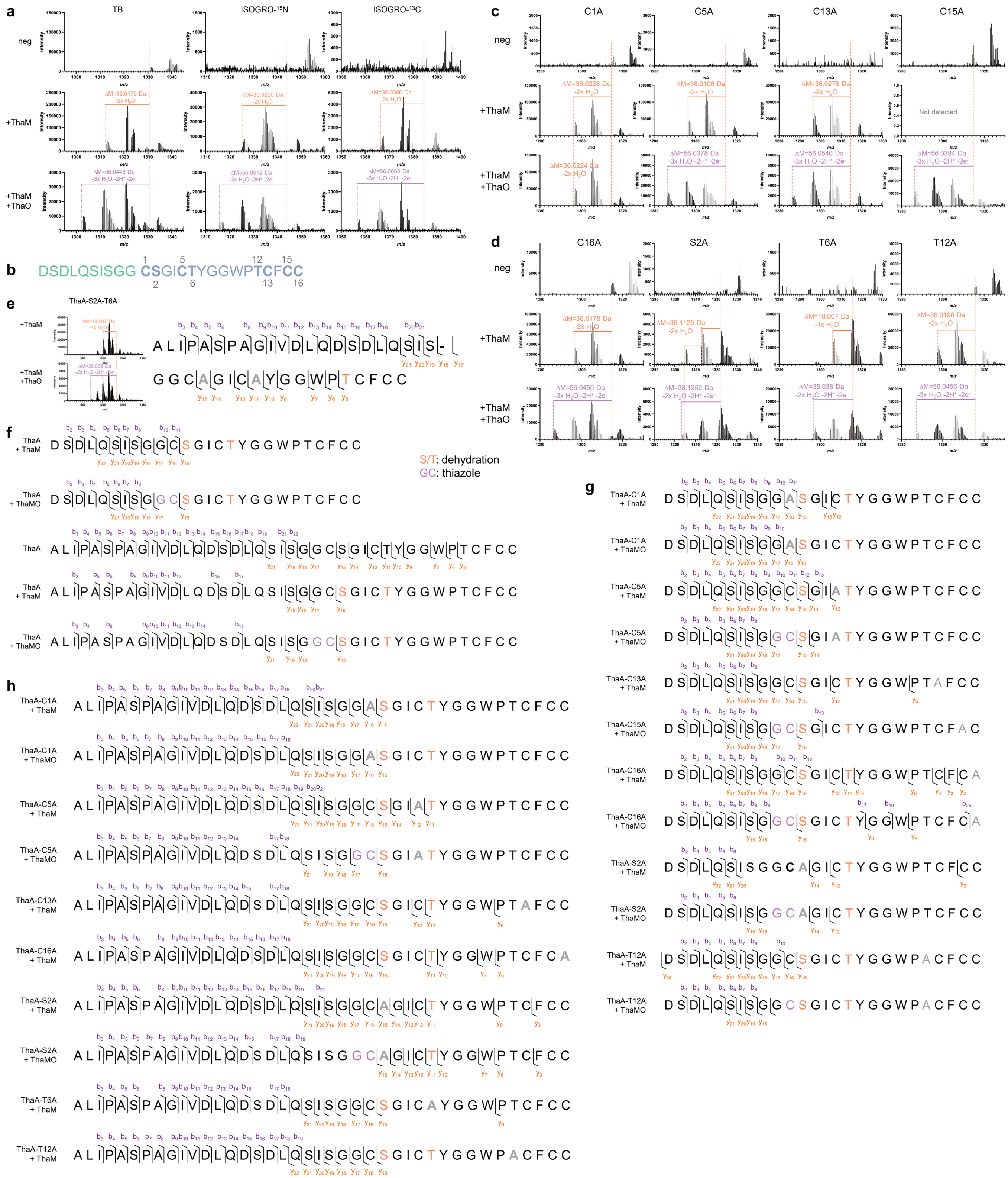

**Extended Data Fig. 7 |** See next page for caption.

**Extended Data Fig. 7 | Isotope labelling and site-directed mutagenesis mapping of ThaM and ThaM/ThaO modification sites in ThaA. a**, ThaA was heterologously produced as N-terminal fusion proteins with $His_6$-bdSUMO in terrific broth (TB, left), ISOGRO-$^{15}$N (centre) and ISOGRO-$^{13}$C (right) alone (top row), in the presence of ThaM (middle row) or in the presence of ThaM and ThaO (bottom row) and digested with AspN. For all labelled media experiments, the same mass losses of 36 Da for the dehydrations and additional 20 Da for ThaO-mediated reactions were detected, excluding carbon or nitrogen loss. **b**, Numbering scheme in the core regions used for point mutations. Positions mutated to alanine are highlighted with numbers. **c**,**d**, Point mutants were produced alone (top row), in the presence of ThaM (middle row) or in the presence of ThaM and ThaO (bottom row) and digested with AspN. Point mutation C1A prevented modification by ThaO, suggesting Cys1 is involved in the post-translational modification. Point mutation T6A abolished production of one dehydration, indicating that Thr6 is dehydrated. While point mutation S2A leads to a product that is dehydrated twice by ThaM, a product with only one dehydration and the 20 Da loss is detected in the presence of ThaM and ThaO, suggesting that Ser2 is also dehydrated in the native product and in the absence of Ser2, Thr12 can be dehydrated by ThaM. Point mutant T12A undergoes normal dehydration and modification by ThaO. **e**, Double mutant ThaA-S2A-T6A digested with trypsin is dehydrated by ThaM (top) and undergoes further modification by ThaO (bottom row) (left). These results are in agreement with the S2A and T12A mutants, suggesting that Thr12 can be alternatively dehydrated by ThaM. Localization of dehydration in the dehydrated species of ThaA-S2A-T6A (right). Summary of observed b (above, purple) and y (below, orange) fragmentation from TOP10-mediated fragmentation (CE 24). **f**, Localization of dehydrations and 20 Da loss in $His_6$-bdSUMO-ThaA digested with AspN (first two rows) or Trypsin (last three rows). Summary of observed b (above, purple) and y (below, orange) fragmentation ions for unmodified (third row), ThaM-modified (first and fourth row) and ThaM/ThaO-modified (second and fifth row) ThaA from TOP10-mediated fragmentation (CE 24). Orange amino acids are dehydrated and purple amino acids are involved in thiazole formation. **g**, Summary of observed b (above, purple) and y (below, orange) fragmentation ions for ThaM-modified (odd rows) and ThaM/ThaO-modified (even rows) point mutants of ThaA from PRM-mediated fragmentation (CE 24). Orange amino acids are dehydrated and purple amino acids are involved in thiazole formation. **h**, Summary of observed b (above, purple) and y (below, orange) fragmentation ions for ThaM-modified (first, third, fifth to seventh and ninth to tenth row) and ThaM/ThaO-modified (second, fourth and eighth row) point mutants of ThaA from PRM-mediated fragmentation (CE 24). Orange amino acids are dehydrated and purple amino acids are involved in thiazole formation.

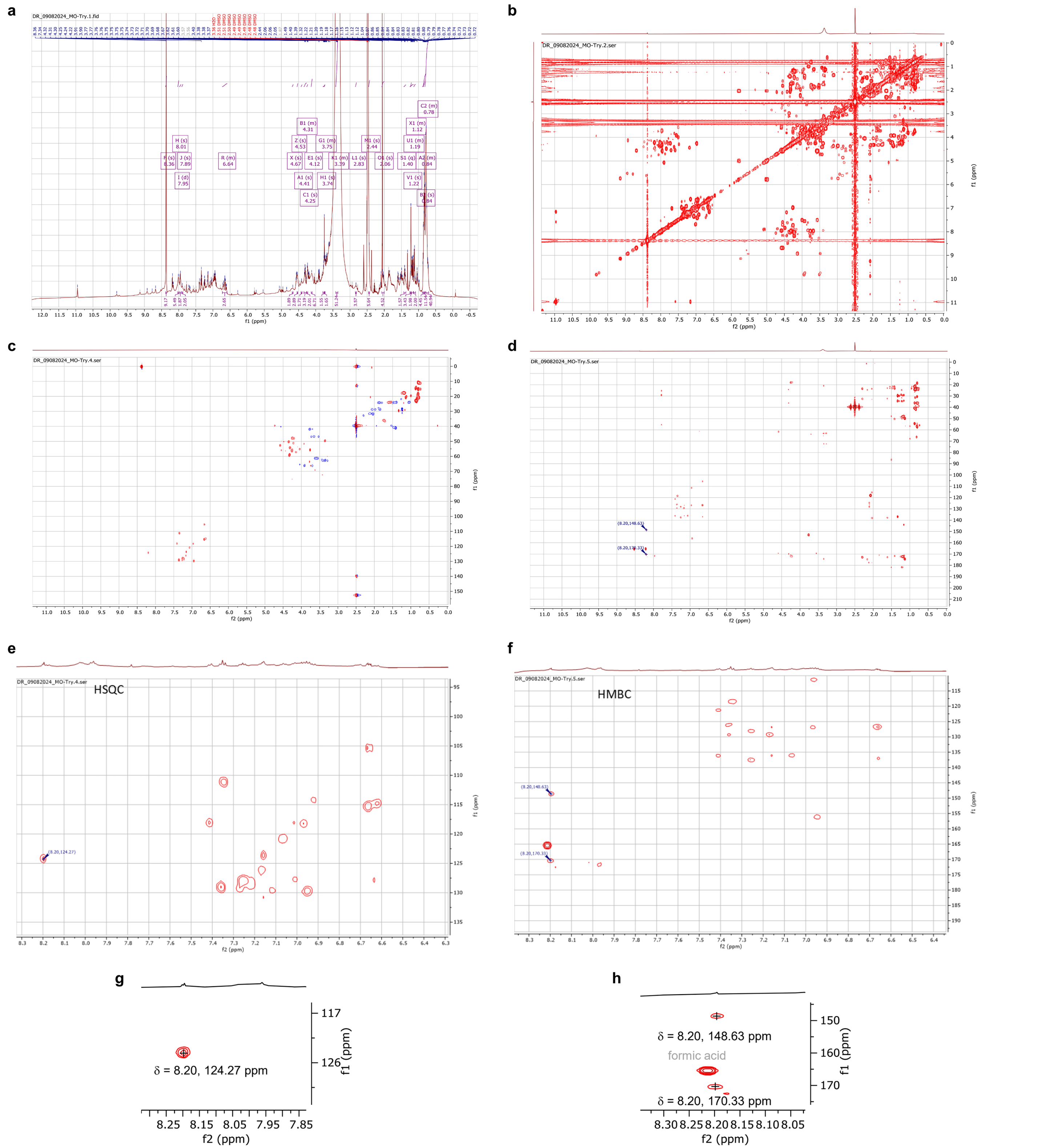


**Extended Data Fig. 8 | NMR spectroscopic characterisation of ThaO-catalysed thiazole formation in ThaA. a**, $^{1}$H NMR (600 MHz) spectrum of fully modified ThaA digested with trypsin in DMSO-$d_6$. **b**, $^{1}$H-$^{1}$H COSY (600 MHz) spectrum of fully modified ThaA digested with trypsin in DMSO-$d_6$. **c**, $^{1}$H-$^{13}$C HSQC (600 MHz) spectrum of fully modified ThaA digested with trypsin in DMSO-$d_6$. **d**, $^{1}$H-$^{13}$C HMBC (600 MHz) spectrum of fully modified ThaA digested with trypsin in DMSO-$d_6$. **e**,**f**, Expanded HSQC (**e**) and HMBC (**f**) spectra of fully modified ThaA digested with trypsin in DMSO-$d_6$ showing correlations in the thiazole moiety. **g**,**h**, Key $^{1}$H–$^{13}$C NMR correlations in the HSQC (**g**) and HMBC (**h**) spectra revealing thiazole formation catalysed by ThaO.

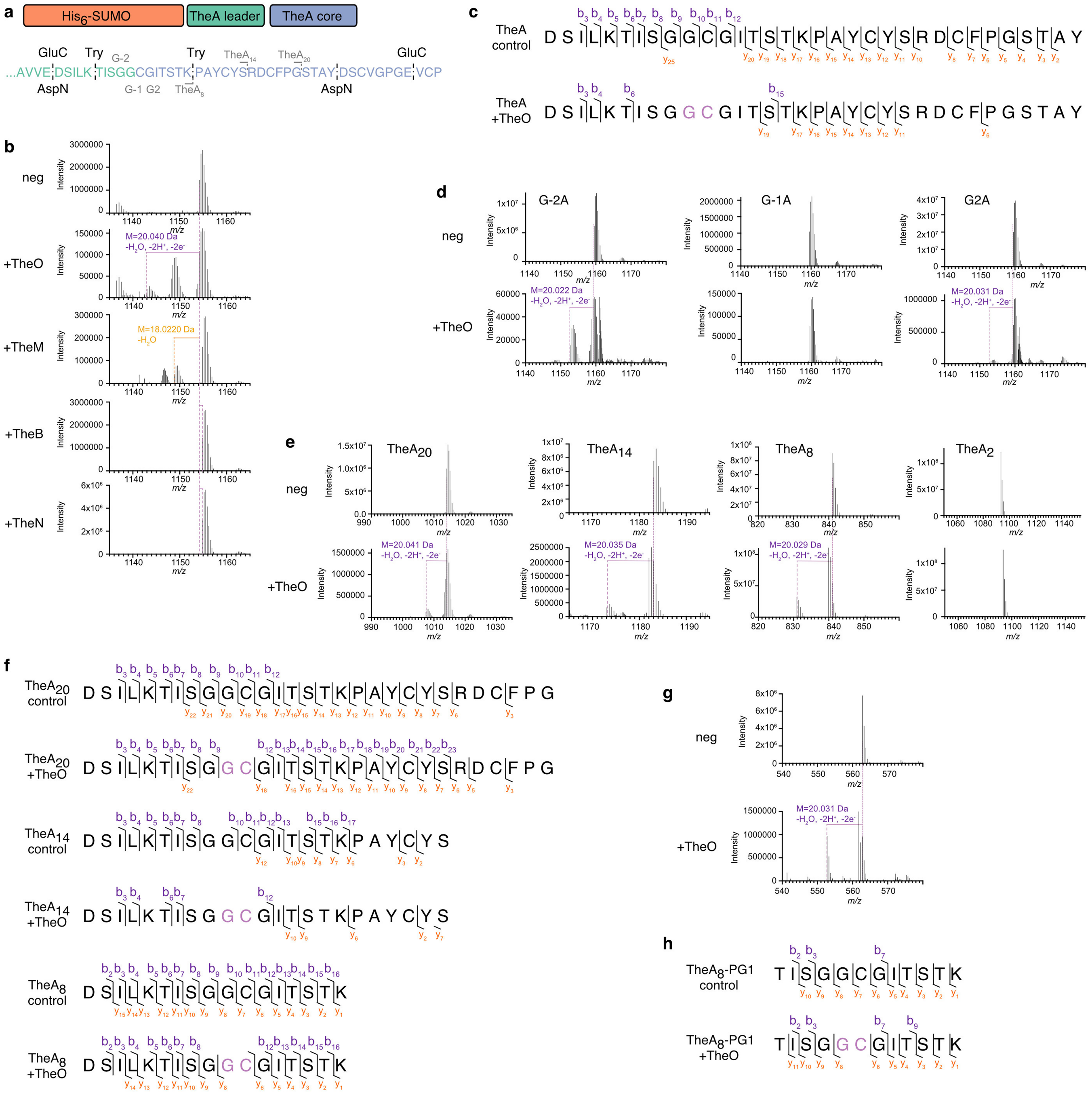


**Extended Data Fig. 9 | Substrate requirements of TheO-catalysed thiazole formation in TheA variants. a**, TheA was heterologously produced as N-terminal fusion proteins with $His_6$-bdSUMO. **b**, A 20 Da loss can be detected in TheA when co-produced with TheO. A 18 Da mass loss can be observed when TheA is co-produced with TheM. For all other enzymes we did not observe any mass shift in TheA when co-produced. **c**, Localization of 20 Da loss in $His_6$-bdSUMO-TheA digested with AspN. **d**, Point mutants were produced alone (top row) or in the presence of TheO (bottom row) and digested with AspN. Point mutations G-1A and G2A impaired modification by TheO. Point mutant G-2A undergoes normal modification by TheO. **e**, Truncated versions of TheA were produced alone (top row) or in the presence of TheO (bottom row) and digested with AspN. Precursors with 20, 14 and 8 amino acid long putative cores undergo modification by TheO, while a 2-amino acid long core does not. **f**, Summary of observed b (above, purple) and y (below, orange) fragmentation ions for TheO-modified truncated TheA variants from PRM-mediated fragmentation (CE 24). Purple amino acids are involved in thiazole formation. **g**, A cargo-fused TheA variant with Protegrin-1 is efficiently modified by TheO when co-produced. **h**, Localization of 20 Da loss in $His_6$-bdSUMO-TheA digested with trypsin, confirming thiazole formation.

nature portfolio



# Reporting Summary


Please do not complete any field with "not applicable" or n/a. Refer to the help text for what text to use if an item is not relevant to your study.
For final submission: please carefully check your responses for accuracy; you will not be able to make changes later.

## Statistics

For all statistical analyses, confirm that the following items are present in the figure legend, table legend, main text, or Methods section.

| n/a | Confirmed | |
|---|---|---|
| ☐ | ☒ | The exact sample size (*n*) for each experimental group/condition, given as a discrete number and unit of measurement |
| ☐ | ☒ | A statement on whether measurements were taken from distinct samples or whether the same sample was measured repeatedly |
| ☐ | ☒ | The statistical test(s) used AND whether they are one- or two-sided<br>*Only common tests should be described solely by name; describe more complex techniques in the Methods section.* |
| ☐ | ☒ | A description of all covariates tested |
| ☐ | ☒ | A description of any assumptions or corrections, such as tests of normality and adjustment for multiple comparisons |
| ☐ | ☒ | A full description of the statistical parameters including central tendency (e.g. means) or other basic estimates (e.g. regression coefficient) AND variation (e.g. standard deviation) or associated estimates of uncertainty (e.g. confidence intervals) |
| ☐ | ☒ | For null hypothesis testing, the test statistic (e.g. *F*, *t*, *r*) with confidence intervals, effect sizes, degrees of freedom and *P* value noted<br>*Give P values as exact values whenever suitable.* |
| ☒ | ☐ | For Bayesian analysis, information on the choice of priors and Markov chain Monte Carlo settings |
| ☒ | ☐ | For hierarchical and complex designs, identification of the appropriate level for tests and full reporting of outcomes |
| ☒ | ☐ | Estimates of effect sizes (e.g. Cohen's *d*, Pearson's *r*), indicating how they were calculated |

*Our web collection on statistics for biologists contains articles on many of the points above.*

## Software and code

Policy information about availability of computer code

| | |
|---|---|
| Data collection | Data collection is described in detail in Lombard, F. et al. Open science resources from the Tara Pacific expedition across coral reef and surface ocean ecosystems. Sci. Data 10, 324 (2023) and Belser, C. et al. Integrative omics framework for characterization of coral reef ecosystems from the Tara Pacific expedition. Sci. Data 10, 326 (2023). |
| Data analysis | Open Source/Custom:<br>BBMap (v38.79), metaSPAdes (v3.14.1 or v3.15), BWA (v0.7.17-r1188), MetaBAT 2 (v2.12.1), CheckM (v1.1.3), Anvi'o (v7.1), dRep (v3.0.0), GTDB-Tk (v2.1.0), Prodigal (v2.6.3), Barrnap (v0.9), Aragorn (v1.2.41), fetchMGs (v1.2), antiSMASH (v6.1.1 or v7.1.0), mOTUs (v3.1), CD-HIT (v4.8.1), emapper (v2.1.7), DIAMOND (v2.0.15.153), clust-o-matic, blastn (v2.15.0+), Flye (v2.9.3), AlphaFold (v2.2.0), rtk (v0.93.2), SINA (v1.6.0), MOTHUR (v1.41.0), FastTree (v2.1.11), PhyCA, BBTools (v38.18), HTseq count (v2.0.2), GNPS, and R (v4.2.2-4.3.1) with packages ggplot2 (v3.4.2), tidyverse (v2.0.0), leaflet (v2.1.2), data.table (v1.14.8), ComplexHeatmap (v2.14.0), UpSetR (v1.4.0), ComplexUpset (v1.3.3), ape (v5.7-1), DESeq2 (v1.37.4)<br>The code used for the analyses performed in this study is accessible at GitHub (https://github.com/SushiLab/reef-microbiomics-paper/) and archived on Zenodo (https://zenodo.org/doi/10.5281/zenodo.10201847).<br>Commercial:<br>CytExpert v2.5 (Beckman Coulter), Thermo Xcalibur Qual Browser 4.1 (Thermo Fisher Scientific), Xcalibur Freestyle 1.8 SP2 (Thermo Fisher Scientific), Mnova v12 (Mestrelab Research), TopSpin 3.5/4.1 (Bruker), and Prism 9 (GraphPad) |

For manuscripts utilizing custom algorithms or software that are central to the research but not yet described in published literature, software must be made available to editors and reviewers. We strongly encourage code deposition in a community repository (e.g. GitHub). See the Nature Portfolio guidelines for submitting code & software for further information.

## Data

Policy information about availability of data

All manuscripts must include a data availability statement. This statement should provide the following information, where applicable:

- Accession codes, unique identifiers, or web links for publicly available datasets
- A description of any restrictions on data availability
- For clinical datasets or third party data, please ensure that the statement adheres to our policy

Tara Pacific short-read metagenomic data generated in this study were submitted to the ENA at the EMBL European Bioinformatics Institute under the Tara Pacific umbrella project PRJEB47249 (accession numbers provided in Supplementary Table 1). Sample provenance and environmental context are available on Zenodo (https://zenodo.org/doi/10.5281/zenodo.4068292). Tara Pacific long-read metagenomic data were submitted to ENA (ERR14224704, ERR14224705, ERR14224706). The publicly available metagenomic data used in this study were downloaded from the ENA, and a summary of their accession numbers is provided in Supplementary Table 1. The MIBiG and BiG-FAM databases can be accessed at https://mibig.secondarymetabolites.org/ and https://bigfam.bioinformatics.nl/, respectively. The reef genomic data used in this study (RMD) are available online (https://microbiomics.io/reef/), and can be interactively explored through the Ocean Microbiomics Database (OMDB; https://omdb.microbiomics.io) as well as contextualised with non-marine environments through the mOTUs online database (https://motus-db.org). All files used for characterising natural products (https://zenodo.org/doi/10.5281/zenodo.14050210) and all other supporting data (https://zenodo.org/doi/10.5281/zenodo.10182966) were deposited on Zenodo.

## Research involving human participants, their data, or biological material

Policy information about studies with human participants or human data. See also policy information about sex, gender (identity/presentation), and sexual orientation and race, ethnicity and racism.

| | |
|---|---|
| Reporting on sex and gender | NA |
| Reporting on race, ethnicity, or other socially relevant groupings | NA |
| Population characteristics | NA |
| Recruitment | NA |
| Ethics oversight | NA |

Note that full information on the approval of the study protocol must also be provided in the manuscript.

# Field-specific reporting

Please select the one below that is the best fit for your research. If you are not sure, read the appropriate sections before making your selection.

☐ Life sciences ☐ Behavioural & social sciences ☒ Ecological, evolutionary & environmental sciences

For a reference copy of the document with all sections, see nature.com/documents/nr-reporting-summary-flat.pdf

# Ecological, evolutionary & environmental sciences study design

All studies must disclose on these points even when the disclosure is negative.

| | |
|---|---|
| Study description | This study is a genome-resolved analysis of the coral reef microbiome, promoting coral microbiomes as reservoirs of novel genomic and biosynthetic diversity. |
| Research sample | Three coral genera were targeted (Millepora spp., Porites spp., Pocillopora spp.) as they are widespread across the Pacific Ocean. |
| Sampling strategy | Samples were collected from the schooner Tara and by scuba diving as described in Lombard, F. et al. Open science resources from the Tara Pacific expedition across coral reef and surface ocean ecosystems. Sci. Data 10, 324 (2023). |
| Data collection | Metagenomic data were generated as described in Belser, C. et al. Integrative omics framework for characterization of coral reef ecosystems from the Tara Pacific expedition. Sci. Data 10, 326 (2023). Furthermore, we searched the European Nucleotide Archive and included publicly available coral and sponge metagenomes. |
| Timing and spatial scale | Samples were collected across the Pacific Ocean during the Tara Pacific expedition from 2016 to 2018. |
| Data exclusions | No data generated from the Tara Pacific expedition were excluded. For the publicly available metagenomes, we excluded samples for which the metadata was insufficient to clearly identify them as sponge or coral samples. |

| | |
|---|---|
| Reproducibility | Not applicable as this is a field study. |
| Randomization | Not applicable as this is a field study. |
| Blinding | Not applicable as this is a field study. |

Did the study involve field work? ☒ Yes ☐ No

## Field work, collection and transport

| | |
|---|---|
| Field conditions | All environmental parameters are reported in Pesant, S. et al. Tara Pacific samples provenance and environmental context - version 2. (2020) doi:10.5281/ZENODO.4068292. |
| Location | 99 reefs from 32 islands across the Pacific Ocean. |
| Access & import/export | Research (UNCLOS) permits<br>Sampling permit for PANAMA under the reference 'SE/AP-18-16' delivered by the Direccion de Areas Protegidas y Vida Silvestre - LIC. Samuel Valdez Diaz Director - Ministerio de Ambiente – Republica de Panama on the 13/06/2016; Sampling permit for PANAMA under the reference '2016-0701-2019-2' delivered by the Smithsonian Tropical Research Institute Instituto Smithsonian de Investigaciones Tropicales - STRI Animal Care and Use Committee (ACUC) on the 28/06/2016; Sampling permit for PANAMA under the reference '2016-0701-2019-2-A1' delivered by the Smithsonian Tropical Research Institute Instituto Smithsonian de Investigaciones Tropicales - STRI Animal Care and Use Committee (ACUC) on the 21/06/2018; Sampling permit for COLOMBIA under the reference 'N°009' delivered by the MINISTERIO DE AMBIENTE Y DESARROLLO SOSTENIBLEPARQUES NACIONALES NATURALES DE COLOMBIA on the 04/03/2016; Sampling permit for CHILE under the reference '13270/24/457/Vrs' delivered by the Servicio Hidrografico y Oceanografico de la Armada de Chile (SHOA) – Patricio Carrasco Hellwig Contraalmirante Director on the 29/08/2016; Sampling permit for UNITED-KINGDOM (PITCAIRN ISLANDS) under the reference 'N/A' delivered by the Government of Pitcairn islands /Environmental, Conservation & Natural Resources Division Manager // Christian Michele on the 25/02/2016; Sampling permit for COOK under the reference '11-16' delivered by the Foundation for National Research – Cook Island Research Committee – Office of the Prime Minister – Elizabeth Wright-Koteka (Chairperson) on the 12/09/2016; Sampling permit for NIUE under the reference '34/16' delivered by the Government of Niue – Office for External Affairs on the 17/11/2016; Sampling permit for SAMOA under the reference 'Memorandum of Agreement' delivered by the THE GOVERNMENT OF THE INDEPENDENT STATE OF SAMOA acting by and through the Ministry of Natural Resources and Environment on the 29/11/2016; Sampling permit for WALLIS AND FUTUNA under the reference 'Arrêté n°2016-527' delivered by the Le Préfet, Administrateur supérieur des îles Wallis et Futuna on the 24/11/2016; Sampling permit for TUVALU under the reference 'MFAT : 449/16' delivered by the Government of Tuvalu – Ministry of Foreign Affairs on the 19/12/2016; Sampling permit for KIRIBATI under the reference '015/16' delivered by the Environment and Conservation Division – Republic of Kiribati on the 24/11/2016; Sampling permit for MICRONESIA under the reference 'Letter' delivered by the Deputy Assistant Secretary – Marine Resources Unit – Department of Resources and Development – Federated States of Micronesia on the 05/04/2017; Sampling permit for GUAM under the reference 'U2021-023' delivered by the Marine Scientific Research Coordinator Office of Ocean and Polar Affairs – United States Department of State Bureau of Oceans and International Environmental and Scientific Affairs on the 27/10/2021; Sampling permit for AMERICAN SAMOA under the reference 'U2021-022' delivered by the Marine Scientific Research Coordinator Office of Ocean and Polar Affairs – United States Department of State Bureau of Oceans and International Environmental and Scientific Affairs on the 27/10/2021; Sampling permit for JAPAN (Tokyo Prefecture; Ogasawara Island) under the reference '28-50' delivered by the Prefecture of Tokyo on the 01/23/2017; Sampling permit for JAPAN (Okinawa Prefecture; Sesoko Island) under the reference '28-74' delivered by the Prefecture of Okinawa on the 04/14/2017; Sampling permit for JAPAN (Japanese EEZ) under the reference 'N/A' delivered by the Ministry of Agriculture, Forestry and Fisheries on the 01/10/2017; Sampling permit for FIJI under the reference '456/2017' delivered by the Ministry of Foreign Affairs – Republic of Fiji on the 11/06/2017; Sampling permit for AUSTRALIA under the reference 'G17/39873.1' delivered by the Great Barrier Reef Marine Park Authority and Department of Foreign Affairs and Trade on the 30/08/2017; Sampling permit for NEW-CALEDONIA (SOUTH PROVINCIA) under the reference 'Arrêté n°2720-2017/ARR/DENV modifiant l'arrêté 1515-2017/ARR/DENV du 04 août 2017' delivered by the Président de l'Assemblée de la Province Sud de la Nouvelle-Calédonie on the 06/09/2017; Sampling permit for NEW-CALEDONIA (CHESTERFIELD) under the reference 'Arrêté n°2017-2069/GNC' delivered by the Haut-Commissariat de la République en Nouvelle-Calédonie – Gouvernement de Nouvelle-Calédonie – République Française on the 29/08/2017; Sampling permit for SOLOMON ISLANDS under the reference 'Form 01' delivered by the Solomon Islands Maritime Safety Administration on the 20/09/2017; Sampling permit for PAPUA NEW-GUINEA under the reference '907/2017 (diplomatic clearance n°0232)' delivered by the Department of Foreign Affairs and Trade of the Independent State of Papua New Guinea on the 27/10/2017; Sampling permit for PALAU under the reference 'RE-18-04' delivered by the Ministry of Natural Resources, Environment and Tourism – Republic of Palau on the 21/12/2017; Sampling permit for CHINA (HONG-KONG) under the reference 'CMO-N00811' delivered by the Marine Department, Hong-Kong, China on the 15/03/2018; Sampling permit for TAIWAN (Pingtung county) under the reference '10707821600' delivered by the Pingtung Agri-Fish; National Taiwan Ocean University on the 06/04/2018; Sampling permit for TAIWAN (Taitung county) under the reference '1070033041' delivered by the Taitung Agri-Fish; National Taiwan Ocean University on the 12/02/2018; Sampling permit for USA (HAWAII) under the reference 'U2018-010' delivered by the United States Department of State Bureau of Oceans and International Environmental and Scientific Affairs on the 06/06/2018; Sampling permit for MEXICO under the reference 'PPF/DGOPA-291/17' delivered by the Secrataria de Agricultura, Ganaderia, Desarrollo rural, pesca y alimentacion – Comision Nacional de Acuacultura y Pesca – Direccion General de Ordenamiento Pesquero y Acuicola – Estados Unidos Mexicanos on the 28/08/2018; Sampling permit for CLIPPERTON under the reference 'HC/1195/CAB' delivered by the Haut-Commissariat de la République Polynésie Française on the 13/06/2018; Sampling permit for COSTA RICA under the reference M-C-SINAC-PNI-SE-002-2022 delivered by the Sistema Nacional de Áreas de Conservación (SINAC) on the 29/09/2022; Sampling permit for USA (MAINLAND) under the reference 'U2018-010' delivered by the United States Department of State Bureau of Oceans and International Environmental and Scientific Affairs on the 06/06/2018; Sampling permit for CANADA under the reference 'Letter of regularization' delivered by the Sécurité et relations de défense (IGR)/Affaires mondiales Canada on the 05/01/2022; Sampling permit for NEW-ZEALAND under the reference 'Letter of regularization' delivered by the Ministry of Foreign Affairs and Trade on the |

05/10/2021; Sampling permit for IRELAND under the reference '572/22' delivered by the Department of Foreign Affairs on the 09/06/2022.

CITES permits

CITES export permit for PANAMA (I01) under the reference 'SEX/A-72-16' delivered by the Autoridad Nacional del Ambiente (ANAM) de la República de Panamá – Autoridad Administrativa CITES on the 28/07/2016; CITES final import permit under the reference 'FR1609100066-I' delivered the 04/08/2016 by the DRIEE ILE-DE-FRANCE; CITES export permit for PANAMA (I02) under the reference 'SEX/A-72-16' delivered by the Autoridad Nacional del Ambiente (ANAM) de la República de Panamá – Autoridad Administrativa CITES on the 28/07/2016; CITES final import permit under the reference 'FR1609100066-I' delivered the 04/08/2016 by the DRIEE ILE-DE-FRANCE; CITES export permit for PANAMA (I31) under the reference 'SEX/APO-1-2018' delivered by the Ministerio de Ambiente on the 30/08/2018; CITES final import permit under the reference 'FR1807523129-I' delivered the 19/10/2018 by the DRIEE ILE-DE-FRANCE; CITES export permit for PANAMA (I32) under the reference 'SEX/APO-1-2018' delivered by the Ministerio de Ambiente on the 30/08/2018; CITES final import permit under the reference 'FR1807523129-I' delivered the 19/10/2018 by the DRIEE ILE-DE-FRANCE; CITES export permit for COLOMBIA (I03) under the reference '41499' delivered by the Ministerio de Ambiente y Desarrollo Sostenible de la República de Colombia on the 13/02/2017; CITES final import permit under the reference 'FR1707506158-I' delivered the 17/03/2017 by the DRIEE ILE-DE-FRANCE; CITES export permit for CHILE (I04) under the reference '16CL000007WS' delivered by the Servicio Nacional de Pesca y Acuicultura on the 02/09/2016; CITES final import permit under the reference 'FR1607525599-I' delivered the 03/11/2016 by the DRIEE ILE-DE-FRANCE; CITES export permit for UNITED-KINGDOM (PITCAIRN ISLANDS; I05) under the reference 'FR1698700198-E' delivered by the Haut-Commissariat de la République en Polynésie Française on the 03/11/2016; CITES final import permit under the reference 'FR1607525646-I' delivered the 04/11/2016 by the DRIEE ILE-DE-FRANCE; CITES export permit for FRENCH POLYNESIA (GAMBIER – TUAMOTU; I06) under the reference 'FR1698700198-E' delivered by the Haut-Commissariat de la République en Polynésie Française on the 03/11/2016; CITES final import permit under the reference 'FR1607525646-I' delivered the 04/11/2016 by the DRIEE ILE-DE-FRANCE; CITES export permit for MOOREA (I07) under the reference 'FR1698700218-E' delivered by the Haut-Commissariat de la République en Polynésie Française on the 21/11/2016; CITES final import permit under the reference 'FR1707503441-I ' delivered the 07/02/2017 by the DRIEE ILE-DE-FRANCE; CITES export permit for COOK (I08) under the reference 'CK/2016 – 14278' delivered by the Tu'anga Taporoporo national environment service of the Cook Islands on the 17/11/2016; CITES final import permit under the reference 'FR1707503442-I ' delivered the 07/02/2017 by the DRIEE ILE-DE-FRANCE; CITES export permit for NIUE (I09) under the reference 'N/A' delivered by the N/A on the N/A; CITES final import permit under the reference 'FR1707511900-I' delivered the 11/06/2017 by the DRIEE ILE-DE-FRANCE; CITES export permit for SAMOA (I10) under the reference 'SAMC16012' delivered by the Ministry of Natural Resources and Environment (MNRE) of the Government of Samoa on the 29/11/2016; CITES final import permit under the reference 'FR1707503440-I' delivered the 07/02/2017 by the DRIEE ILE-DE-FRANCE; CITES export permit for WALLIS AND FUTUNA (I11) under the reference 'WF/C/16/01' delivered by the Préfet – Administrateur Supérieur – Chef du territoire des Îles Wallis et Futuna on the 25/12/2016; CITES final import permit under the reference 'FR1707503441-I ' delivered the 07/02/2017 by the DRIEE ILE-DE-FRANCE; CITES export permit for TUVALU (I12) under the reference '1204 (Quarantine document)' delivered by the Plant Protection and Quarantine Services - Ministry of Natural Resources - TUVALU GOVERNMENT on the 03/01/2017; CITES final import permit under the reference 'FR1707511900-I' delivered the 11/06/2017 by the DRIEE ILE-DE-FRANCE; CITES export permit for KIRIBATI (I13) under the reference '015/16 (UNCLOS permit)' delivered by the Fisheries Division, Ministry of Fisheries & Marine Resources Development -- GOVERNMENT OF KIRIBATI on the 12/01/2017; CITES final import permit under the reference 'FR1707511900-I' delivered the 11/06/2017 by the DRIEE ILE-DE-FRANCE; CITES export permit for MICRONESIA (CHUUK; I14) under the reference 'CFM17-01-01' delivered by the Department of Resources and Development – Division of Resource Management and Development – Office of Marine Resources on the 19/01/2017; CITES final import permit under the reference 'FR1707511900-I' delivered the 11/06/2017 by the DRIEE ILE-DE-FRANCE; CITES export permit for GUAM (I15) under the reference '17US18844C/9' delivered by the U.S. Fish and Wildlife service – Division of management authority – Branch of permits on the 02/03/2017; CITES final import permit under the reference 'FR1707503440-I' delivered the 07/02/2017 by the DRIEE ILE-DE-FRANCE; CITES export permit for JAPAN (OGASAWARA; I16) under the reference '17JP001279/TE' delivered by the Trade and Economic Cooperation Bureau – Ministry of Economy, Trade and Industry (METI) on the 16/05/2017; CITES final import permit under the reference 'FR1707511899-I' delivered the 06/06/2017 by the DRIEE ILE-DE-FRANCE; CITES export permit for JAPAN (SESOKO; I17) under the reference '17JP001280/TE' delivered by the Trade and Economic Cooperation Bureau – Ministry of Economy, Trade and Industry (METI) on the 16/05/2017; CITES final import permit under the reference ' FR1707511898-I ' delivered the 06/06/2017 by the DRIEE ILE-DE-FRANCE; CITES export permit for FIJI (I18) under the reference 'FJ/EXP-03055' delivered by the Fisheries Department - Government of Fiji on the 08/06/2017; CITES final import permit under the reference 'FR1707521006-I' delivered the 27/09/2017 by the DRIEE ILE-DE-FRANCE; CITES export permit for AUSTRALIA (I19) under the reference 'PWS2017-AU-001613 ' delivered by the Department of the Environment and Energy of the Australian Government on the 22/08/2017; CITES final import permit under the reference 'FR1707521097-I' delivered the 29/09/2017 by the DRIEE ILE-DE-FRANCE; CITES export permit for NEW-CALEDONIA (SOUTH PROVINCIA; I21) under the reference 'FR1798800075-E' delivered by the Haut-Commissariat de la République en Nouvelle-Calédonie / DAFE on the 22/09/2017; CITES final import permit under the reference 'FR1707521096-I' delivered the 29/09/2017 by the DRIEE ILE-DE-FRANCE; CITES export permit for NEW-CALEDONIA (CHESTERFIELD; I20) under the reference 'FR1798800075-E' delivered by the Haut-Commissariat de la République en Nouvelle-Calédonie / DAFE on the 22/09/2017; CITES final import permit under the reference 'FR1707521096-I' delivered the 29/09/2017 by the DRIEE ILE-DE-FRANCE; CITES export permit for SOLOMON ISLANDS (I22) under the reference 'EX2017/188' delivered by the Ministry of Environment, Climate Change, Disaster Management and Met on the 19/10/2017; CITES final import permit under the reference 'FR1807501178-I' delivered the 15/01/2018 by the DRIEE ILE-DE-FRANCE; CITES export permit for PAPUA NEW-GUINEA (I23) under the reference '18004' delivered by the Conservation and Environment Protection Authority (CEPA) on the 06/12/2017; CITES final import permit under the reference 'FR1807508641-I' delivered the 24/04/2018 by the DRIEE ILE-DE-FRANCE; CITES export permit for PAPUA NEW-GUINEA (I24) under the reference '18004' delivered by the Conservation and Environment Protection Authority (CEPA) on the 06/12/2017; CITES final import permit under the reference 'FR1807508641-I' delivered the 24/04/2018 by the DRIEE ILE-DE-FRANCE; CITES export permit for PALAU (transect; I25) under the reference 'PW18-004' delivered by the Office of the Minister – Ministry of Natural Resources, Environment and Tourism on the 10/01/2018; CITES final import permit under the reference 'FR1807501177-I' delivered the 16/01/2018 by the DRIEE ILE-DE-FRANCE; CITES export permit for PALAU (leg; I25) under the reference 'PW18-009' delivered by the Office of the Minister – Ministry of Natural Resources, Environment and Tourism on the 01/10/2018; CITES final import permit under the reference 'FR1807512823-I' delivered the 13/06/2018 by the DRIEE ILE-DE-FRANCE; CITES export permit for CHINA (HONG-KONG; I26) under the reference 'APO/EL 3/18' delivered by the Agriculture, Fisheries and Conservation Department of Hong-Kong Special Administrative Region on the 16/04/2018; CITES final import permit under the reference 'FR1807508518-I' delivered the 23/04/2018 by the DRIEE ILE-DE-FRANCE; CITES export permit for TAIWAN (I27) under the reference 'FTS507W0147330' delivered by the Bureau of Foreign Trade – Ministry of Economic Affairs on the 21/06/2018; CITES final import permit under the reference 'FR1807515565-I' delivered the 18/07/2018 by the DRIEE

ILE-DE-FRANCE; CITES export permit for USA (HAWAII; I28) under the reference '18US97917C/9' delivered by the U.S. Fish and Wildlife service – Division of management authority – Branch of permits on the 26/07/2018. CITES final import permit under the reference 'FR180751766-I' delivered the 09/08/2018 by the DRIEE ILE-DE-FRANCE.

Disturbance: Contact with corals was minimised.

# Reporting for specific materials, systems and methods

We require information from authors about some types of materials, experimental systems and methods used in many studies. Here, indicate whether each material, system or method listed is relevant to your study. If you are not sure if a list item applies to your research, read the appropriate section before selecting a response.

## Materials & experimental systems

| n/a | Involved in the study | |
|---|---|---|
| ☒ | ☐ | Antibodies |
| ☒ | ☐ | Eukaryotic cell lines |
| ☒ | ☐ | Palaeontology and archaeology |
| ☒ | ☐ | Animals and other organisms |
| ☒ | ☐ | Clinical data |
| ☒ | ☐ | Dual use research of concern |
| ☒ | ☐ | Plants |

## Methods

| n/a | Involved in the study | |
|---|---|---|
| ☒ | ☐ | ChIP-seq |
| ☒ | ☐ | Flow cytometry |
| ☒ | ☐ | MRI-based neuroimaging |

# Plants

Seed stocks: NA

Novel plant genotypes: NA

Authentication: NA