## [Peer Review File · Nature]

Coral microbiomes as reservoirs of unknown genomic and biosynthetic diversity

Corresponding Authors: Shinichi Sunagawa, Lucas Paoli and Jörn Piel

Version 1:

Reviewer comments:

Referee #1

(Remarks to the Author)

In this manuscript, Paoli et al present a metagenomic dataset from 820 coral samples, predict metagenome assembled genomes and biosynthetic gene clusters from them, and show that these samples are “rich in biosynthetically talented” organisms. I do certainly value the new dataset and the analysis performed, but beyond that, I do not see much contribution of this study to our understanding of symbiosis, chemical defense, ecology, or even basic biology of one of the most important organisms in the Ocean. The study, as presented, is a good fit as a resource in a technically oriented journal – as all the value gained is in the deposited dataset. Otherwise, functional studies of specific bacteria, proving of real symbiosis cases, characterization of small molecule products from symbionts, detection of these molecules in the corals and localizing them, and providing evidence for their activities and contribution to the ecology of the host is all needed to make this manuscript a complete story for Nature or even Nature Microbiology. A decade ago, it was enough to publish metagenomic data as a contribution to the field. Today, with the ease of this process, much more scientific and experimental insights are needed to provide value for the data generated.

Beyond this main concern, there are a several aspects of the study that are clearly misinterpreted from the data generated. For example, average genome size can easily be an artifact of binning and MAG generation/contamination. Number of MAGs generated is not necessarily a good estimation of diversity or richness in samples, as a significant proportion of metagenomic data do not end up in MAGs. Comparison of coral data from this study to sponge data from other studies is not a good comparison, how do the authors manage batch effects of sample processing, sequencing, etc.? BGC analysis is somewhat flawed as well, have the authors looked at BGC completeness – one large BGC can be fragmented into 10, which may inflate their numbers, especially with NRPS and PKS that are extremely hard to assemble from short read data.

Referee #2

(Remarks to the Author)

This new manuscript from the Tara Oceans consortium is yet another great contribution by this team. In this particular case, the focus is on uncovering the biosynthetic potential of natural products from coral-associated microbiomes collected during the recent Indo-Pacific Ocean expedition. The authors sampled three different cnidarian genera from reef environments: scleractinian corals in the genera *Porites* and *Pocillopora*, and the hydrozoan fire coral genus *Millepora*. The sample set included coral samples from 32 different islands along the entire ocean basin covering a vast geographic range. They sequenced more than 13K metagenomes that unveiled more than 600 new microbial species and expanded the genome-wide picture of several thousand more, including about 2K metagenome-assembled genome (MAG) reconstructions. These data will open new research avenues for years to come, not only for Tara researchers but many other labs worldwide.

This study's most remarkable outcome is how these data put a magnifying lens over the current threats to microbial diversity (i.e., the invisible extinction), highlighting how critical it is to recognize the value of host-associated microbes. This analysis also demonstrates how the loss of endangered macroorganisms such as corals, will carry a much larger species diversity loss than usually considered when discussing the extinction threats of the climate crisis. This work is especially important

given the biosynthetic potential of these microbial taxa in light of the heavy hopes that are being put on possible microbially mediated processes (e.g., through natural products chemistry) to address climate change challenges. The findings and how they are presented should therefore be of great interest and significance to a broad audience.

The computational analyses are thorough and state of the art, clearly presented. The figures are informative and easy to understand. The supplementary material is sufficient. The one issue I would like to have seen addressed in a bit more detail is how to deal with host species identity in real-time rather than after the fact. With the advance in sequencing technology, it seems like doing taxonomic identification in the field would be possible through field sequencers to guide the sampling strategy. Better resolution and better biological inference can be made when the microbiomes of a single species are analyzed over such a vast biogeographic range. Adding a bit on this in the discussion for future studies would be welcome by many readers, I suspect.

The data are the result of a massive collaborative undertaking, a laudable effort that in the current political climate deserves widespread recognition for the positive outcomes of science diplomacy. Coral reef science is prominently challenging to carry out across boundaries due to collecting and export permits, often siloing local results and derailing sampling efforts. While this is not mentioned explicitly in the manuscript, I do recognize how challenging the efforts must have been to successfully achieve such an impressive multi-national sampling scheme.

Referee #3

(Remarks to the Author)

Paoli et al report a comprehensive survey of metagenomics from pan pacific coral reef. Around 1600 coral-associated microbial draft genomes were constructed based on the genome sequencing efforts. The authors showed that the microbial genomic contexts from coral reef are clearly distinct from the ones from open seawater and marine sponge. Coral-associated microbes possess considerable larger genome sizes than the ones in the open seawater, suggesting the higher gene maintenance pressures existing in the relatively nutrient-rich, complex symbiotic ecology in coral reefs. Although coral reefs have received less attention in natural product discovery compared to marine sponges, the genomic analysis suggested that the coral-associated microbiomes appear to contain richer biosynthetic potentials than the ones in marine sponges with a particular concentration of non-ribosomal peptide synthetase gene clusters. The dominant coral-associated microbes are likely to be different from the ones in marine sponges.

Strength:

1. The teams have excellent track records in metagenomic analysis/marine biology/lead compound discovery.
2. An extensive expedition of coral reef and open seawater collections resulted in the first comprehensive understanding of microbiomes in coral reefs across the world.
3. A massive metagenomic analysis gives a new insight into the coral-associated microbiome which would have significant impact on marine ecology.
4. Although natural products from coral reefs received much less attentions compared to other marine sources, information gathered here will clearly provide guidance for future natural product discovery from coral reef.
5. The climate changes clearly threaten the existence of coral reefs. Therefore, it is timely to obtain information for future marine ecology/marine preservation/lead compound discovery.

This reviewer would like to think that the information presented in this manuscript will have a high impact on these fields.

Issues:

1. Most of the manuscript contains in silico analyses. The bioinformatic techniques are relatively mature as cited. SEM images were presented. this also is the common technique. Can the authors use single cell sorting technique to isolate some microbes to confirm their IDs?
2. Recently Lima et al published an interesting paper of metagenomic analysis of coral reef (*Microb. Ecology*, 2023, 86, 392). The authors should cite this. The paper presented a detailed analysis of coral inner, coral outer and open seawater on a specific coral reef, the methodology of which were clearly different to the current manuscript. I am curious about the authors' comments.
3. The conclusion in this paper appears to be different from the one presented in this manuscript. I would encourage the authors to discuss this discrepancy.
4. Although interesting NRPS gene clusters were observed, no further analysis were presented. This reviewer would recommend the authors to considering some levels of functional analysis, i.e. characterization of functional domains of these NRPSs.
5. The arrangements of trans PKSs are interesting. Some reported PKSs were found containing PLP domains in their assembly lines. The authors should provide some levels of analysis on the comparison of characterization PLP domains with the ones presented here.
6. Any in-depth analysis of DUF domains would be encouraged. i.e. chemical or structural functions?

Referee #4

(Remarks to the Author)

This manuscript describes the analysis of microbial genomes reconstructed from 820 reef-building coral samples of three representative coral genera collected at 99 reefs across 32 islands during a two-year expedition throughout the Pacific Ocean. While it clearly describes an enormous amount of work by a very large team of talented and dedicated researchers, I struggle to see how it reports the kind of conceptual advance that merits publication in *Nature*.

For more than two decades, progress in (meta)genome sequencing has revealed numerous examples of microbes with hidden (often enormous) potential to biosynthesize structurally novel, potentially bioactive natural products. Most scientists

in relevant fields are now very familiar with the concept that we have only sampled the tip of the iceberg that microbes (both cultivated and uncultivated) have to offer in terms of novel biosynthetic potential.

The best papers in this field tackle the more relevant challenge of how to access the metabolic products of the many novel biosynthetic gene clusters uncovered by such efforts. Previous work of a broadly similar nature by some of the authors (Nature, 2022, 607, 111–118) did a better job of addressing this challenge, by identifying two metabolic products of the novel gene clusters discovered. Curiously, this has also already been done in connection with the current study, as evidenced by supplementary figure 9, which is adapted from a figure in another paper (ref 31; Chem, 2023, 9, 3696-3713).

Overall, this paper is best suited to publication in a more specialist journal, because it falls short of describing the kind of major conceptual advance expected for a paper in Nature.

Version 2:

Reviewer comments:

Referee #1

(Remarks to the Author)

I would like to thank the authors for their consideration of my previous comments and concerns. Unfortunately, though, my main concerns have not been appropriately addressed, and I still do not see this manuscript reporting a major advance or a major discovery that is of broad importance. Beyond providing a database of MAGs and BGCs – most of which do not come from corals anyways and are sponge derived from other studies' samples, despite the misleading statements in the title, abstract and throughout the manuscript – I see no transformative biology or ecology or chemistry findings coming out of this study in its current state.

I can try to be more specific below with my previous comments and how they were responded to, without being overly repetitive, and focusing only on the very major concerns:

The authors claim that coral microbiomes (which they directly extrapolate from their MAG analysis) are important for two reasons: ecologically for the coral host, and biotechnologically for drug and enzymatic discovery. None of these two claims is adequately supported in the current form of the manuscript.

Ecologically, there is not a single case of a proven new symbiont that the authors report or shed light on, with proven association with the host using imaging or other techniques, with proven integration in the host ecology by production of primary or secondary metabolites that can be protective or helpful to the host and detected in the host.

The authors respond to these major gaps by looking for genes that encode eukaryotic like proteins in the entire dataset “to support the symbiotic nature of coral-associated bacteria in our work”, performing some microbiome-host transcriptome correlation studies to “we assessed the impact of coral-associated communities on the ecology of the host”, and incubating a coral isolate with coral extract and showing that its transcription changes to “Complementing the identification of host-associated and ecologically relevant BGC-rich Acidobacteriota spp., we performed functional studies”. These are all inadequate analyses and experiments to respond to the main concerns listed above. Eukaryotic like proteins are encoded by major groups of bacteria, and without identifying specific symbiont-host pairs where these proteins actually matter, merely counting them in a big dataset is meaningless. Predicting host transcriptional dissimilarity states using microbiome data and presence of specific taxa does not mean that “they were also the most significant drivers of the transcriptomic state of coral holobionts” – it simply means that there is a correlation signal here that requires further analysis, in a sea of many potential confounding factors. Finally, bacterial transcriptional changes upon incubating a coral isolate with chemical extracts from the coral host is not at all a reflection of a specific interaction. In the most simplistic way, adequate controls were not used in this experiment (for example, extracts from other sources, unrelated corals, sponges, or even the bacterial own extract). Even if the controls were included, what does this experiment tell us without following up on the results and showing what these transcriptional changes actually mean for the symbiont-host pair? There isn't even evidence that this same bacterium is ever transcriptionally active when it is in the host itself.

Biotechnologically speaking, it is again the same story – the authors fell short of providing any convincing evidence that coral microbiomes can be a significant source of new chemicals. While I commend the efforts conducted by the biosynthetic world expert included on the team during the revision, the results are falling short of the conclusions. From a computational perspective, most MAGs and BGCs in the new dataset are derived from sponge microbiomes and not coral microbiomes, despite the big difference in the number of samples included (>1200 coral samples and only 371 sponges from other studies). From a novelty perspective, both BGCs picked for characterization belonged to a very known class of natural products, lanthipeptides, the fully mature natural products were not identified (exact cut sites are unknown) nor detected in the host coral, and the activity studies were done on peptide substrates that have been enzymatically modified and not on fully mature natural products. Yes, thiazole formation by ThaO is interesting from an enzymatic standpoint, but not enough for this claim: “we experimentally characterised the first RiPPs from Acidobacteriota, described unusual enzymology with potential relevance for drug development, and showed that coral microbiomes represent an untapped resource of bioactive and accessible natural products”.

What is really hard to grasp here is the consistent effort by the authors to compare the coral microbiome to sponges and other microbiomes. In my opinion, there is no need to do that. Corals are extremely important organisms, and even if the authors uncover 10 new true symbionts of corals and characterize only two of them showing that they produce bioactive molecules that are actually important for host biology or ecology – this would be an extremely impactful finding! Without

detailed molecular characterization of specific members of the coral microbiome and their interaction with the host as explained several times above, the current manuscript simply reports thousands of MAGs and BGCs that have no biological relevance except in a searchable database.

Referee #2

(Remarks to the Author)

The summary is well written and incorporates succinctly the new natural products chemistry used to validate the bioinformatics results.

The study is original and timely as corals have been undergoing a global bleaching event for about two years with many reefs being decimated. This study highlights the microbially-encoded functional diversity that we are rapidly losing. These results alone warrant publication in a broad reach journal as Nature. The novel biosynthetic pathways uncovered seem to be just the tip of the iceberg in the potential for reef harbored microbial diversity.

The data are unique, vast and of unmatched quality.

The analysis and conclusions are sound and the authors have made a concerted effort to address my comments and the comments of others.

Mónica Medina

Referee #3

(Remarks to the Author)

This is a revised manuscript describing in silico analysis of metagenomic dataset from coral samples, followed by extensive functional analysis of identified biosynthetic potentials. The authors answered almost all of my previous questions in relative satisfactions. The current form has a significant improvement. As stated in the previous assessment, I do value the importance of such reports in which collected datasets are uniquely positioned in various fields. Therefore the report has a good degree of originality and significance.

Methodology used in the report is relatively mature in the corresponding fields, i.e. metagenomic analysis, RIPP BGC analysis and functional verification which became a significant part of the revised manuscript.

I do need further clarifications below from the authors:

1. the holobiont-bacteria interaction study is interesting. I wonder whether stimulation by the organic matter from *Porites* tissue do necessarily prove the interactions or symbiotic relationship. Stimulation of cryptic BGCs in a given bacterium may arise from various reasons. I am not sure that expression level changes of BGCs after supplementing organic matters will reflect what the symbiosis has. Will organic matters from other invertebrates give similar stimulations? This need to be clarified.
2. I am not sure the AF predicted image in Figure 5 is helpful. Such predictions may be completely wrong in term of protein-protein interactions unless there are experimental evidence that prove to be right.
3. what does "Biodiversity loss" in the new title mean? will the authors try to say that, compared to the past, the current biodiversity in the reef-building coral community is lost due to the climate change? Is there any evidence or datasets to support such a claim?

Version 4:

Reviewer comments:

Referee #1

(Remarks to the Author)

I would like to start by thanking the authors for responding to my previous comments, and especially for explaining their reasoning when making certain claims that I did not agree with. I would also like to state clearly that I do acknowledge the great effort put into this study, from sample collection to bioinformatics to laboratory experiments - it is truly a heroic effort. Finally, I would also like to state that I have no doubt that the resulting database will be useful to researchers worldwide and that many more publications will emerge from analyzing this rich resource.

Having stated all of that, I would like to clarify that my major concerns were and are still with the biological claim made by the authors - namely that their work reveals a rich biosynthetic potential of the coral microbiome that is important for the host and the coral reef. I still do not see evidence supporting this claim in the current manuscript. The authors shifted the focus to biotechnological applications of the newly discovered enzyme, which is definitely an interesting finding on its own but far removed from the goal of the manuscript: the importance of the coral microbiome and the molecules it produces to the reef ecosystem. A similar enzyme could have been discovered from a soil or open ocean microbiome, and the biotechnological advance would have been exactly the same. Nothing here is specific to the host organism. For example, the BGC/gene they actually use for engineering in this paper (The/TheO) actually comes from this *Thermoanaerobaculia* (accession number GCA_035278895.1) that is obtained from a soil metagenome.

With all of this stated, I would like to end by saying that I do not have to be convinced for this paper to be accepted, nor should the authors try to do so. The authors have made their case already on what to include and what not to include to support their claims, and I totally respect that. In fact, there seems to be a consensus among the authors and other reviewers about the importance of this work as it stands and the level of support for the stated claims. Honestly, it does not matter that I am still not convinced that the main claim is supported and that I would have liked to see a true symbiont connected to a

detectable coral molecule connected to a bioactivity. I am truly sincere in saying that convincing me is not important, and that it should not delay the paper if the editors and other reviewers are on board. In science, we are all allowed to have different opinions in our perception of which claims are strongly supported and which are not. One person's opinion should be respected, yes, but it should not halt the decision of moving forward with an important publication like yours, reporting an unprecedented dataset and involving dozens of great scientists who worked extremely hard to bring it to fruition.

Referee #3

(Remarks to the Author)

The authors have created an extensive database of over 13,000 metagenome-assembled genomes (MAGs) and over 1,700 biosynthetic gene clusters from coral and sponge microbiomes. This resource is of broad value to the scientific community. The authors have clarified their comparison between coral and sponge microbiomes by normalizing for sampling effort. The revised analysis shows that the coral microbiome is richer in BGCs per genome and per microbial species compared to sponges, which is a key finding. The revision strengthens the manuscript by providing significant new experimental data on the unique RiPP maturase enzyme TheO from a *Thermoanaerobaculum* species, demonstrating its unique ability to form thiazoles without a lanthionine synthetase. This includes demonstrating its relaxed substrate specificity, showing that a large portion of the peptide is dispensable for recognition, and successfully installing thiazoles on an engineered hybrid peptide. This reviewer thinks that this version of revision significantly addressed most of the comments raised.

I still have two minor issues below:

The title's reference to "Biodiversity loss" was not well-supported by evidence or datasets in the manuscript itself. The authors should clarify this claim or remove it to ensure the title accurately reflects the manuscript's content.

Some of the methodology, particularly for the ecological analyses, could be more clearly summarized in the main text to improve readability, as the authors acknowledge they have moved some detailed descriptions to the supplementary information.

Point-by-point response to reviewers

Referee #1 (Remarks to the Author); Metagenomics, Marine Natural Products:

General #1

In this manuscript, Paoli et al present a metagenomic dataset from 820 coral samples, predict metagenome assembled genomes and biosynthetic gene clusters from them, and show that these samples are “rich in biosynthetically talented” organisms. I do certainly value the new dataset and the analysis performed, but beyond that, I do not see much contribution of this study to our understanding of symbiosis, chemical defense, ecology, or even basic biology of one of the most important organisms in the Ocean. The study, as presented, is a good fit as a resource in a technically oriented journal – as all the value gained is in the deposited dataset. Otherwise, functional studies of specific bacteria, proving of real symbiosis cases, characterization of small molecule products from symbionts, detection of these molecules in the corals and localizing them, and providing evidence for their activities and contribution to the ecology of the host is all needed to make this manuscript a complete story for Nature or even Nature Microbiology. A decade ago, it was enough to publish metagenomic data as a contribution to the field. Today, with the ease of this process, much more scientific and experimental insights are needed to provide value for the data generated.

We thank the reviewer for their comment and for requesting more emphasis on biological insights. To meet this request, we rose to the challenges of working with non-model organisms for molecular work and using cryopreserved samples from an expedition as a starting point. In the following, we summarise the new analytical and experimental results we included in the revised version of the manuscript.

Previous work using genome-resolved information to gain insights into the symbiotic relationship between corals and their associated bacteria focused on host-metabolism complementarity and found enrichment in genes linked to the production of fixed carbon, vitamins, and amino acids (Robbins et al., 2019, Nature Microbiology, 4, 2090–2100). Instead of studying primary metabolic complementarity to support the symbiotic nature of coral-associated bacteria in our work, we decided to test for enrichment of eukaryotic-like proteins (ELPs), as these proteins have been implicated in host infection and symbiosis establishment (Voolstra et al., 2024, Nature Reviews Microbiology, 22, 460–475). Our results showed a significantly higher density of ELPs in the proteomes of MAGs reconstructed from coral samples than those from open ocean water samples. These results are also congruent with our isolation efforts from cryopreserved samples (see details in response to reviewer 3), where we found genus-level representatives of the strains we isolated to be more prevalent and abundant in corals than in seawater samples. Circling back to ELPs, and in line with the focus of our work, we found BGC-rich lineages to be more enriched in ELPs than other previously reported coral bacterial symbionts, such as *Endozoicomonas* spp.

Furthermore, we assessed the impact of coral-associated communities on the ecology of the host by integrating data from sample-matched coral host gene expression profiles. Strikingly,

we found that novel BGC-rich Acidobacteria uncovered in our work were not only enriched in reef-building corals, but they were also the most significant drivers of the transcriptomic state of coral holobionts.

Complementing the identification of host-associated and ecologically relevant BGC-rich Acidobacteriota spp., we performed functional studies using *Sulfidibacter corallicola*, the only acidobacterial strain isolated from a reef-building coral (*Porites lutea*; same genus as targeted in our study). Testing for interactions of *S. corallicola* with coral holobionts, we found that exposure to dissolved organic matter derived from *Porites* tissue extracts led to an overall increased expression of BGCs and the production of specific metabolites despite growing equally.

Beyond the functional studies using a cultivated coral host-interacting bacterium, we characterised two ribosomally synthesised and post-translationally modified peptides (RiPPs) from two different lanthipeptide classes, which were derived from MAGs from two different acidobacterial classes (Thermoanaerobaculia and UBA6911). As such, these RiPPs represent the first natural products characterised from these classes and to our knowledge the first RiPPs from Acidobacteria spp. in general. One RiPP exhibited biotechnologically relevant human neutrophil elastase inhibitory activity in the low micromolar range. Additionally, we measured thiazole formation by an enzyme lacking homology to ATP-binding proteins, illustrating that unusual enzymology and biotechnologically accessible natural products can be found in coral reef microbiomes. Detecting these metabolites in corals and localising them would be an exciting avenue for future studies.

In summary, we believe these new analyses and experimental findings significantly enhance the biological insights presented in this manuscript and we thank the reviewer again for their constructive input.

Comment #1.1

Beyond this main concern, there are a several aspects of the study that are clearly misinterpreted from the data generated. For example, average genome size can easily be an artifact of binning and MAG generation/contamination.

We acknowledge the limitations that arise from working with MAGs, in particular that incompleteness and contamination may lead to uncertainties in predicted genome sizes. However, we would like to clarify that the methods used here include many steps to address these pitfalls, and their robustness was demonstrated in previous work (Figure R1; Paoli et al., 2022, Nature, 607, 111–118). Specifically, the binning process leverages abundance correlations of genomic fragments across metagenomic samples, a computationally expensive but key step to ensure higher quality of MAGs compared to alternative methods as benchmarked in Extended Data Fig. 2 (Paoli et al.) and discussed in Mattock and Watson (2023, Nature Methods, 20, 1170–1173). To demonstrate the transferability of robustness of genome size estimates in this work, we extended the original analysis in Paoli et al. to include additional genomes used in this study. The result shows that isolate vs MAG-based genome size estimates fall well within the same range (Figure R1). Given the robustness of our estimates ($R^2=0.83$) and the observed differences (3.6 Mbp vs 2.2 Mbp) of genome sizes

between open ocean and reef invertebrate microbial species, we are confident that this results is based on a biological signal rather than potential artefacts due to a systematic biases in the completeness or contamination of MAGs.

Figure R1. Comparing genome sizes from MAG-based predictions and isolate genomes for 84 mOTUs (species-level) clusters from the OMD and 4 mOTUs from the RMD. The mOTUs were selected to contain at least one MAG and one isolate genome. The number of RMD mOTUs is much smaller since most MAGs reconstructed in this study belong to species with no reference genomes. Genome sizes are estimated using MAGs with completeness of at least 70%, a criterion that is met for >80% of the mOTUs clusters, and corrected for incompleteness. The results show that differences between estimates of genome sizes for MAGs and isolate genome sizes from the same species are relatively small and unlikely to explain the significant differences in estimated genome sizes between the RMD and the OMD.

ACTIONS TAKEN:

- We confirmed the accuracy of genome size estimates of MAGs by comparing species-level genome clusters (mOTUs) that contain both MAGs and isolate genomes (Figure R1).
- We extended the method description (lines 529–534) to better describe genome size estimation and refer to Paoli et al. for this analysis:

Genome size estimates

We estimated genome size of microbial species using a method robust to potential MAG incompleteness or contamination²⁹. Briefly, we only retained genomes of at least 70% completeness (which represents >80% of all species in the RMD), and computed an average estimated genome size per species accounting for incompleteness by multiplying actual genome sizes by 100/completeness.

Comment #1.2

Number of MAGs generated is not necessarily a good estimation of diversity or richness in samples, as a significant proportion of metagenomic data do not end up in MAGs.

This may be a misunderstanding. By reporting the number of MAGs recovered from the different sample types we did not aim to estimate the diversity or richness of microbial species in the analysed samples.

To avoid any ambiguity, we added a sentence in the main text and also refer to results based on 16S rRNA gene sequencing to assess differences in species diversity or richness.

ACTIONS TAKEN:

- We added a sentence in lines 130–132:

Although these numbers do not necessarily reflect microbial species richness, they are congruent with the observed differences across these hosts based on 16S rRNA gene sequencing results⁹.

Comment #1.3

Comparison of coral data from this study to sponge data from other studies is not a good comparison, how do the authors manage batch effects of sample processing, sequencing, etc.?

We thank the reviewer for pointing out challenges in comparative analyses including samples from different studies and we agree that batch effects are indeed a challenging aspect in “meta-analyses”.

In fact, already in the original version, we had included supplementary figures describing how these effects explain some of the differences. For example, to justify the required normalisations we applied (by genome / species) for our comparative analyses, we showed how higher raw GCF numbers in sponges vs corals can be explained by sequencing depth (now Suppl Fig. 2e) and the number of host taxa sampled in different studies (now Suppl. Fig. 2f). However, we acknowledge that this information was not clearly presented and that the references to the supplementary figures were not well placed.

ACTIONS TAKEN:

- We draw the attention of the readers to the differences in sequencing depth, sampling effort, and geographic coverage by adding sentences in lines 222–223 and 233–235:

Original text:

Overall, we recovered fewer GCFs from corals than from sponges (2,817 vs 3,920 GCFs; Fig. 3e).

Revised text:

Overall, we recovered fewer GCFs from corals than from sponges (2,781 vs 3,920 GCFs; Fig. 3e), which is largely due to differences in effective sequencing depth (Suppl. Fig. 2e) and sampling effort (Suppl. Fig. 2f).

Original text:

Although, the RMD still contains relatively few genomes from soft corals (72 MAGs) and other coral reef species, our results highlight that the microbiome of reef-building corals, and fire corals in particular, encodes an immense and yet untapped source of novel biosynthetic enzymes and natural products (Suppl. Fig. 6–8).

Revised text:

While our results suggest that the number of GCFs detected is still increasing equally with each new host (Suppl. Fig. 2f) or island (Suppl. Fig. 2g) sampled and although the RMD contains relatively few genomes from soft corals (72 MAGs) and other coral reef species, our results highlight that the microbiome of reef-building corals, and fire corals in particular, encodes an immense and yet untapped source of novel biosynthetic enzymes and natural products.

Comment #1.4

BGC analysis is somewhat flawed as well, have the authors looked at BGC completeness – one large BGC can be fragmented into 10, which may inflate their numbers, especially with NRPS and PKS that are extremely hard to assemble from short read data.

We thank the reviewer for raising concerns regarding incomplete BGCs in metagenomic samples, which we agree should not be underestimated and are in itself challenging to estimate (e.g. the antiSMASH software defines completeness based on the presence of flanking regions of a given size up- and downstream of the BGC).

We would like to highlight that the methods used here have been extensively benchmarked in previous work (Paoli et al., 2022, Nature, 607, 111–118). Firstly, we assembled metagenomic data on a per sample basis (as opposed to co-assembling data from several samples) using the metagenomic assembler metaSPAdes, which has been shown to improve genome contiguity. Secondly BGCs are only predicted on contigs ≥ 5 kbp (a threshold we chose to include small RiPP BGCs) to reduce the impact of fragmentation. Thirdly, we compared the length and number of genes recovered from BGCs that were metagenomically predicted to MIBiG BGCs, finding that only NRPS and PKS clusters were shorter in metagenomic data, although the number of genes were higher. Fourthly, we dereplicated BGCs into GCFs so that fragments of a given BGC can be grouped with a similar, more contiguous BGC reconstructed within another sample or genome. Lastly, using antiSMASH-based completeness estimates, we found that incomplete BGCs did not have a significantly lower dereplication rate compared to complete BGCs, which results in a proportionally higher number of GCFs.

Nonetheless, we further corroborated that BGC incompleteness did not impact the biological differences found across host types. Specifically, we found that the distribution of BGC completeness per genome was similar across host types (Figure R2a) and was mainly dependent on genome fragmentation rather than host type (Figure R2b). Since genome

fragmentation was not substantially different across hosts (Figure R2c), we are confident that the differences reported in this work are not driven by BGC fragmentation.

As BGC fragmentation could inflate the number of BGCs per genome and thus the number of candidate BGC-rich species, already in our initial definition of BGC-rich species, we did not identify them based on the maximal number of BGCs across genomes of a given species, but rather by selecting a representative genome per species. For each species, we selected a representative based on quality scores (completeness - $5 \times$ contamination), contiguity (maximising N50), and the number of BGCs. Furthermore, representatives with at least 15 BGCs were manually inspected before considering them candidate BGC-rich species. However, we acknowledge that this point was not clearly communicated.

In the revised manuscript, we not only improved the clarity of our procedure, but also chose to use even more conservative estimates by maximising the number of complete BGCs while still considering the quality and N50 metrics for each species (see actions taken below).

Additionally, to validate our identification of new candidate BGC-rich species, we used leftover DNA to generate a deep (~150 Gbp) long-read metagenome (PacBio ultra-low input protocol). This additional, substantial sequencing effort allowed us to reconstruct complete (1 contig, N50 12.2 Mbp) and near complete (7 contigs, N50 5.4 Mbp) genomes for two of the candidate BGC-rich species identified using Illumina-sequenced MAGs (*Thermoanaerobaculia* sp. and *Acanthopleuribacteraceae* sp., respectively, both belonging to the phylum Acidobacteriota). This changed the ratio of the number of complete BGCs to the number of total BGCs in favour of complete ones: from 26/30 to 30/30 and from 20/38 to 36/41, respectively, confirming the trends we reported based on short-read sequencing data.

Figure R2. (a) The percentage of complete BGCs per genome is stable across host types. (b) While a number of genomes do not encode any complete BGCs, this largely improves with genome contiguity (here $N50 \geq 10$ Kbp) and the distribution remains similar across host types. (c) Genome contiguity, as captured by the N50 is similar across host types (N50s of 14.9, 13.5, and 13.1 Kbp for sponges, fire corals, and stony corals, respectively).

ACTIONS TAKEN:

- We used a more conservative approach to define BGC-rich species, reducing the number of candidate BGC-rich species to 20. Acidobacteriota remained the most BGC-rich phylum.
- We added a method section to detail the methods described above to lines 615–622:

Defining candidate BGC-rich species

To define candidate BGC-rich species, we selected a representative genome per species based on the number of complete BGCs, the quality Q-score (based on completeness - 5 × contamination), and the N50. The number of complete BGCs and the N50 were normalised per species using $y = \log(x) / \max(\log(x)) \times 100$. The three indices (each ranging from 0 to 100) were summed to form a composite index. If the highest scoring genome encoded at least 15 BGCs, we considered that species a candidate BGC-rich species. For comparative purposes, we used the same definition of BGC-rich species on the OMD MAGs.

- We generated additional sequencing data using long-read technology to validate the BGC content of two candidate BGC-rich species, which we describe in lines 635–649:

Validation of BGC-rich acidobacterial MAGs by long-read metagenomics

A HiFi library was prepared from leftover DNA from Tara Pacific sample SAMEA6034818 using the SMRTbell® Express Template Prep kit 2.0 (PacBio) for ultra low input PacBio sequencing. A 5 kbp size-selection was performed on the library using Ampure PacBio beads diluted with PacBio elution buffer to 35% (volume by volume), following manufacturer's instructions. Three runs were sequenced to generate a total of 151.6 Gbp of deduplicated PacBio HiFi reads. Using blast (v2.15.0+), we aligned the reads to the two BGC-rich acidobacterial MAGs reconstructed from the short-read metagenome of the same sample (TARA_SAMEA6034818_MAG_00000048 and TARA_SAMEA6034818_MAG_00000020). Retaining reads with a minimum length of 1 kbp that aligned to the reference with ≥95% identity across ≥200 bp, we assembled the reads with Flye (v2.9.3)¹⁵⁴ (with the options --pacbio-hifi and --scaffold) to generate new MAGs and repeated the read mapping and assembly step to generate two MAGs of 1 and 7 contigs, respectively. We annotated the long-read MAGs with Barrnap (v0.9) and antiSMASH (v6.1.1)²⁴ as described for the short-read MAGs.

Referee #2 (Remarks to the Author); Coral reef microbiology:

General #2

This new manuscript from the Tara Oceans consortium is yet another great contribution by this team. In this particular case, the focus is on uncovering the biosynthetic potential of natural products from coral-associated microbiomes collected during the recent Indo-Pacific Ocean expedition. The authors sampled three different cnidarian genera from reef environments: scleractinian corals in the genera *Porites* and *Pocillopora*, and the hydrozoan fire coral genus *Millepora*. The sample set included coral samples from 32 different islands along the entire ocean basin covering a vast geographic range. They sequenced more than 13K metagenomes that unveiled more than 600 new microbial species and expanded the genome-wide picture of several thousand more, including about 2K metagenome-assembled genome (MAG) reconstructions. These data will open new research avenues for years to come, not only for Tara researchers but many other labs worldwide.

This study's most remarkable outcome is how these data put a magnifying lens over the current threats to microbial diversity (i.e., the invisible extinction), highlighting how critical it is to recognize the value of host-associated microbes. This analysis also demonstrates how the loss of endangered macroorganisms such as corals, will carry a much larger species diversity loss than usually considered when discussing the extinction threats of the climate crisis. This work is especially important given the biosynthetic potential of these microbial taxa in light of the heavy hopes that are being put on possible microbially mediated processes (e.g., through natural products chemistry) to address climate change challenges. The findings and how they are presented should therefore be of great interest and significance to a broad audience.

We thank the reviewer for this positive assessment of our work.

Comment #2.1

The computational analyses are thorough and state of the art, clearly presented. The figures are informative and easy to understand. The supplementary material is sufficient. The one issue I would like to have seen addressed in a bit more detail is how to deal with host species identity in real-time rather than after the fact. With the advance in sequencing technology, it seems like doing taxonomic identification in the field would be possible through field sequencers to guide the sampling strategy. Better resolution and better biological inference can be made when the microbiomes of a single species are analyzed over such a vast biogeographic range. Adding a bit on this in the discussion for future studies would be welcome by many readers, I suspect.

We thank the reviewer for appreciating the difficulty in real-time coral species delineation. Due to their morphological plasticity and cryptic diversity, coral species are notoriously difficult (to virtually impossible) to identify based on morphology alone, and even genetic analyses based on traditional loci often do not allow delineation at the species level (Arrigoni et al., 2020, Coral Reefs, 39, 1001–1025). While the resolution of cryptic lineages matters to better understand response patterns (Gómez-Corrales & Prada, 2020, Molecular Ecology,

29(22), 4265–4273), their presence also highlights the need to resolve species diversity for conservation efforts. In the context of *Tara Pacific*, we chose target species, which we treated as verifiable hypotheses, and performed genome-wide analyses on a subset of the sampled fragments to identify coral genetic lineages (Voolstra et al., 2023, *npj Biodiversity*, 2, 15). To circumvent the need to deeply sequence entire genomes for which a priori knowledge is present, our colleagues from the *Tara Pacific* consortium developed “Divergent Fragment”, a technique based on the amplification and sequencing of a single diagnostic genomic fragment of less than 2kb (Deshuraid et al., 2022, [bioRxiv, doi.org/10.1101/2022.10.21.513203](https://doi.org/10.1101/2022.10.21.513203)). Leveraging Oxford Nanopore Technology, these diagnostic sequences could already be generated and analysed in the field.

ACTIONS TAKEN:

- Thanks to the constructive comment by this reviewer, we rephrased the section on sample collection to include the future potential to use diagnostic genomic fragments in lines 450–452:

Original text:

*Based on morphology, we targeted three reef-building corals: the stony corals *Pocillopora meandrina* and *Porites lobata*, and the fire coral *Millepora cf. platyphylla*. However, because corals within the same genus can be difficult to differentiate by eye, results were aggregated at the genus level (*Pocillopora*, *Porites*, and *Millepora*). Population genomic analysis of the sampled morphotypes revealed the presence of five, three, and one putative species for *P. meandrina*, *P. lobata*, and *M. cf. platyphylla*, respectively³².*

Revised text:

*Based on morphology, we targeted three reef-building corals: the stony corals *Pocillopora meandrina* and *Porites lobata* and the fire coral *Millepora cf. platyphylla*. However, because corals within the same genus can be difficult to differentiate by eye, we here aggregated the results at the genus level (*Pocillopora*, *Porites*, and *Millepora*) and performed population genomic analyses of the sampled morphotypes to confirm their identities in a subset of 11 islands³². The design and utilisation of diagnostic genomic fragments⁷⁷ will allow future studies to identify coral species in near real-time.*

Comment #2.2

The data are the result of a massive collaborative undertaking, a laudable effort that in the current political climate deserves widespread recognition for the positive outcomes of science diplomacy. Coral reef science is prominently challenging to carry out across boundaries due to collecting and export permits, often siloing local results and derailing sampling efforts. While this is not mentioned explicitly in the manuscript, I do recognize how challenging the efforts must have been to successfully achieve such an impressive multi-national sampling scheme.

We appreciate the reviewer's acknowledgement of the efforts that were undertaken to make this work feasible.

Referee #3 (Remarks to the Author); Natural products biosynthesis:

General #3

Paoli et al report a comprehensive survey of metagenomics from pan pacific coral reef. Around 1600 coral-associated microbial draft genomes were constructed based on the genome sequencing efforts. The authors showed that the microbial genomic contexts from coral reef are clearly distinct from the ones from open seawater and marine sponge. Coral-associated microbes possess considerable larger genome sizes than the ones in the open seawater, suggesting the higher gene maintenance pressures existing in the relatively nutrient-rich, complex symbiotic ecology in coral reefs. Although coral reefs have received less attention in natural product discovery compared to marine sponges, the genomic analysis suggested that the coral-associated microbiomes appear to contain richer biosynthetic potentials than the ones in marine sponges with a particular concentration of non-ribosomal peptide synthetase gene clusters. The dominant coral-associated microbes are likely to be different from the ones in marine sponges.

Strength:

1. The teams have excellent track records in metagenomic analysis/marine biology/lead compound discovery.
2. An extensive expedition of coral reef and open seawater collections resulted in the first comprehensive understanding of microbiomes in coral reefs across the world.
3. A massive metagenomic analysis gives a new insight into the coral-associated microbiome which would have significant impact on marine ecology.
4. Although natural products from coral reefs received much less attentions compared to other marine sources, information gathered here will clearly provide guidance for future natural product discovery from coral reef.
5. The climate changes clearly threaten the existence of coral reefs. Therefore, it is timely to obtain information for future marine ecology/marine preservation/lead compound discovery.

This reviewer would like to think that the information presented in this manuscript will have a high impact on these fields.

We thank the reviewer for the positive assessment of our work. We hope it will draw critically needed attention to conserving remaining reefs and restoring degraded ones, and that it will reach the widest possible audience.

Issues:

Comments #3.1

1. Most of the manuscript contains in silico analyses. The bioinformatic techniques are relatively mature as cited. SEM images were presented. this also is the common technique. Can the authors use single cell assorting technique to isolate some microbes to confirm their IDs?

We agree with the reviewer that the isolation and experimental validation of coral-associated microorganisms adds to their relevance, enabling downstream applications. In line with the focus of our manuscript, we identified two samples from *Millepora* with high loads of novel candidate BGC-rich Acidobacteria. We then traced the colonies of origin and identified cryopreserved replicate fragments collected from the same coral colonies. After applying for the necessary CITES permits and clearing customs, we performed dilution-to-extinction experiments to isolate single microbial cells from blasted coral tissue. Our efforts produced more than 2,500 isolates, which forced us to pre-screen the isolates using a PCR-based approach. To connect our isolates to microbial populations detected in the environment, we sequenced the 16S rRNA gene (Sanger sequencing) of 322 isolated organisms and the full genomes (hybrid long-read Oxford Nanopore Technology and short-read Illumina sequencing) of three of them. While the isolates matched our MAGs at the genus level, we could connect those microorganisms to environmental populations through 16S rRNA gene-based comparisons (two with 100% identity matches and one with a 99.7% identity match to the Tara Pacific 16S rRNA gene OTU dataset), supporting their prevalence/abundance across the Pacific Ocean. In addition, all the populations identified were more abundant in corals compared to surrounding seawater samples, demonstrating the isolation of coral-associated bacteria (Figure R3).

However, as no BGC-rich target strains were identified, we decided to redirect and focused instead on functionally characterising holobiont–bacteria interactions using the only acidobacterial strain isolated from a reef-building coral (*Porites lutea*), *Sulfidibacter corallicola*, and—though much more resource and time intensive as well as ambitious—aimed to biochemically characterise heterologously expressed BGCs. Our results revealed that exposure to ecologically relevant cues (dissolved organic matter from *Porites* tissue extracts) induced the transcription of otherwise silent BGCs and the production of distinct metabolites in *S. corallicola* (while its growth remained unaffected). These data have been included in the section on *BGC-rich microbial lineages in reef-building corals*.

We furthermore aimed to overcome the limitation of working with cultivated bacteria, and targeted, based on mining MAGs from previously unknown acidobacterial groups, two ribosomally synthesised and post-translationally modified peptides (RiPPs) from two different lanthipeptide classes. The biochemical characterisation of these RiPPs revealed not only a bioactive product but also an unusual maturase, demonstrating the molecular discovery potential in coral-associated acidobacteria. The results have been included as a new section on *Unusual enzymology and peptide natural products from novel Acidobacteriota spp. associated with reef-building corals*.

Figure R3. Relative abundance of the Tara Pacific 16S rRNA gene OTUs matching the 16S rRNA gene sequences of the three isolates.

Comments #3.2

2. Recently Lima et al published an interesting paper of metagenomic analysis of coral reef (Microb. Ecology, 2023, 86, 392). The authors should cite this. The paper presented a detailed analysis of coral inner, coral outer and open seawater on a specific coral reef, the methodology of which were clearly different to the current manuscript. I am curious about the authors' comments.

We thank the reviewer for bringing this work to our attention. In the suggested study, Lima et al. worked with data generated for their previous work (Lima et al., 2020, mBio, 11, mbio.02691-19) and released as BioProject PRJNA595374. We cited this original study and included their data in our analyses. However, we decided not cite the suggested study as it (i) addressed specific research questions (Are the microbial taxonomic and functional profiles in the coral surface mucus layer shaped by their local reef environment? What is their role in coral health and ecosystem functioning?) that do not overlap with ours. Furthermore, as summarised by the reviewer, Lima et al. (ii) sampled more thermally variable inner patch reefs and more stable outer reefs in Bermuda to compare two different habitat types. With our 32 islands, which spanned a wide range of environmental conditions (from temperate latitudes to the equator and from low to high biodiversity systems) and included continental islands, remote volcanic islands, and atolls, we aimed for a pan-Pacific view to capture as much diversity as possible. Finally, while Lima et al. (iii) targeted the surface mucus layer microbiome of the corals to study the interface between the coral host and the environment, we collected entire coral fragments to characterise the overall coral microbiome (see also below). Therefore, we feel that the differences in study aims, scope, and methods do not allow us to adequately contextualise and thus cite the suggested study.

Comments #3.3

3. The conclusion in this paper appears to be different from the one presented in this manuscript. I would encourage the authors to discuss this discrepancy.

We appreciate the reviewer's comment. Lima et al. conducted a study to characterise and compare the composition and functional potential of the coral surface mucus layer

microbiome in fluctuating and stable environments, aiming to identify potential sources for a beneficial microbiome for coral restoration programs. Their findings revealed specific taxa and functions associated with the coral mucus microbiome unique to each reef zone, and they also confirmed that the coral-associated microbiome is distinct from that of the water column, irrespective of the reef zone. However, their Principal Coordinate Analysis showed that reef zone explained more variance (38.1%) in microbial communities than host vs environment (27.5%).

We believe the reviewer's comment refers to this apparent discrepancy with our emphasis on host-specificity. It is important to note that Lima et al. sampled the interface between the coral host and the water column, which is heavily influenced by the latter (Aprill, Weber & Santoro, 2016, *mSystems*, 1, e00143-16). To explore this in more detail, we modified Suppl. Fig. 1d (now) for Figure R4: We used a third colour (orange) for samples reported as coral mucus in the European Nucleotide Archive biosamples database and differentiated all Lima et al. samples by symbols (triangles). We found that amongst all coral samples, the mucus microbiome was most similar to the water samples, however, despite including coral mucus, we were able to identify two distinct clusters of microbiomes.

Figure R4. Jaccard distance-based Principal Coordinate Analysis of the microbial species detected in coral and seawater metagenomes from publicly available coral studies with mucus (orange) and Lima et al. (triangles) samples highlighted compared to Suppl. Fig. 1d (now).

ACTIONS TAKEN:

- We changed the figure caption of Suppl. Fig. 1d (now) as follows:

Original text:

A clear separation between the microbiomes of corals and seawater is found based on a Jaccard distance-based Principal Coordinate Analysis of the microbial species detected in coral and seawater metagenomes from publicly available coral studies (PERMANOVA, p -value \leq 0.001, R^2 =0.17, see Methods).

Revised text:

A clear separation between the microbiomes of corals and seawater is found based on a Jaccard distance-based Principal Coordinate Analysis of the microbial species detected in coral and seawater metagenomes from publicly available coral studies (PERMANOVA, p -value \leq 0.001, R^2 =0.17, see Methods). The coral samples found within the water sample cluster originate from the surface mucus layer of corals, which is known to be influenced by the water column¹⁷⁷.

Based on these results, and as mentioned above, we believe the methodological differences do not allow for a systematic comparison and therefore a discussion about the discrepancies between the studies.

Comments #3.4

4. Although interesting NRPS gene clusters were observed, no further analysis were presented. This reviewer would recommend the authors to considering some levels of functional analysis, i.e. characterization of functional domains of these NRPSs.

We appreciate the reviewer's comment and agree with the value of functionally characterising BGCs. However, since PKS and NRPS pathways were recently characterised from Acidobacteriota spp. (Leopold-Messer et al., 2023, Chem, 9, 3696–3713), we aimed to characterise RiPP BGCs, a class of natural products that, to our knowledge, had not been characterised in this phylum (see point 3.1).

More specifically, we characterised two RiPPs from two different lanthipeptide classes, which were derived from MAGs from two different acidobacterial classes (Thermoanaerobaculia and UBA6911), for which no BGC had been described before. One of the compounds exhibited bioactivity in the low micromolar range (neutrophil elastase inhibition) and we described thiazole formation by an enzyme previously unknown in BGCs, lacking homology to ATP-binding proteins. This functional work illustrates how unusual enzymology and biotechnologically accessible natural products can be found in coral reef microbiomes.

Comments #3.5

5. The arrangements of trans PKSs are interesting. Some reported PKSs were found containing PLP domains in their assembly lines. The authors should provide some levels of analysis on the comparison of characterization PLP domains with the ones presented here.

While we decided to focus on the functional characterisation of RiPP pathways, we concur with the reviewer that PLP-domains containing BGCs are of interest. We therefore used the PFAM HMM model of PLP-dependent enzymes (PF00291) and annotated all biosynthetic

core genes of the RMD BGCs using hmmsearch. We identified 856 PLP domains across 152 GCFs, 70 of which were annotated as NRPSs or PKSs. While PLP-domains were previously reported in those biosynthetic classes (Milano et al., 2013, BMC Structural Biology, 13, 26), we also found candidate PLP-domain containing enzymes across 38 terpene GCFs and 24 RiPP GCFs. We found that 84% of PLP-domains containing GCFs were predicted to be new, compared to 64% across the RMD (Suppl. Fig. 2d) suggesting that these pathways may be of particular interest for the discovery of new compounds and enzymology.

ACTIONS TAKEN:

- We added the figure as a panel (d) to Supplementary Figure 2 to summarise those results.

New Suppl. Fig. 2d. We screened all biosynthetic core genes of the RMD BGCs for putative PLP domains using the PFAM HMM model of PLP-dependent enzymes (PF00291) and hmmsearch (HMMER v.3.4) with the option `--cut_nc`. We identified a total of 856 PLP-domains containing BGCs, spanning 152 GCFs. Those GCFs were found to have an increased novelty rate (84%) compared to the novelty of all GCFs across the RMD (64%), with variation across GCF classes.

Comments #3.6

6. Any in-depth analysis of DUF domains would be encouraged. i.e. chemical or structural functions?

In our functional characterisation of RiPP BGCs (see point 3.4), we focused on a pathway containing a candidate maturase unannotated by antiSMASH v6.1.1. We reasoned this maturase may represent new enzymology. Further bioinformatic analysis identified remote homology to a glucose–methanol–choline (GMC)-family oxidoreductase, a class of enzymes that, to our knowledge, had not previously been found in RiPP biosynthesis. Through heterologous expression, stable-isotope labelling, introduction of point mutations, ¹H-nuclear

magnetic resonance (NMR), and 2D-NMR experiments, we demonstrate that this enzyme catalyses thiazole formation in the glycine–cysteine motif of the RiPP precursor. While thiazoles usually form through adenosine triphosphate (ATP)-dependent phosphorylation catalysed by YcaO-type enzymes (Cox, Doroghazi & Mitchell, 2015, BMC Genomics, 16, 778), to the best of our knowledge, we are the first to report on thiazole formation by a GMC oxidase member lacking homology to ATP-binding enzymes.

Referee #4 (Remarks to the Author); Biology of natural products:

General #4

This manuscript describes the analysis of microbial genomes reconstructed from 820 reef-building coral samples of three representative coral genera collected at 99 reefs across 32 islands during a two-year expedition throughout the Pacific Ocean. While it clearly describes an enormous amount of work by a very large team of talented and dedicated researchers, I struggle to see how it reports the kind of conceptual advance that merits publication in Nature.

We thank the reviewer for appreciating the amount of work that is summarised in this manuscript. We think this study is a valuable contribution by systematically exploring the genome-resolved diversity and host-specificity of reef-building coral microbiomes (highlighting the molecular resources at stake), providing an interactive platform through which the genome-resolved and biosynthetic data can be explored (facilitating and enabling the study of reef microorganisms by the entire scientific community), identifying reef-building corals as harbouring microbial communities with a biosynthetic potential richer or as rich as that of marine sponges (emphasising the biotechnological value of reef-building corals), and highlighting new, biosynthetically rich microbial lineages (suggesting new biotechnological targets).

Thanks to the reviewers' comments, we revised our manuscript to better emphasise the biological insights gained by and substantiate the biotechnological potential of our work. Building on the original work and among other additions (see responses to other reviewers), we incorporated a new dataset of microbiome-matched coral host transcriptomic profiles and show that the novel BGC-rich acidobacterial groups uncovered in our study are not only enriched in reef-building corals, but (i) they represent the strongest drivers in shaping the transcriptomic state of coral holobionts.

Moving beyond computational analyses, we conducted functional studies with *Sulfidibacter corallicola*, the only acidobacterial strain isolated from a reef-building coral. We found that (ii) exposing *S. corallicola* to coral-derived dissolved organic matter triggered the expression of specific BGCs and metabolites. This finding not only sheds light on ecologically relevant molecular host–microbial interactions but also suggests that overcoming the long-standing challenge of BGC expression may be achievable by leveraging ecologically relevant induction cues. To further substantiate the impact of coral biodiversity loss, we (iii) biochemically characterised to our knowledge the first acidobacterial RiPPs, with one showing human neutrophil elastase inhibitory activity at low micromolar concentrations, as well as (iv) non-canonical thiazole formation by an enzyme lacking homology to ATP-binding proteins, together demonstrating the potential to discover unusual enzymology and biotechnologically significant natural products in coral reef microbiomes.

In the face of the climate crisis, particularly recent reports of recording the highest sea-surface temperatures in the Great Barrier Reef over the past 400 years this year and the second to fifth highest temperatures having been recorded within the past nine years (Henley et al., 2024, Nature, 632, 320–326), we believe the re-submission of our work is of

particular topical relevance to draw critically needed attention to conserving the remaining reefs and restoring degraded ones and thus safeguard the enormous resources harboured within coral reefs. In light of the topical relevance of our work and after including new computational and experimental results extending the original report, we hope that the reviewer will reconsider whether the revised manuscript warrants publication in Nature.

For more than two decades, progress in (meta)genome sequencing has revealed numerous examples of microbes with hidden (often enormous) potential to biosynthesize structurally novel, potentially bioactive natural products. Most scientists in relevant fields are now very familiar with the concept that we have only sampled the tip of the iceberg that microbes (both cultivated and uncultivated) have to offer in terms of novel biosynthetic potential.

We agree that finding novel uncultivated microbial species *per se* is no novelty. However, our systematic assessment not only reports the 3,774 (out of 4,224) microbial species lacking prior genomic information and thus biosynthetic characterisation, but also hints at the huge microbial diversity that can be found within the microbiomes of all reef-building coral species (current estimates: >750 extant species). We believe identifying these inconspicuous yet dire consequences of coral biodiversity loss warrants recognition and should reach policy makers and stakeholders to be taken into consideration for reef conservation and restoration. To bring this message across, we changed the title to “Coral microbiome mining reveals manifold implications of biodiversity loss”.

The best papers in this field tackle the more relevant challenge of how to access the metabolic products of the many novel biosynthetic gene clusters uncovered by such efforts. Previous work of a broadly similar nature by some of the authors (Nature, 2022, 607, 111–118) did a better job of addressing this challenge, by identifying two metabolic products of the novel gene clusters discovered. Curiously, this has also already been done in connection with the current study, as evidenced by supplementary figure 9, which is adapted from a figure in another paper (ref 31; Chem, 2023, 9, 3696-3713).

We thank the reviewer for the positive evaluation of our previous work. In this revised manuscript, we not only fill the gap in the availability of microbial genome data from reef-building corals (820 metagenomes from three representative coral genera throughout the Pacific Ocean) but also characterised two ribosomally synthesised and post-translationally modified peptides (RiPPs) from distinct lanthipeptide classes, both derived from metagenome-assembled genomes (MAGs) of two different acidobacterial classes (Thermoanaerobaculia and UBA6911). As stated above, these RiPPs represent the first natural products identified from these lineages, and to our knowledge, the first RiPPs reported from Acidobacteria in general. Apart from human neutrophil elastase inhibitory activity at low micromolar concentrations, thiazole formation by an enzyme lacking homology to ATP-binding proteins highlights how unique enzymatic processes and valuable natural products can be discovered within coral reef microbiomes.

Furthermore, by functionally characterising holobiont–*S. corallicola* interactions, we explore ecologically relevant molecular host–microbial interactions and provide evidence that the induction of BGCs (notoriously challenging) could be enhanced by adding ecologically relevant cues.

Overall, this paper is best suited to publication in a more specialist journal, because it falls short of describing the kind of major conceptual advance expected for a paper in Nature.

We hope the reviewer will reconsider their recommendation based on the new computational and experimental data presented in this revised manuscript as well as our arguments regarding the implication of our work and its importance for reaching the widest audience possible.

Point-by-point response to reviewers

Referee #1 (Remarks to the Author); Metagenomics, Marine Natural Products:

General #1

I would like to thank the authors for their consideration of my previous comments and concerns. Unfortunately, though, my main concerns have not been appropriately addressed, and I still do not see this manuscript reporting a major advance or a major discovery that is of broad importance. Beyond providing a database of MAGs and BGCs – most of which do not come from corals anyways and are sponge derived from other studies' samples, despite the misleading statements in the title, abstract and throughout the manuscript – I see no transformative biology or ecology or chemistry findings coming out of this study in its current state.

I can try to be more specific below with my previous comments and how they were responded to, without being overly repetitive, and focusing only on the very major concerns:

The authors claim that coral microbiomes (which they directly extrapolate from their MAG analysis) are important for two reasons: ecologically for the coral host, and biotechnologically for drug and enzymatic discovery. None of these two claims is adequately supported in the current form of the manuscript.

Ecologically, there is not a single case of a proven new symbiont that the authors report or shed light on, with proven association with the host using imaging or other techniques, with proven integration in the host ecology by production of primary or secondary metabolites that can be protective or helpful to the host and detected in the host.

The authors respond to these major gaps by looking for genes that encode eukaryotic like proteins in the entire dataset “to support the symbiotic nature of coral-associated bacteria in our work”, performing some microbiome-host transcriptome correlation studies to “we assessed the impact of coral-associated communities on the ecology of the host”, and incubating a coral isolate with coral extract and showing that its transcription changes to “Complementing the identification of host-associated and ecologically relevant BGC-rich Acidobacteriota spp., we performed functional studies”.

These are all inadequate analyses and experiments to respond to the main concerns listed above.

We thank the reviewer for their continued engagement with our manuscript.

In revising the original manuscript, we have incorporated substantial new analyses and experiments, as acknowledged by the other reviewers, which—consistent with the feedback from other reviewers—we believe directly addressed their concerns and significantly extended the scope of the initial submission.

The reviewer argues that without direct imaging, in-host metabolite detection, or mechanistic validation of microbiome-derived compounds benefiting corals, our findings lack significance. While such experiments would certainly be valuable in the long term, they are neither practical within the scope of this study nor necessary to establish the relevance of our findings. Conducting them would require fresh sample collection under extensive permitting regulations, the development of *in situ* hybridisation protocols for non-model organisms, and full biochemical characterisation of natural products—each a major endeavour on its own. Moreover, we note that even such data would not by itself demonstrate functional roles in the host, and thus would not directly resolve the reviewer's concerns. As such, we are concerned that the reviewer dismissed the new data, experiments, and biochemical work without offering specific or actionable suggestions for further improvement.

In summary, we remain of the opinion that the breadth of our work—pan-Pacific sample collection, data generation, computational analyses, and biochemical characterisation—and the novel insights presented represent important advances in understanding coral microbiomes. Beyond immediate biological and biotechnological insights, this work has broad value in highlighting the microbial capacity that may be lost alongside corals under ongoing reef decline. For these reasons, we respectfully disagree with the assessment that our study does not report advances of broad importance.

Comment #1.1

Eukaryotic like proteins are encoded by major groups of bacteria, and without identifying specific symbiont-host pairs where these proteins actually matter, merely counting them in a big dataset is meaningless.

We respectfully disagree with the reviewer's statement that "merely counting them in a big dataset is meaningless". While it is true that eukaryotic-like proteins (ELPs) are encoded by various bacterial groups, our study goes beyond simply counting ELPs by directly comparing host-associated (RMD) and free-living (OMD) microbiomes. The marked enrichment of ELPs, a hallmark of host-associated organisms, in the RMD, and specifically in *Acidobacteriota* spp., provides compelling genomic evidence of close host association. However, we acknowledge that these results are not functional evidence of an interaction and we more clearly communicate the description of ELPs in the main text.

ACTIONS TAKEN:

- We phrased the comparison between the OMD and RMD more clearly (lines 191–194):

Supporting the host-associated nature of these organisms, compared to the OMD, we identified a significant enrichment of eukaryotic-like proteins (ELPs), which have been implicated in host infection and symbiosis establishment^{46,47}, in the RMD (Suppl. Fig. 2b).

- Due to the shifted focus of our manuscript, we de-emphasised the ELP enrichment in *Acidobacteriota* spp. and other BGC-rich lineages in the main text (lines 277–280):

Acidobacteriota lineages, including those from which we recovered BGC-rich genomes, were abundant and prevalent across the three coral genera (Fig. 4c), potentially influencing coral hosts, as indicated by their [...] enrichment of ELPs (Suppl. Information; Suppl. Fig. 2b), [...]

and moved the detailed description to the Supplementary Information (lines 916–919):

Screening for ELPs in host-associated microbial species

We found BGC-rich microbial species in general, and Acidobacteriota spp. in particular, to be more enriched in ELPs (Suppl. Fig. 2b) than other previously reported coral symbionts such as Endozoicomonas spp.^{166,167}.

Comment #1.2

Predicting host transcriptional dissimilarity states using microbiome data and presence of specific taxa does not mean that “they were also the most significant drivers of the transcriptomic state of coral holobionts” – it simply means that there is a correlation signal here that requires further analysis, in a sea of many potential confounding factors.

We would like to clarify that the reviewer’s statement does not accurately reflect our analysis or conclusions. As described in the manuscript, the generalised dissimilarity model employed incorporates a broad set of explanatory variables, including microbiome composition, symbiome, environmental parameters, host genetics, and biomarker data. Thus, the analysis already controlled for many sources of variability, making it a particularly rigorous method for disentangling complex holobiont relationships. All of these details were presented in the original version of the manuscript.

Reflecting the revised focus of the manuscript, we have retained only a brief, reworded statement in the main text and moved the detailed result and method descriptions to the Supplementary Information.

ACTIONS TAKEN:

- We reworded the reference to the GDM results in the main text (lines 277–280):

Acidobacteriota lineages, including those from which we recovered BGC-rich genomes, were abundant and prevalent across the three coral genera (Fig. 4c), potentially influencing coral hosts, as indicated by their impact on *Pocillopora* gene expression (Suppl. Information; Suppl. Fig. 3bc; Suppl. Table 8), [...]

- We described methodology and results in one cohesive section in the Supplementary Information to avoid further misinterpretation (lines 884–915):

Coral host transcriptomics and generalised dissimilarity models

We leveraged previously sequenced transcriptomic samples from the coral host *Pocillopora*¹⁶¹ to explore host transcriptomic responses to microbiome community composition (16S amplicons)¹⁵¹ along with algal symbiont community composition (ITS2 amplicons)¹⁶¹, environmental (pH, temperature, oxygen concentration, and nutrients)⁷⁵, host phylogenetic distance¹⁶², and host biomarker data¹⁶³. Distances based on normalised transcript-per-million were used as the response variable in a Generalised Dissimilarity Model (GDM)¹⁶⁴.

Firstly, several predictor variables were pre-processed to reduce dimensionality. Community compositions were rarefied using *rtk* (v0.93.2) to 20,000 reads for 16S amplicons and 4,000 reads for ITS2 amplicons. 16S gene and ITS2 sequences were subsequently aligned against the SILVA SSU reference tree (NR99, release 138) using *SINA* (v1.6.0). The alignment was processed using *MOTHUR* v1.41.0 (*filter.seqs*) and used to build a phylogenetic tree with *FastTree* v2.1.11 (options -gtr -nt -fastest -mlnni 4). The rarefied abundances (after excluding ASVs with total abundances below 20 and 4 for microbiome and symbiome data, respectively) together with the phylogenies were then used to reduce the dimensionality of both datasets by grouping 16S and ITS2 ASVs into 116 bins with *PhyCA*¹⁶⁵. Environmental and biomarker dimensions were reduced by selecting the columns listed in Supplementary Table 8.

We subsequently computed the model using the R package *gdm* v1.5¹⁶⁴, with host transcriptomic distances as the response variable and the other variables (after range normalisation) as predictors, across the 28 samples for which all data types were available (Suppl. Table 8). In addition, a simpler GDM was built using host transcriptomic distances as the response variable and all *Acidobacteriota* ASVs with total rarefied abundance greater than five across the 57 samples where both data types were available (Suppl. Table 8). Predictor importance was computed with the *gmd.varImp* function using 100 permutations.

We found the most important predictor to be a microbiome cluster exclusively containing *Acidobacteriota* spp. (Suppl. Fig. 3b; Suppl. Table 8). Furthermore, when assessing individual abundances of *Acidobacteriota* spp. (the most BGC-rich bacteria identified in this study; Fig. 4a) for their predictability of host transcriptomic dissimilarities, several candidate superproducer genera and candidate superproducer-containing lineages stood out as the explanatory features (Suppl. Fig. 3c).

Comment #1.3

Finally, bacterial transcriptional changes upon incubating a coral isolate with chemical extracts from the coral host is not at all a reflection of a specific interaction. In the most simplistic way, adequate controls were not used in this experiment (for example, extracts from other sources, unrelated corals, sponges, or even the bacterial own extract). Even if the controls were included, what does this experiment tell us without following up on the results and showing what these transcriptional changes actually mean for the symbiont-host pair?

There isn't even evidence that this same bacterium is ever transcriptionally active when it is in the host itself.

We appreciate the reviewer's comment and acknowledge the limitations of our experiment. While our focus was on the response to coral-derived organic matter, given that the coral *Porites lutea* is the source of our acidobacterial strain, we cannot exclude the possibility that similar responses might occur with organic matter from other invertebrates, which were unfortunately not available for this study. We have now more explicitly acknowledged this limitation in the revised text. Regarding the concern about the relevance of these transcriptional changes, we agree that future studies should address the functional implications of these changes within the host environment. However, we see this initial experiment as an intriguing starting point for future studies on microorganism–coral interactions (once robust model systems have been established). Additionally, our findings suggest that host-derived compounds may play a key role in BGC induction, opening exciting possibilities for future studies to harness these interactions for biotechnological applications and a deeper understanding of coral holobiont dynamics.

ACTIONS TAKEN:

- We moved the description of this experiment to Supplementary Information and added a disclaiming sentence on other organic matter sources (lines 1,024–1,025):

Whether similar BGC transcription and expression changes occur upon exposure to organic matter from other sources is currently unknown.

- We reworded the results in the main text and stated the need for further validation (lines 277–285):

Acidobacteriota lineages, including those from which we recovered BGC-rich genomes, were abundant and prevalent across the three coral genera (Fig. 4c), potentially influencing coral hosts, as indicated by their [...] transcriptional and metabolic responsiveness to coral-derived compounds (Suppl. Information; Suppl. Fig. 4). Although the nature and specificity of this relationship remain to be further explored, our findings demonstrate that reef-building corals host biosynthetically rich bacterial lineages, including Acidobacteriota spp., that emerge as new biotechnological targets³¹ given their untapped yet promising potential to yield new natural products.

Comment #1.4

Biotechnologically speaking, it is again the same story – the authors fell short of providing any convincing evidence that coral microbiomes can be a significant source of new chemicals. While I commend the efforts conducted by the biosynthetic world expert included on the team during the revision, the results are falling short of the conclusions. From a computational perspective, most MAGs and BGCs in the new dataset are derived from sponge microbiomes and not coral microbiomes, despite the big difference in the number of samples included (>1200 coral samples and only 371 sponges from other studies). From a

novelty perspective, both BGCs picked for characterization belonged to a very known class of natural products, lanthipeptides, the fully mature natural products were not identified (exact cut sites are unknown) nor detected in the host coral, and the activity studies were done on peptide substrates that have been enzymatically modified and not on fully mature natural products. Yes, thiazole formation by ThaO is interesting from an enzymatic standpoint, but not enough for this claim: “we experimentally characterised the first RiPPs from Acidobacteriota, described unusual enzymology with potential relevance for drug development, and showed that coral microbiomes represent an untapped resource of bioactive and accessible natural products”.

We appreciate the reviewer’s feedback and the opportunity to clarify our conclusions. While we acknowledge the reviewer’s concerns, we respectfully disagree with the assertion that our findings do not support our claims.

From a computational perspective, the reviewer states that most MAGs and BGCs in our dataset originate from sponge rather than coral microbiomes, despite the larger number of coral samples. However, we have not claimed otherwise. Rather, our study specifically highlights the biosynthetic potential of coral-associated microbiomes relative to sampling effort, which takes into account the sequencing depth, the number of host genera sampled, and the geographic range covered (as detailed in Methods). In the revised version, we define this more clearly in the main text.

Regarding novelty, the reviewer argues that the BGCs we characterised belong to a well-known class of natural products (lanthipeptides) and that we did not identify fully mature natural products in the host coral. However, our claim is not that we discovered a new class of natural products, but that we provide the first experimental characterisation of enzymes from Acidobacteriota involved in RiPP biosynthesis. Furthermore, our work uncovers enzymatic features with potential relevance for drug development—a statement directly supported by our findings on ThaO and its role in thiazole formation.

Following suggestions by the editors and Reviewer 3 to build on our findings and strengthen the value of our work, we expanded our characterisation of RiPP-associated GMC-family oxidoreductases and described a lanthionine synthetase-independent homologue from an unclassified Thermoanaerobaculia species. We further explored the potential of this cluster by modifying, truncating, and cargo-fusing TheA. Point mutations revealed critical residues for activity, and truncation experiments showed that much of the peptide is dispensable for recognition by TheO. Finally, we demonstrated that TheO can install thiazoles on engineered hybrid peptide substrates, highlighting its potential utility for peptide and protein engineering.

ACTIONS TAKEN:

- We defined sampling effort more clearly in the main text (lines 227–233):

Overall, we recovered fewer GCFs from corals than from sponges (2,781 vs 3,920 GCFs; Fig. 3e), primarily due to differences in effective sequencing depth (Suppl. Fig. 2e) and host genera sampled (Suppl. Fig. 2f). However, once we normalised by sampling effort (including sequencing depth and number of host genera sampled; Methods), the coral microbiome was richer in GCFs both per genome (1.4 vs 0.3 GCFs/genome) and per microbial species (2.9 vs 1.2 GCFs/species).

- We extended the biotechnological description in the main text (lines 330–352):

Prompted by these prospects, we further explored the diversity of RiPP-associated GMC-family oxidoreductases with the aim to find a lanthionine synthetase-independent homologue. We identified a promising candidate (Methods) in an unclassified Thermoanaerobaculia species (MAG, GenBank accession: GCA_035278895.1), which we termed the ‘the’ cluster. Heterologous expression of this cluster and HPLC-MS/MS analysis revealed that indeed, TheO efficiently modified TheA without requiring co-production of the associated lanthionine synthetase TheM. The resulting mass shift (–20 Da) in TheA near the GGC motif is consistent with thiazole formation (Fig. 5d; Suppl. Fig. 9).

To further explore this cluster’s biotechnological potential, we modified, shortened, and cargo-fused TheA. Firstly, we introduced glycine-to-alanine point mutations and found that a substitution of either glycine adjacent to the cysteine in TheA impaired enzyme activity, whereas the glycine preceding the thiazole could be replaced by alanine without loss of activity (Suppl. Fig. 9). Secondly, TheA variants with truncated cores of 20, 14, and 8 amino acids (but not a 2-amino acid variant) were thiazole-modified when co-produced with TheO (Fig. 5d; Suppl. Fig. 9), indicating that much of the peptide is dispensable for recognition by TheO. Thirdly, encouraged by this relaxed substrate specificity, we explored enzymatic thiazole installation N-terminal to proteins of interest. Because thiazoles restrict peptide backbone conformation and can enhance activity, stability, and other physicochemical properties of peptide-mimetic therapeutics^{67,69}, we fused porcine Protegrin-1⁷⁰—an 18-residue β -sheet antimicrobial peptide—to the C-terminus of the smallest modifiable core peptide. We detected thiazole formation in the fusion protein, demonstrating that TheO can install thiazoles on engineered hybrid peptide substrates. Future work will explore truncating the N-terminal sequence and testing different cargo proteins.

Comment #1.5

What is really hard to grasp here is the consistent effort by the authors to compare the coral microbiome to sponges and other microbiomes. In my opinion, there is no need to do that. Corals are extremely important organisms, and even if the authors uncover 10 new true symbionts of corals and characterize only two of them showing that they produce bioactive molecules that are actually important for host biology or ecology – this would be an extremely impactful finding! Without detailed molecular characterization of specific members of the coral microbiome and their interaction with the host as explained several times above, the current manuscript simply reports thousands of MAGs and BGCs that have no biological relevance except in a searchable database.

Given the significant attention sponges and soft corals have received in microbiome research and natural product discovery, we believe it is essential to place the reef-building coral microbiome into this broader comparative framework.

We respectfully disagree with the reviewer's assertion that our study "simply reports thousands of MAGs and BGCs that have no biological relevance except in a searchable database". While our work indeed provides a large-scale genomic and biosynthetic resource, it goes well beyond cataloguing data. Specifically, our work is grounded in a unique global resource derived from a 100,000 km ocean expedition encompassing more than 3,000 scuba dives and producing 820 deeply sequenced coral metagenomes. These data are complemented by site-matched seawater metagenomes, sample-matched 16S ASVs, and whole-organism transcriptomes. Building on this foundation, we identified new enzymology, heterologously expressed bioactive compounds from bacteria detected exclusively in the sampled corals, and in this newly revised manuscript even extended our analyses to the in-depth characterisation of a biotechnologically advantageous homologue within the same RiPP maturase enzyme family.

Referee #2 (Remarks to the Author); Coral reef microbiology:

General #2

The summary is well written and incorporates succinctly the new natural products chemistry used to validate the bioinformatics results.

The study is original and timely as corals have been undergoing a global bleaching event for about two years with many reefs being decimated. This study highlights the microbially-encoded functional diversity that we are rapidly losing. These results alone warrant publication in a broad reach journal as Nature. The novel biosynthetic pathways uncovered seem to be just the tip of the iceberg in the potential for reef harbored microbial diversity.

The data are unique, vast and of unmatched quality.

The analysis and conclusions are sound and the authors have made a concerted effort to address my comments and the comments of others.

We thank the reviewer for this positive assessment of our work.

Referee #3 (Remarks to the Author); Natural products biosynthesis:

General #3

This is a revised manuscript describing in silico analysis of metagenomic dataset from coral samples, followed by extensive functional analysis of identified biosynthetic potentials. The authors answered almost all of my previous questions in relative satisfactions. The current form has a significant improvement. As stated in the previous assessment, I do value the importance of such reports in which collected datasets are uniquely positioned in various fields. Therefore the report has a good degree of originality and significance.

Methodology used in the report is relatively mature in the corresponding fields, i.e. metagenomic analysis, RIPP BGC analysis and functional verification which became a significant part of the revised manuscript.

We thank the reviewer for the constructive and supportive feedback on our research, and in particular for the additional discussion with the editor suggesting a stronger focus on the biotechnological potential to make our work acceptable for publication. In response, we are pleased to present new results and findings that reinforce this aspect.

I do need further clarifications below from the authors:

Comment #3.1

The holobiont-bacteria interaction study is interesting. I wonder whether stimulation by the organic matter from *Porites* tissue do necessarily prove the interactions or symbiotic relationship. Stimulation of cryptic BGCs in a given bacterium may arise from various reasons. I am not sure that expression level changes of BGCs after supplementing organic matters will reflect what the symbiosis has. Will organic matters from other invertebrates give similar stimulations? This need to be clarified.

We appreciate the reviewer's comment and acknowledge the limitations of our experiment's scope. While we focused on the response to coral-derived organic matter, we cannot exclude the possibility that similar responses might occur with organic matter from other invertebrates, which were unfortunately not available for this study. We have clarified this limitation in the revised text and moved the experiment to the supplement. Nonetheless, we view this as an intriguing starting point for future investigations into microorganism–coral interactions once robust model systems are established. Moreover, our results suggest that host-derived compounds may play a role in BGC induction, opening opportunities both for biotechnological applications and for advancing our understanding of coral holobiont dynamics.

ACTIONS TAKEN:

- As a result of shifting the focus, we moved this experiment to Supplementary Information, and added a disclaiming sentence on other organic matter sources (lines 1,024–1,025):

Whether similar BGC transcription and expression changes occur upon exposure to organic matter from other sources is currently unknown.

Comment #3.2

I am not sure the AF predicted image in Figure 5 is helpful. Such predictions may be completely wrong in term of protein-protein interactions unless there are experimental evidence that prove to be right.

We thank the reviewer for this comment. We acknowledge that predictions for shorter proteins carry a higher degree of uncertainty and have therefore removed the AF predicted images from Figure 5.

Comment #3.3

What does "Biodiversity loss" in the new title mean? will the authors try to say that, compared to the past, the current biodiversity in the reef-building coral community is lost due to the climate change? Is there any evidence or datasets to support such a claim?

We appreciate the reviewer's concern regarding the term "biodiversity loss" in the title. Our use of this term was based on extensive evidence documenting the decline of coral reef biodiversity due to climate change, habitat degradation, and other anthropogenic stressors (e.g. Henley et al., 2024, *Nature*, 632, 320–326; Hoegh-Guldberg et al., 2019, *Science*, 365, eaaw6974; IPCC Global Warming of 1.5 °C, 2022; Logan et al., 2021, *Nat. Clim. Change*, 537–542). While our study does not directly quantify biodiversity loss in reef-building corals, it highlights the molecular resources at risk of loss under ongoing reef decline, with increasing numbers of coral species being threatened or endangered.

We recognise, however, that the phrasing created a disconnect between our data and the broader phenomenon of biodiversity loss, which may set misleading expectations—as reflected in the reviewer's comment. To avoid ambiguity, we have reverted to a title closer to the original.

ACTIONS TAKEN:

- We revised the title:

Coral microbiomes as reservoirs of unique genomic and biosynthetic diversity

Point-by-point response to reviewers

Referee #1 (Remarks to the Author); Metagenomics, Marine Natural Products:

I would like to start by thanking the authors for responding to my previous comments, and especially for explaining their reasoning when making certain claims that I did not agree with. I would also like to state clearly that I do acknowledge the great effort put into this study, from sample collection to bioinformatics to laboratory experiments - it is truly a heroic effort. Finally, I would also like to state that I have no doubt that the resulting database will be useful to researchers worldwide and that many more publications will emerge from analyzing this rich resource.

Having stated all of that, I would like to clarify that my major concerns were and are still with the biological claim made by the authors - namely that their work reveals a rich biosynthetic potential of the coral microbiome that is important for the host and the coral reef. I still do not see evidence supporting this claim in the current manuscript. The authors shifted the focus to biotechnological applications of the newly discovered enzyme, which is definitely an interesting finding on its own but far removed from the goal of the manuscript: the importance of the coral microbiome and the molecules it produces to the reef ecosystem. A similar enzyme could have been discovered from a soil or open ocean microbiome, and the biotechnological advance would have been exactly the same. Nothing here is specific to the host organism. For example, the BGC/gene they actually use for engineering in this paper (The/TheO) actually comes from this *Thermoanaerobaculia* (accession number GCA_035278895.1) that is obtained from a soil metagenome.

With all of this stated, I would like to end by saying that I do not have to be convinced for this paper to be accepted, nor should the authors try to do so. The authors have made their case already on what to include and what not to include to support their claims, and I totally respect that. In fact, there seems to be a consensus among the authors and other reviewers about the importance of this work as it stands and the level of support for the stated claims. Honestly, it does not matter that I am still not convinced that the main claim is supported and that I would have liked to see a true symbiont connected to a detectable coral molecule connected to a bioactivity. I am truly sincere in saying that convincing me is not important, and that it should not delay the paper if the editors and other reviewers are on board. In science, we are all allowed to have different opinions in our perception of which claims are strongly supported and which are not. One person's opinion should be respected, yes, but it should not halt the decision of moving forward with an important publication like yours, reporting an unprecedented dataset and involving dozens of great scientists who worked extremely hard to bring it to fruition.

We thank the reviewer for their thoughtful comments and for acknowledging the substantial effort behind this work.

We appreciate the opportunity to address the reviewer's lingering concerns regarding our manuscript's framing and claims. We would like to clarify several points:

Firstly, we wish to emphasise that we have not claimed "that [the biosynthetic potential of the coral microbiome] is important for the host and the coral reef" or stated "the goal of the

manuscript: the importance of the coral microbiome and the molecules it produces to the reef ecosystem” as suggested. Rather, our work promotes “coral microbiomes as reservoirs of novel genomic and biosynthetic diversity”. Our conceptual framing distinguishes “symbiosis” to refer to +/0, +/+, or +/- relationships, and “association” to describe cases where strains are enriched or occur predominantly within a given habitat, while a 0/0 relationship cannot be excluded. To clearly reflect this, we have carefully refined the terminology throughout the manuscript in the previous round, replacing “interactions” with “associations” where appropriate, and have refrained from using the terms “symbiosis/symbionts” or “ecology/ecological relevance” when referring to our results. Additionally, we have made several revisions to the manuscript as detailed below to avoid misinterpretation.

Secondly, in their statement, the reviewer overlooks the fact that we (i) experimentally characterised the first RiPP modifications from Acidobacteriota and (ii) described a novel RiPP maturase enzyme family both discovered from coral-associated MAGs. To demonstrate the broader biotechnological potential of this enzyme family, as noted by the reviewer, we expanded our characterisation to a lanthionine synthetase-independent homologue identified in a MAG assembled from a soil metagenomic sample. We now clearly state the origin of this unclassified *Thermoanaerobaculia* species in our revised manuscript on lines 339–342.

We believe these revisions appropriately frame our contributions as the discovery and characterisation of novel biosynthetic enzymology from coral-associated bacteria, with potential biotechnological applications, without overstating ecological or functional implications for the coral holobiont.

ACTIONS TAKEN:

- We revised the title, exchanging “unique” for “novel” to avoid implying host- or ecosystem-specific exclusivity.
- We replaced “microbial symbionts of” with “microorganisms associated with” (Abstract).
- To clearly separate the reef-derived discovery from the non-reef biotechnological application, we rephrased the abstract as follows:

Original text:

Among the biosynthetically rich bacteria in the reef microbiome, we identified new groups of Acidobacteriota associated with coral reef holobionts that encode previously unknown enzymology with promising avenues for functional protein engineering.

Revised text:

Among the biosynthetically rich bacteria in the reef microbiome, we identified new groups of Acidobacteriota that encode previously unknown enzymology, in turn opening promising avenues for functional protein engineering.

- To clearly communicate our conceptual framing, we rephrased lines 291–296:

Original text:

Although the nature and specificity of this relationship remain to be further explored, our findings demonstrate that reef-building corals host biosynthetically rich bacterial

lineages, including *Acidobacteriota* spp., that emerge as new biotechnological targets²⁶ given their untapped yet promising potential to yield new natural products.

Revised text:

Our findings demonstrate that reef-building corals are enriched in host-specific (Fig. 2) and biosynthetically rich bacterial lineages (Fig. 3), including *Acidobacteriota* spp. (Fig. 4). While the nature and specificity of their relationship with corals remain to be further explored to validate true symbiotic interactions, several uncultivated clades of coral-associated *Acidobacteriota* emerge as new biotechnological targets²⁶ given their untapped yet promising potential to yield new natural products.

- We added a clear disclaimer regarding the origin of TheA/TheO used for biotechnological exploration to the main text:

Original text:

Prompted by these prospects, we further explored the diversity of RiPP-associated GMC-family oxidoreductases with the aim to find a lanthionine synthetase-independent homologue. We identified a promising candidate (Methods) in an unclassified *Thermoanaerobaculia* species (MAG, GenBank accession: GCA_035278895.1), which we termed the 'the' cluster.

Revised text:

Prompted by these prospects, we further explored the diversity of RiPP-associated GMC-family oxidoreductases beyond coral reefs with the aim to find a lanthionine synthetase-independent homologue. We identified a promising candidate (Methods) in an unclassified *Thermoanaerobaculia* species recovered from a soil metagenomic sample (MAG, GenBank accession: GCA_035278895.1), which we termed the 'the' cluster.

Referee #3 (Remarks to the Author): Natural products biosynthesis:

The authors have created an extensive database of over 13,000 metagenome-assembled genomes (MAGs) and over 1,700 biosynthetic gene clusters from coral and sponge microbiomes. This resource is of broad value to the scientific community. The authors have clarified their comparison between coral and sponge microbiomes by normalizing for sampling effort. The revised analysis shows that the coral microbiome is richer in BGCs per genome and per microbial species compared to sponges, which is a key finding. The revision strengthens the manuscript by providing significant new experimental data on the unique RiPP maturase enzyme TheO from a *Thermoanaerobaculia* species, demonstrating its unique ability to form thiazoles without a lanthionine synthetase. This includes demonstrating its relaxed substrate specificity, showing that a large portion of the peptide is dispensable for recognition, and successfully installing thiazoles on an engineered hybrid peptide. This reviewer think that this version of revision significantly addressed most of the comments raised.

I still have two minor issues below:

The title's reference to "Biodiversity loss" was not well-supported by evidence or datasets in the manuscript itself. The authors should clarify this claim or remove it to ensure the title accurately reflects the manuscript's content.

Some of the methodology, particularly for the ecological analyses, could be more clearly summarized in the main text to improve readability, as the authors acknowledge they have moved some detailed descriptions to the supplementary information.

We thank the reviewer for their continued engagement and supportive assessment of the revised manuscript. Regarding the remaining minor points:

Firstly, we agree with the reviewer's concern. We had already revised the title in the previous round to remove the phrase "biodiversity loss," ensuring that the title accurately reflects the scope and evidence presented in the manuscript. The current title no longer includes this term. We additionally note that, as per editorial request, we replaced "unique" with "novel" to shorten the title by one character.

Secondly, in line with the reviewer's suggestion, we have added methodological details to the ecological analyses in the main text, which improves the accessibility.

ACTIONS TAKEN:

- We confirm that we removed "biodiversity loss" from the title (implemented in previous revision).
- As noted above, we additionally replaced "unique" with "novel" in the title to meet editorial requirements.
- We added methodological details to the ecological analyses:

Original text:

Acidobacteriota lineages, including those from which we recovered BGC-rich genomes, were abundant and prevalent across the three coral genera (Fig. 4c), potentially influencing coral hosts, as indicated by their impact on Pocillopora gene expression (Suppl. Information; Suppl. Fig. 3bc; Suppl. Table 8), enrichment of ELPs (Suppl. Information; Suppl. Fig. 2b), and transcriptional and metabolic responsiveness to coral-derived compounds (Suppl. Information; Suppl. Fig. 4).

Revised text:

Acidobacteriota lineages, including those from which we recovered BGC-rich genomes, were abundant and prevalent across the three coral genera (Fig. 4c), potentially influencing coral hosts, as indicated by their impact on Pocillopora gene expression (using a generalised dissimilarity model; Supplementary Information; Extended Data Fig. 3bc; Supplementary Table 8), enrichment of ELPs (Supplementary Information; Extended Data Fig. 2b), and transcriptional and metabolic responsiveness to coral-derived compounds (demonstrated through a co-cultivation experiment, RNA-sequencing, and high-performance liquid chromatography coupled to heated electrospray ionisation high-resolution tandem mass spectrometry; Supplementary Information; Extended Data Fig. 4).